# CEP192 localises mitotic Aurora-A activity by priming its interaction with TPX2

James Holder [ID][1,7], Jennifer A Miles [ID][2,3,7], Matthew Batchelor [ID][2,3], Harrison Popple [ID][1], Martin Walko[2,4], Wayland Yeung[5], Natarajan Kannan[5], Andrew J Wilson [ID][6], Richard Bayliss [ID][2,3 ✉] & Fanni Gergely [ID][1 ✉]

## Abstract

**Aurora-A is an essential cell-cycle kinase with critical roles in mitotic entry and spindle dynamics. These functions require binding partners such as CEP192 and TPX2, which modulate both kinase activity and localisation of Aurora-A. Here we investigate the structure and role of the centrosomal Aurora-A:CEP192 complex in the wider molecular network. We find that CEP192 wraps around Aurora-A, occupies the binding sites for mitotic spindle-associated partners, and thus competes with them. Comparison of two different Aurora-A conformations reveals how CEP192 modifies kinase activity through the site used for TPX2-mediated activation. Deleting the Aurora-A-binding interface in CEP192 prevents centrosomal accumulation of Aurora-A, curtails its activation-loop phosphorylation, and reduces spindle-bound TPX2:Aurora-A complexes, resulting in error-prone mitosis. Thus, by supplying the pool of phosphorylated Aurora-A necessary for TPX2 binding, CEP192:Aurora-A complexes regulate spindle function. We propose an evolutionarily conserved spatial hierarchy, which protects genome integrity through fine-tuning and correctly localising Aurora-A activity.**

**Keywords** Mitosis; Aurora-A; Kinase; Mitotic Spindle; Centrosome
**Subject Categories** Cell Cycle; Post-translational Modifications & Proteolysis; Structural Biology

## Introduction

During the division of a eukaryotic cell, a striking change in form occurs as intracellular structures undergo a significant transformation. The centrosome, a cytoplasmic organelle, comprises a centriole pair embedded in a protein-rich matrix (PCM) responsible for microtubule nucleation. In preparation for mitosis, the centrosome must duplicate to help establish the spindle poles, and the resulting two centrosomes then mature by recruiting additional gamma-tubulin ring complexes (γ-TuRC) to increase their microtubule nucleation capacity. As the two centrosomes separate, the interphase microtubule network is reorganised into a mitotic bipolar spindle, a molecular apparatus responsible for the alignment and faithful segregation of chromosomes. These orchestrated cellular processes are regulated by large-scale yet precise protein phosphorylation and dephosphorylation events that are catalysed by specialised protein kinases. Members of the Aurora family of kinases are ubiquitous in eukaryotes and they are central to the assembly and proper functioning of the mitotic spindle (Carmena and Earnshaw, 2003). Inhibition or disruption of their function can lead to faulty chromosome segregation (Ditchfield et al, 2003; Hauf et al, 2003; Kallio et al, 2002; Murata-Hori and Wang, 2002; Roghi et al, 1998). Notably there are three Aurora kinases in humans, Aurora-A, -B and -C, although the latter two kinases have largely overlapping functions, albeit in different tissue types. Aurora-A is located at the spindle poles and along the surrounding microtubules and is critical in ensuring that spindles have a robust, bipolar morphology. Aurora-B/C are found at the kinetochores, the contact points between microtubules and chromosomes, and are critical in the spindle assembly checkpoint that ensures a symmetric connection between paired sister chromatids and the two spindle poles. The localisation of Aurora kinases depends on interactions with specific partners: Aurora-B/C associates with INCENP, whereas Aurora-A associates with CEP192 at the centrosome and TPX2 on spindle microtubules (Joukov et al, 2010; Joukov et al, 2014; Kufer et al, 2002). Deciphering the factors that govern the distribution and balance of activity of Aurora-A at these two locations, and dissecting their specific contributions to spindle assembly, have been long-standing goals in understanding the molecular mechanisms of spatiotemporal control during mitotic progression.

Like many protein kinases, Aurora-A becomes activated through autophosphorylation on a specific residue in its activation loop (T288). This autophosphorylation process is blocked by ATP-competitive inhibitors of Aurora-A, but autophosphorylation is inefficient except when the kinase interacts with a specific binding partner. TPX2 was the first protein to be identified as an Aurora-A activator and is needed to sustain high mitotic Aurora-A phosphorylation. The Aurora-A substrate TACC3 is also an

[1]Department of Biochemistry, University of Oxford, Oxford, United Kingdom. [2]Astbury Centre for Structural Molecular Biology, University of Leeds, Leeds, United Kingdom. [3]School of Molecular and Cellular Biology, Faculty of Biological Sciences, University of Leeds, Leeds, United Kingdom. [4]School of Chemistry, Faculty of Engineering and Physical Sciences, University of Leeds, Leeds, United Kingdom. [5]Department of Biochemistry & Molecular Biology, University of Georgia, Athens, GA, USA. [6]School of Chemistry, University of Birmingham, Birmingham, United Kingdom. [7]These authors contributed equally: James Holder, Jennifer A Miles. ✉E-mail: r.w.bayliss@leeds.ac.uk; fanni.gergely@bioch.ox.ac.uk

activator, and its subsequent phosphorylation is vital for recruitment to the mitotic spindle in a complex with clathrin heavy chain and the microtubule polymerase ch-TOG (Booth et al, 2011; Burgess et al, 2018; Burgess et al, 2015). However, most phosphorylated Aurora-A is concentrated at spindle poles and centrosomes, where CEP192 is thought to promote its activation through concentration (Joukov et al, 2010; Joukov et al, 2014). CEP192, a large PCM scaffolding protein, also interacts with and recruits Polo-like kinase 1 (PLK1) to centrosomes, another essential cell cycle kinase with a central role in centrosome maturation (Dobbelaere et al, 2008; Joukov et al, 2010; Lane and Nigg, 1996; Sunkel and Glover, 1988; Woodruff et al, 2015). PLK1, in turn, phosphorylates CEP192 at several residues to generate further γ-TuRC attachment sites (Alvarez-Rodrigo et al, 2019; Joukov et al, 2014; Meng et al, 2015; Ohta et al, 2021). Additionally, BORA has a specialised role in promoting Aurora-A-mediated activation of PLK1 during mitotic entry (Seki et al, 2008b; Tavernier et al, 2021). Spatiotemporal control of Aurora-A activity, therefore depends on several scaffolding proteins and co-activators that tether and modify kinase activity according to cellular needs. Despite considerable insight into individual Aurora-A:co-activator complexes, the precise relationship between these distinct kinase pools is yet to be determined.

## Results

### Activation-loop phosphorylation of Aurora-A in mitosis requires both CEP192 and TPX2

To establish whether crosstalk exists between different Aurora-A pools, we began by assessing the impact of each Aurora-A co-activator on the localisation and autophosphorylation of the kinase during mitosis. Since Aurora-A is upregulated in certain cancer cell lines, which could influence the stoichiometry and function of the kinase:co-activator complexes, expression levels of these factors (i.e. Aurora-A, CEP192, TPX2, TACC3 and BORA) were determined across cell lines (Fig. EV1A). Telomerase-immortalised untransformed hTERT-RPE1 (RPE1, a normal diploid cell line) and U251 glioblastoma cells showed comparable expression levels for all five proteins and hence were selected for this study. Depletion of Aurora-A co-activators from asynchronous U251 cells, particularly that of BORA, CEP192 and TPX2, led to increased levels of mitotic phosphorylations (PPP1CA-pT320, PRC1-pT481 and Histone-H3-pS10), indicative of delayed mitotic progression (Fig. EV1B). As expected, TPX2 depletion by RNA interference (siRNA) strongly reduced activation loop phosphorylation (pT288) of Aurora-A in mitotic U251 and RPE1 cell lysates, and also decreased total Aurora-A protein levels (Figs. 1A,B and EV1C,D) (Eyers et al, 2003; Giubettini et al, 2011). Depletion of CEP192 triggered a comparable reduction in pT288 Aurora-A levels to that of TPX2, whereas BORA and TACC3 depletion had no discernible effect. In the former, normal pT288 Aurora-A levels could be due to BORA activating Aurora-A in G2, and not in mitosis (Seki et al, 2008b; Tavernier et al, 2021), whereas the co-activator function of TACC3 may be restricted to a minor (TACC3-bound) pool of the kinase (Burgess et al, 2018).

Because CEP192 and TPX2 localise to distinct cellular compartments (centrosome and spindle microtubules, respectively), we

analysed the distribution of total and pT288 Aurora-A during mitosis. Kinase activity was also assayed by detecting an Aurora-A phosphosite in the centrosomal LATS2 protein (S83). In control cells, clear centrosomal signals were seen for phospho-LATS2-S83 and pT288, both obliterated by Aurora-A inhibitor treatment (Fig. 1C). Depletion of TPX2 or CEP192, despite difficulties in achieving complete loss of the CEP192 centrosomal pool (Fig. EV1E), diminished the intensity of pT288 signal whilst also reducing LATS2 phosphorylation (Fig. 1C–E). Closer examination revealed a tight dot-like pT288 signal at the centrosome of TPX2-depleted cells, whereas in CEP192-depleted cells a weak and diffuse pT288 signal was detectable at spindle poles.

Consistent with previous studies, depletion of TPX2 (Giubettini et al, 2011; Kufer et al, 2002) diminished levels of spindle pole-associated Aurora-A, and a significant drop was also seen upon Aurora-A inhibition and CEP192 depletion (Figs. 1F,G and EV1E). Despite more Aurora-A being retained on spindles of CEP192-depleted cells relative to TPX2 depletion, the degree of reduction in pT288 signal was comparable between CEP192- and TPX2-depleted spindle poles (Fig. 1D,G).

Collectively, these findings point to a more substantial contribution of CEP192 to activating and localising Aurora-A in mitosis than previously appreciated. However, due to pleiotropic roles of CEP192 in centriole biogenesis, centrosome maturation and microtubule nucleation, this impact of CEP192 depletion on Aurora-A may be indirect. A direct relationship can be proven only by exclusively targeting the pool of CEP192 associated with the kinase, an approach that necessitated structural characterisation of the Aurora-A:CEP192 binding interface.

### Human CEP192 468–533 is required for a high-affinity interaction with Aurora-A

The first step towards the structure of the complex was to map the regions of the two proteins required for the interaction. Initial co-precipitation and co-purification experiments confirmed the interaction between purified Aurora-A kinase and CEP192 (Appendix Fig. S1). To facilitate biophysical interaction studies, a more stable variant of the Aurora-A kinase domain was used that lacks surface cysteines (Aurora-A$^{CAKD}$; aa 122–403 C290A, C393A) (Burgess and Bayliss, 2015). Using NMR spectroscopy, titration of Aurora-A$^{CAKD}$ into $^{15}$N-labelled CEP192 $_{442-533}$ resulted in reduced intensity for a subset of $^1$H-$^{15}$N Heteronuclear Single Quantum Coherence (HSQC) peaks, corresponding to aa 470–471, 482–531 (Fig. EV2A,E; Appendix Fig. S2). Hydrogen-deuterium eXchange mass spectrometry (HDX-MS) also showed a consistent set of interactions involving CEP192 480–490 and 507–530 (Fig. EV2C,E). Furthermore, sites all around the N-lobe of Aurora-A were most protected from the exchange, including the region above the glycine-rich loop (β1-β2) and the F-pocket that interacts with TPX2 (Fig. EV2D, named such as F16 and F19 from TPX2 interact with this site) (McIntyre et al, 2017).

To obtain reliable binding affinity data via isothermal titration calorimetry (ITC), we used a well-behaved, inactive, unphosphorylated Aurora-A kinase domain (Aurora-A$^{M3KD}$, which is Aurora-A$^{CAKD}$ with an additional D274N mutation). CEP192$_{442-533}$ had a $K_d$ of 72 nM for Aurora-A$^{M3KD}$ (Fig. EV2B). This is a higher affinity than observed in a previous study that used CEP192 proteins fused to MBP, a large protein tag that might have affected

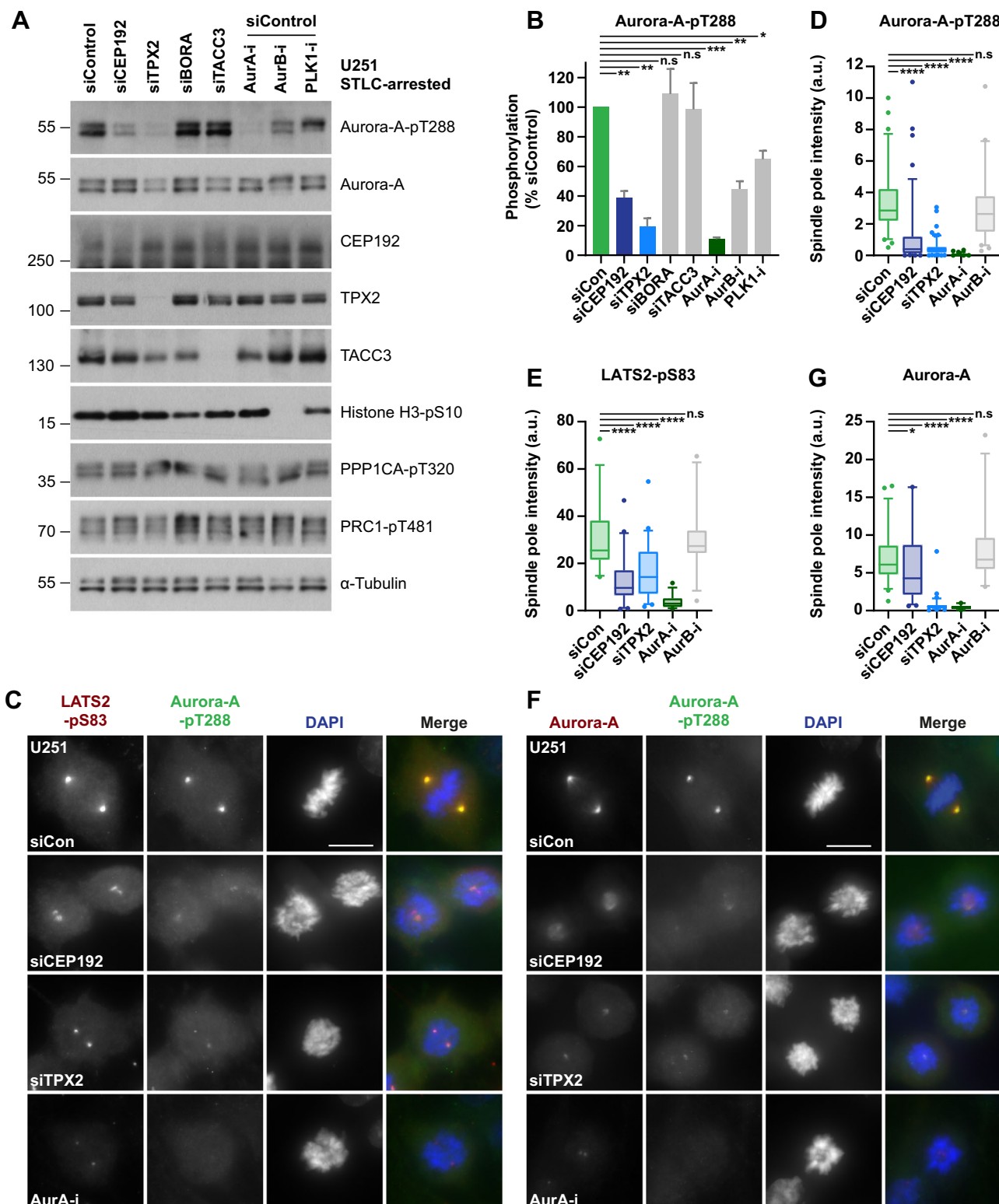

the interaction (Park et al, 2023). For context, the affinity of Aurora-A for its other key mitotic partners are: $TPX2_{1-43}$ ($K_d$ 0.3–2 µM from ITC) and $TACC3_{519-563}$ ($K_d$ ~6 µM from fluorescence polarisation) (Burgess et al, 2015; McIntyre et al, 2017; Zorba et al, 2014).

Thus, in summary, using three tag-free solution based techniques we discovered that the interaction involves two regions within CEP192 and the N-lobe of Aurora-A, and is more extensive than was reported in a crystal structure of Aurora-A kinase domain fused to a CEP192 peptide (aa 506–527) (Park et al, 2023). We

**Figure 1.   Activation-loop phosphorylation of Aurora-A in mitosis requires both CEP192 and TPX2.**

(A) Western blot analysis of U251 cells treated with the indicated siRNA (48 h total) and arrested in mitosis with STLC (20 h), prior to the addition of a proteasome inhibitor, MG-132 (20 min). Cells were then additionally treated with either DMSO control or one of Aurora-A, Aurora-B or PLK1-inhibitors for 30 min prior to lysis. (B) Densitometric quantification of Aurora-A-pT288 signal from (A). Grey bars indicate mean ± S.D ($n = 3$ biological replicates). Exact $p$ values (L-R): 0.0051, 0.0049, 0.651, 0.9399, 0.0002, 0.0087, 0.0255. (C) Immunofluorescence images of U251 cells treated with the indicated siRNA (48 h) and either DMSO control or Aurora-A or -B inhibitor (30 min) prior to methanol fixation. Antibodies against LATS2-pS83 and Aurora-A-pT288 are red and green in merged images, respectively, with DNA stained with DAPI (blue). (D, E) Box plots of (D) Aurora-A-pT288 ($n = 6$) or (E) LATS2-pS83 ($n = 3$) spindle pole signal intensity in U251 cells, with representative images shown in (C) (≥10 cells/ biological replicate). Exact $p$ values from (D) (L-R): <0.0001, <0.0001, <0.0001, 0.1721. Exact $p$ values from (E) (L-R): <0.0001, <0.0001, <0.0001, 0.9631. (F) Immunofluorescence images of U251 cells treated as in (C) prior to methanol fixation. Antibodies against Aurora-A and Aurora-A-pT288 are red and green in merged images, respectively, with DNA stained with DAPI (blue). (G) Box plot of Aurora-A spindle pole signal intensity in U251 cells, with representative images shown in (F) ($n = 3$, ≥10 cells/biological replicate). Exact $p$ values from (L-R): 0.0274, <0.0001, <0.0001, 0.2671. Data information Two adjacent mitotic cells are shown in (C) siCEP192 and (F) siCEP192 and siTPX2. Box plots in (D, E and G) indicate the median and interquartile ranges (25th–75th percentile) with coloured whiskers representing 5th–95th percentile ranges. $p$ values are denoted as follows: ****$p < 0.0001$, ***$p < 0.001$, **$p < 0.01$, *$p < 0.05$, n.s not significant (B Welch's $t$-test, D, E, G Mann–Whitney test). Scale bars in (C) and (F) represent 10 μm. Source data are available online for this figure.

concluded that the minimal binding region of CEP192 is aa 468–533, based on the NMR and HDX-MS studies.

## CEP192 wraps around the N-lobe of the Aurora-A kinase domain

Initial attempts to crystallise the Aurora-A kinase domain with CEP192$_{468-533}$ were unsuccessful, but we noted the prevalence of Aurora-A crystals in which a packing interaction obscured part of the CEP192 interaction site on the N-lobe determined by HDX-MS (Fig. EV2D). An Aurora-A variant with additional mutations in the complementary surface of the C-lobe to disrupt this contact was generated (Aurora-A$^{M7KD}$, 122–403 D274N C290A C393A N332A Q335A T347A D350A) and its interaction with CEP192$_{468-533}$ was validated (Appendix Fig. S3A) (Burgess and Bayliss, 2015). Aurora-A, with or without the contact mutations, interacts with similar affinity to the helical region of fluorescein labelled CEP192$_{501-533}$ (FAM-CEP192$_{501-533}$) in a direct, fluorescence anisotropy binding assay (Appendix Fig. S3B, Aurora-A$^{M7KD}$ $K_d$ 1.25 μM ± 0.15 μM and Aurora-A$^{CAKD}$ $K_d$ 1.17 μM ± 0.07 μM).

Aurora-A$^{M7KD}$ and CEP192$_{468-533}$ crystallised in a new form, but the diffraction was poor. We explored the addition of further Aurora-A binding partners to the complex to provide extra surfaces for crystal contact formation. Robust crystals that diffracted to 2.76 Å were formed with Mb2, an inhibitory monobody that interacts with the TPX2-binding Y-pocket in Aurora-A and does not affect the interaction of Aurora-A$^{M7KD}$ and CEP192$_{468-533}$ (Appendix Fig. S3C,D) (McIntyre et al, 2017; Zorba et al, 2019). The structure was solved by molecular replacement and the asymmetric unit (ASU) comprised two copies of each of the three proteins. Unambiguous electron density for CEP192$_{468-533}$ was observed and almost the entire sequence was modelled, wrapped around the N-lobe of Aurora-A$^{M7KD}$ chain A (Figs. 2A and EV3). The beta-hairpin of CEP192 predicted by NMR points out away from the surface of the kinase, followed by a region lacking a secondary structure that interacts with the F-pocket of Aurora-A and then wraps around the N-lobe, leading to two helices that interact with the kinase above the glycine-rich loop, a short helix between 502 and 507 (αS) and a long helix between 512 and 529 (αL) (Fig. 2A). In contrast to αL, αS was not predicted in the AlphaFold2 model of CEP192 (Fig. EV2C). The interface between Aurora-A$^{M7KD\_A}$ and CEP192$_{468-533\_C}$ is 1702.9 Å², involving 39 residues from CEP192$_{468-533\_C}$ and 54 residues from Aurora-A$^{M7KD\_A}$.

In the second complex in the ASU, only residues CEP192$_{506-527}$ could be modelled into the density (CEP192$_{506-527\_F}$, Fig. EV3A). No binding of CEP192 was observed in the F-pocket of this copy of Aurora-A as aa 468–505 were disordered. The buried surface area between the two chains with the smaller interface is 432.2 Å², only 25% of that observed in the other copy of the complex in the crystal, and involves 12 residues from CEP192$_{506-527\_F}$ and 22 residues from Aurora-A$^{M7KD\_D}$. The RMSD over 94 atoms in both copies of αL of CEP192 bound to their respective Aurora-A is 0.41 Å.

Our analysis focussed initially on the complex with the more extensive interface because that is consistent with the in-solution data collected on the complex. CEP192$_{468-533}$ residues L484, Y486, Y487 and F490 make significant contributions to binding at the F-pocket site (Fig. 2B,C), with F490 from CEP192 having the highest predicted solvation energy effect of 2.37 kcal/mol compared to other residues in the interface (as estimated using PDBePISA (Proteins, Interfaces, Structures and Assemblies, (Krissinel and Henrick, 2007)). Within the F-pocket of Aurora-A, residues W128 and R126 form hydrogen bonds with Y486/L484 and Y487 of CEP192, respectively.

On the other side of the Aurora-A N-lobe, F508 from CEP192 is inserted into a pocket formed by the side chains of E134, L149, R151 and I158 from Aurora-A (Fig. 2B,D). Residues I518, E522, F525 and H529 of CEP192 lie along one side of αL in the $i$, $i + 4$, $i + 7$ and $i + 11$ positions, interacting with residues in the glycine-rich loop of Aurora-A. Further hydrogen bonds occur between D521 and E522 of the CEP192 long helix and Aurora-A residues Y148 and T204/R205 respectively. A salt bridge is formed between Glu522 of CEP192 and Arg205 of Aurora-A.

Having determined the molecular structure of CEP192 bound to Aurora-A, which wraps around the N-lobe of the kinase, burying a large surface area, we next compared the structure with previously determined structures of Aurora-A in mitotic complexes.

## CEP192 competes for binding with both TPX2 and TACC3 by using two distinct binding sites

The binding site of CEP192 on Aurora-A overlaps significantly with that of TPX2 and TACC3 (Fig. 2B). At the F-pocket site, residues Y487 and F490 in CEP192 are structurally equivalent to

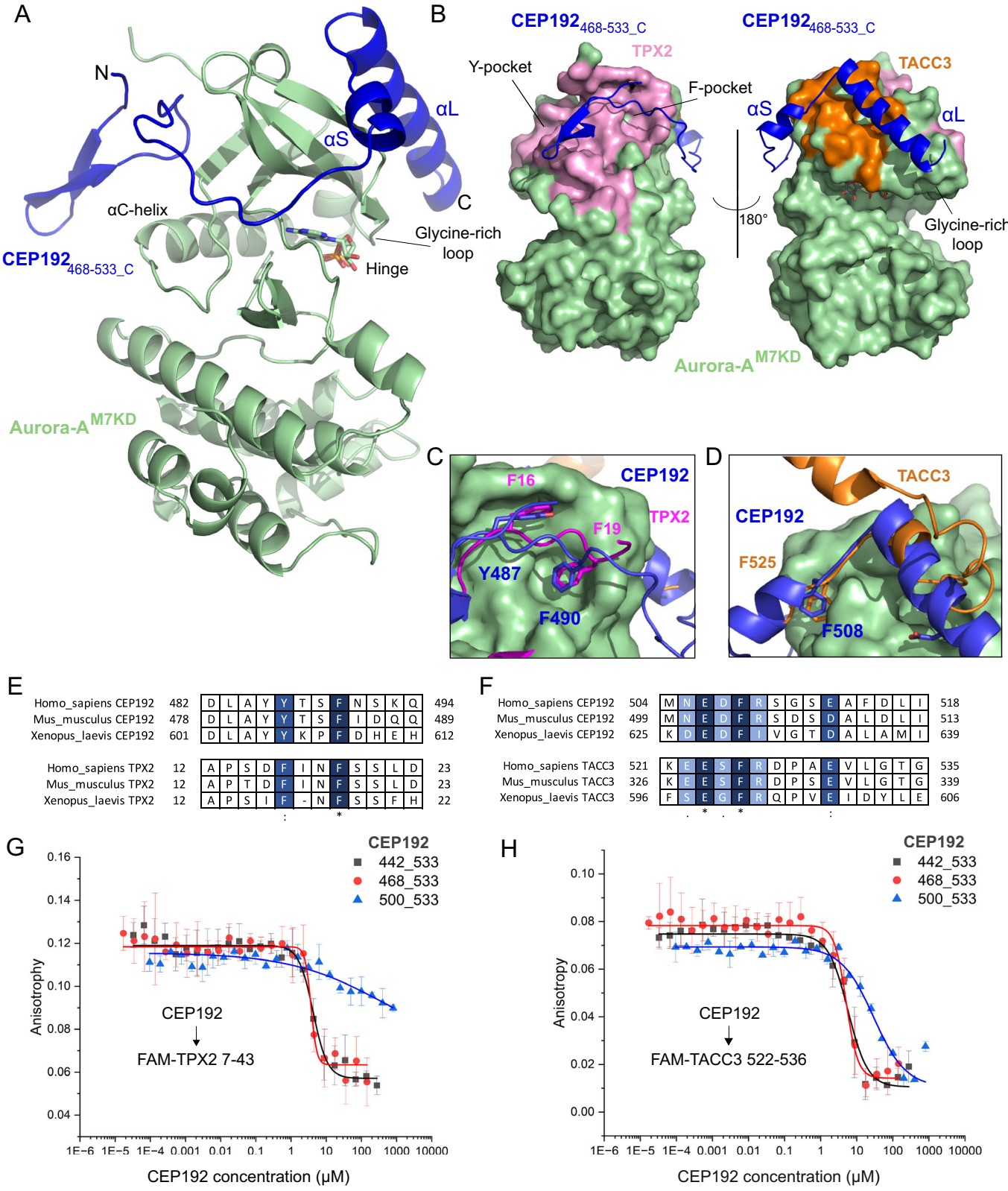

**Figure 2.  Human CEP192 468–533 wraps around the N-lobe of Aurora-A kinase.**

(A) Cartoon representation of the crystal structure of CEP192$_{468-533\_C}$ (dark blue), bound to Aurora-A$^{M7KD\_A}$ (light green). (B) Comparison of the structure of CEP192$_{468-533}$ bound to Aurora-A$^{M7KD}$ with the surfaces that interact with other known binders in mitosis highlighted. Residues required for interacting with TPX2 are shown in pink (from PDB:1OL5), and TACC3 are shown in orange (from PDB:5ODT). CEP192 binding overlaps with that of TPX2 and TACC3. (C) Magnified view of CEP192$_{468-533}$ compared with TPX2$_{1-43}$ bound to Aurora-A (PDB:1OL5). The residues Phe16 and Phe19 from TPX2 (pink) overlap with CEP192 (dark blue) Tyr487 and Phe490 to bind into the F-pocket. (D) Magnified view of CEP192$_{468-533}$ compared with TACC3$_{518-563}$ bound to Aurora-A (PDB:5ODT). Phe508 of CEP192 (dark blue) binds into the same pocket on the N-lobe that is utilised by Phe525 in TACC3 (orange). (E) Sequence alignment shows the conservation of the structurally equivalent residues between orthologues of CEP192 and TPX2. Higher conservation is shown with darker shading. (F) Sequence alignment shows the conservation of the structurally equivalent residues between orthologues of CEP192 and TACC3. Higher conservation is shown with darker shading. (G) Fluorescence anisotropy-based competition assay with various CEP192 protein constructs (442–533 (black, IC$_{50}$ 4.2 μM ± 0.42 μM), 468–533 (red, IC$_{50}$ 3.7 μM ± 0.4 μM), 501–533 (blue, IC$_{50}$ not calculated)) binding to unphosphorylated Aurora-A$^{CAKD}$ in competition with FAM-TPX2 7-43. (H) Fluorescence anisotropy-based competition assay with various CEP192 protein constructs (442–533 (black, IC$_{50}$ 5.7 μM ± 1 μM), 468–533 (red, IC$_{50}$ 5 μM ± 0.5 μM), 501–533 (blue, IC$_{50}$ 29 μM ± 3 μM)) binding to Aurora-A kinase domain in competition with FAM-TACC3 522-536. Displayed data points and IC$_{50}$ values represent the average anisotropy for each reaction, with the standard deviation for the mean shown as error bars ($n = 3$ independent experimental samples). Data information: In (G) and (H), displayed data points and IC$_{50}$ values represent the average anisotropy for each reaction with the standard deviation for the mean shown as error bars ($n = 3$ independent experimental samples). Source data are available online for this figure.

F16 and F19 in TPX2 (Fig. 2C). These two aromatic residues are conserved in orthologues of CEP192 and TPX2, but the surrounding residues are divergent, and the absence of a consistent motif might explain why the binding of CEP192 into this site was not previously identified (Fig. 2E). At the other site, F508 in CEP192 is analogous to F525 in TACC3 (Fig. 2D). As well as this aromatic residue, many of the surrounding residues are also conserved, including charged residues that form key interactions (Fig. 2F). Notably, αL of CEP192 forms a more extensive interface with Aurora-A than the shorter helix in TACC3, which twists away from the surface of the kinase forming a region lacking secondary structure that leads to the Aurora-A substrate site of TACC3 (Ser558). In contrast, αL of CEP192 continues along the full length of the glycine-rich loop, with H529 in CEP192 interacting with Q168 in the αB-helix in Aurora-A through a hydrogen bond. Building on previous work showing competition for Aurora-A binding between TPX2 and CEP192 (Joukov et al, 2010), we tested various CEP192 constructs in a fluorescence anisotropy-based competition assay with Aurora-A$^{CAKD}$ and FAM-TPX2$_{7-43}$ or FAM-TACC3$_{522-536}$. While CEP192$_{468-533}$ and CEP192$_{442-533}$ competed for binding with FAM-TPX2$_{7-43}$, N-terminally truncated CEP192$_{501-533}$ did not compete (Fig. 2G). In contrast, all three CEP192 constructs competed for binding with FAM-TACC3$_{522-536}$ (Fig. 2H).

To further validate the interactions observed in the crystal structure (Fig. 3A,B), mutations were introduced into Aurora-A$^{CAKD}$ and CEP192$_{468-533}$ that were predicted to disrupt the interaction and binding affinities were determined using ITC (Fig. 3C–F). The truncated variant of CEP192$_{468-533}$ has a comparable affinity for Aurora-A to the longer construct (Fig. 3C CEP192$_{468-533}$ $K_d$ 80.6 nM compared to CEP192$_{442-533}$ with a $K_d$ of 72 nM in EV2B). A mutant of CEP192$_{468-533}$ (Y487A, F490A), in which the two residues that bind the F-pocket were replaced with alanine residues, had a binding affinity of 2 μM for Aurora-A$^{CAKD}$, over 20 times weaker compared to the wild-type CEP192$_{468-533}$ (Fig. 3D). A variant of CEP192$_{468-533}$ (F490D, F508D, I518D) that replaces hydrophobic residues involved in recognising the two regions of the kinase N-lobe completely knocked out the interaction with Aurora-A$^{CAKD}$ (Fig. 3E). Disruption of the CEP192 long helix interaction through a variant of Aurora-A$^{KD}$ (F165D, R205A) that replaces two key residues in the binding site showed a reduced binding affinity for CEP192$_{468-533}$, with a

dissociation constant of 26 ± 16 μM (Fig. 3F). Point mutants of Aurora-A$^{KD}$ were also tested in a direct FA binding assay with FAM-CEP192$_{501-533}$. The single mutants F165D and R205A, residues that form an interaction with the long helix of CEP192, had a significant effect on the interaction between Aurora-A and CEP192, with $K_d$ values >5 μM for FAM-CEP192$_{501-533}$ (Appendix Fig. S4A). The R151A mutant of Aurora-A, previously identified to reduce the binding affinity to TACC3 (Burgess et al, 2018), also showed reduced binding to FAM-CEP192$_{501-533}$ ($K_d$ > 10 μM, Appendix Fig. S4A), consistent with the similar interactions of TACC3 and CEP192 with Aurora-A at this key residue.

TPX2 has a higher affinity for phosphorylated Aurora-A, mediated through direct interaction with the activation loop (McIntyre et al, 2017). In a direct FA binding assay, the binding affinity of FAM-CEP192$_{501-533}$ for Aurora-A was similar for unphosphorylated (420 ± 244 nM) or phosphorylated kinase (370 ± 130 nM) (Appendix Fig. S4A). However, this CEP192 fragment lacks the region that binds the F-pocket so we used a competition-FA-assay format, in which CEP192$_{468-533}$ was titrated against a constant concentration of Aurora-A and tracer FAM-CEP192$_{501-533}$. No significant difference was seen in the ability of the longer CEP192$_{468-533}$ fragment to compete for binding regardless of whether Aurora-A was phosphorylated or not (Appendix Fig. S4B,C). We also used ITC to quantify the binding of CEP192$_{468-533}$ to phosphorylated Aurora-A$^{CAKD}$ (not co-expressed with lambda phosphatase, Appendix Fig. S4G). The $K_d$ was measured at 134 nM ± 16 nM (Appendix Fig. S4F), which is only around a twofold difference in affinity compared to the unphosphorylated Aurora-A. These findings are consistent with CEP192 making no contacts with the activation loop and thus having no significant preference for phosphorylated versus unphosphorylated Aurora-A.

To summarise, we have shown that the interactions of the centrosomal protein CEP192 with Aurora-A compete with spindle microtubule-associated TACC3 and TPX2 binding, and validated the crystal structure using site-specific mutagenesis. CEP192 binding to Aurora-A is not affected by its phosphorylation state, unlike TPX2 which interacts more strongly with the kinase phosphorylated on T288. These observations are consistent with the crystal structures of these complexes, and raise the question of the specific contributions of each site in CEP192 to the function of its interaction with Aurora-A.

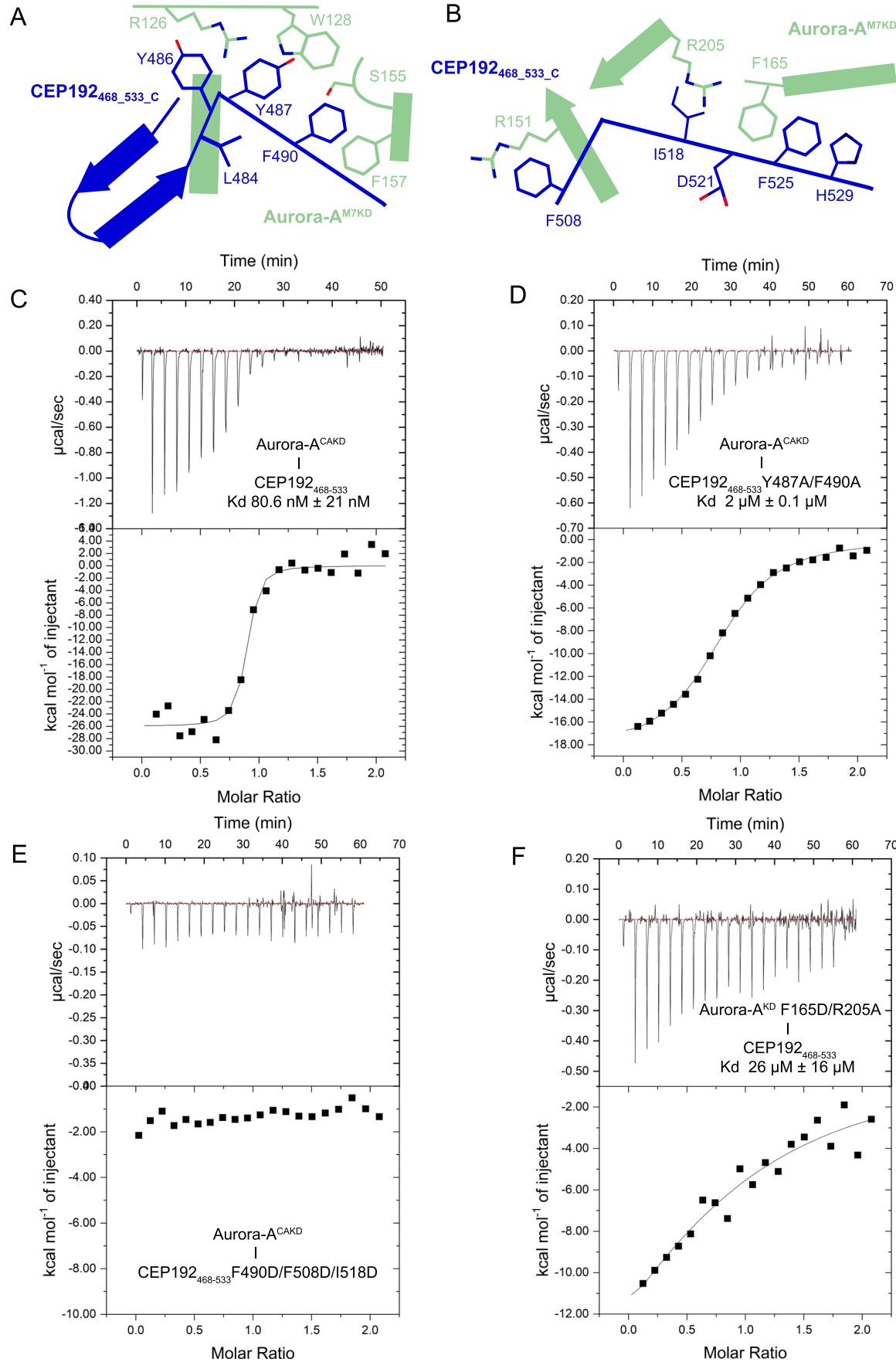

◀ **Figure 3. CEP192 interacts with Aurora-A using two sites.**

(A) Schematic diagram of the CEP192 interaction at the F-pocket site on Aurora-A kinase domain. CEP192 is shown in blue, with Aurora-A in green. (B) Schematic diagram of the CEP192 interaction at the TACC3 site above the glycine-rich loop of Aurora-A kinase domain. CEP192$_{468-533}$ is shown in blue, with Aurora-A$^{M7KD}$ in green. (C) Isothermal titration calorimetry of Aurora-A$^{CAKD}$ into CEP192$_{468-533}$. The measured Kd was 80.6 ± 21 nM with a molar ratio of 0.85. (D) Isothermal titration calorimetry of Aurora-A$^{CAKD}$ into CEP192$_{468-533}$ Y487A/F490A. The measured Kd was 2 ± 0.1 μM, with a molar ratio of 0.84. (E) Isothermal titration calorimetry of Aurora-A$^{CAKD}$ into CEP192$_{468-533}$ F490D/F508D/I518D. The Kd was not determined. (F) Isothermal titration calorimetry of Aurora-A$^{KD}$ F165D/R205A into CEP192$_{468-533}$. The measured Kd was 26 ± 16 μM, with a molar ratio of 0.87. Source data are available online for this figure.

## The structural basis of CEP192 modulation of Aurora-A kinase activity

The Aurora-A binding domain of *Xenopus* CEP192 has previously been described as an inhibitor of kinase activity (Joukov et al, 2010). We therefore, explored whether this was true for the human protein in an ADP-Glo kinase assay using phosphorylated Aurora-A$^{CAKD}$ and a standard kemptide substrate. As controls, we observed the stimulation of Aurora-A activity by TPX2$_{1-43}$ (EC$_{50}$ of 6.8 ± 0.5 nM) and that alisertib fully inhibited activity (IC$_{50}$ 3.7 ± 0.1 nM) (Appendix Fig. S4D). In the same assay, CEP192$_{468-533}$ and CEP192$_{442-533}$ partially inhibited the activity of phosphorylated Aurora-A$^{CAKD}$ with an IC$_{50}$ of 18 ± 5 nM and 32 ± 7 nM, respectively (Fig. 4A, shown in black and red, respectively). This inhibition is less potent and less complete if the residues that interact with the F-pocket on Aurora-A are mutated (Y487A F490A 468–533) (Fig. 4A, shown in blue). The shortest CEP192$_{501-533}$ peptide, spanning only αS and αL of CEP192 that bind above the glycine-rich loop, had a mildly activatory effect on the kinase activity (Fig. 4A, shown in green), but was much less effective than TPX2 (Appendix Fig. S4D). Our results differ from a previous study that observed no effect of CEP192 protein fragments on the activity of Aurora-A (Park et al, 2023). However, there are differences in the assay conditions and fragments of CEP192 used. For example, our shorter CEP192 fragment includes the αS sequence, which is absent from the fragment used in the previous study (aa 506–536).

To understand why binding of the longer fragments of CEP192 reduced the activity of Aurora-A, we compared the two copies of Aurora-A$^{M7KD}$ in the ASU that are bound to either the long region CEP192$_{468-533}$ or a shorter region CEP192$_{506-527}$. The two copies have a similar conformation overall (RMSD of 1.0 Å over all Cα atoms), however, there are clear differences in the position of the DFG-motif and the assembly of the R-spine (Fig. 4B and Appendix Fig. S5A,B).

In Aurora-A (chain A), where CEP192$_{468-533}$ is wrapped around the N-lobe, the R-spine is broken and the DFG-motif is in a DFG-inter/DFG-up position (Fig. 4B; Appendix Fig. S5B), characteristic of the inactive state of the kinase (Cyphers et al, 2017). This is similar to the published structure of Aurora-A in complex with Mb2 (Fig. EV5A,C, PDB: 6CPG and 6C83 (Pitsawong et al, 2018; Zorba et al, 2019) or with an inhibitory vNAR (variable new antigen receptor) domain (Appendix Fig. S5A,B, PDB: 5L8L (Burgess et al, 2016). But the second copy of Aurora-A (chain D), with only aa 506–527 of CEP192 bound, was in an active DFG-in conformation and the R-spine was assembled (Fig. 4B; Appendix Fig. S5B). This resembles the structure of the αL region of CEP192 506–527 fused to the N-terminus of Aurora-A, in which all three copies in the ASU have a complete R-spine and are DFG-in (Fig. 4B) (Park et al, 2023).

In addition to binding the F-pocket, CEP192$_{468-533}$ sits on the adjacent αC-β4 region of Aurora-A, in a similar position to the activator TPX2, but with opposing biochemical and structural effects. The αC-β4 region is a key regulatory feature of protein kinases that tethers the αC-helix to the kinase core (Yeung et al, 2020). In CEP192, L484, which is just C-terminal to the β-hairpin, overlays with TPX2 P13 (Fig. 4C). This region of CEP192 or TPX2 packs against the αC-helix of Aurora-A. However, the leucine residue of CEP192 penetrates deeper into the αC/β4 interface, into a larger pocket created by movement of the αC-helix (Fig. 4C). The CEP192$_{468-533}$ (L484D) protein variant was weaker in competition with FAM-TPX2 7-43 (Appendix Fig. S4E, 19.9 μM compared to ~4 μM for WT CEP192). Consistent with the position of L484 in the interface with Aurora-A, CEP192$_{468-533}$ L484D was a weaker inhibitor of Aurora-A activity (Fig. 4A, shown in purple, IC$_{50}$ 220 ± 44 nM). We concluded that the binding of the αS/αL of CEP192 (aa 506–527) is compatible with the active conformation of Aurora-A whereas the binding of the N-terminally extended CEP192 (aa 468–505) is incompatible with the active conformation and stabilises the inactive conformation.

Taken together, these data validate the structural model of the two binding sites, and show that the long helix provides the most binding affinity, while binding to the F-pocket provides a substantial enhancement of binding. CEP192 also acts as an inhibitor of substrate phosphorylation through an allosteric mechanism based on its interaction with the F-pocket and the top of the R-spine. Structural characterisation of the Aurora-A:CEP192 binding interface enabled us to design a strategy to exclusively target the pool of CEP192 associated with the kinase.

## Deleting the Aurora-A binding interface of CEP192 in cells

The Aurora-A binding interface of CEP192 (aa 468–533) is almost entirely encoded by exons 11 and 12 of the human *CEP192* gene. The former encodes amino acids K464-G511, including the critical residues F490 and F508, and therefore removal of exon 11 is predicted to disrupt the interaction of CEP192 with Aurora-A (Fig. 5A,B). To this end, homozygous in-frame deletion of exon 11 was performed by CrispR-Cas9 with successful gene targeting confirmed in three U251 clones (UΔ11-1, UΔ11-2, UΔ11-3) whereas a clone with intact *CEP192* was taken forward as a control (UCon) (Appendix Fig. S6A). Unlike in U251, the p53 pathway is functional in RPE1 cells, and therefore to avoid potential activation of the p53/USP28/53BP1 mitotic surveillance checkpoint, p53 was co-targeted with *CEP192*(exon 11) (Lambrus et al, 2016; Lambrus and Holland, 2017; Wong et al, 2015; Zhu et al, 2008). We isolated two clones with intact *CEP192* (RCon-1 and RCon-2) and two with homozygous deletion of exon 11 (RΔ11-1, RΔ11-2; Appendix Fig.

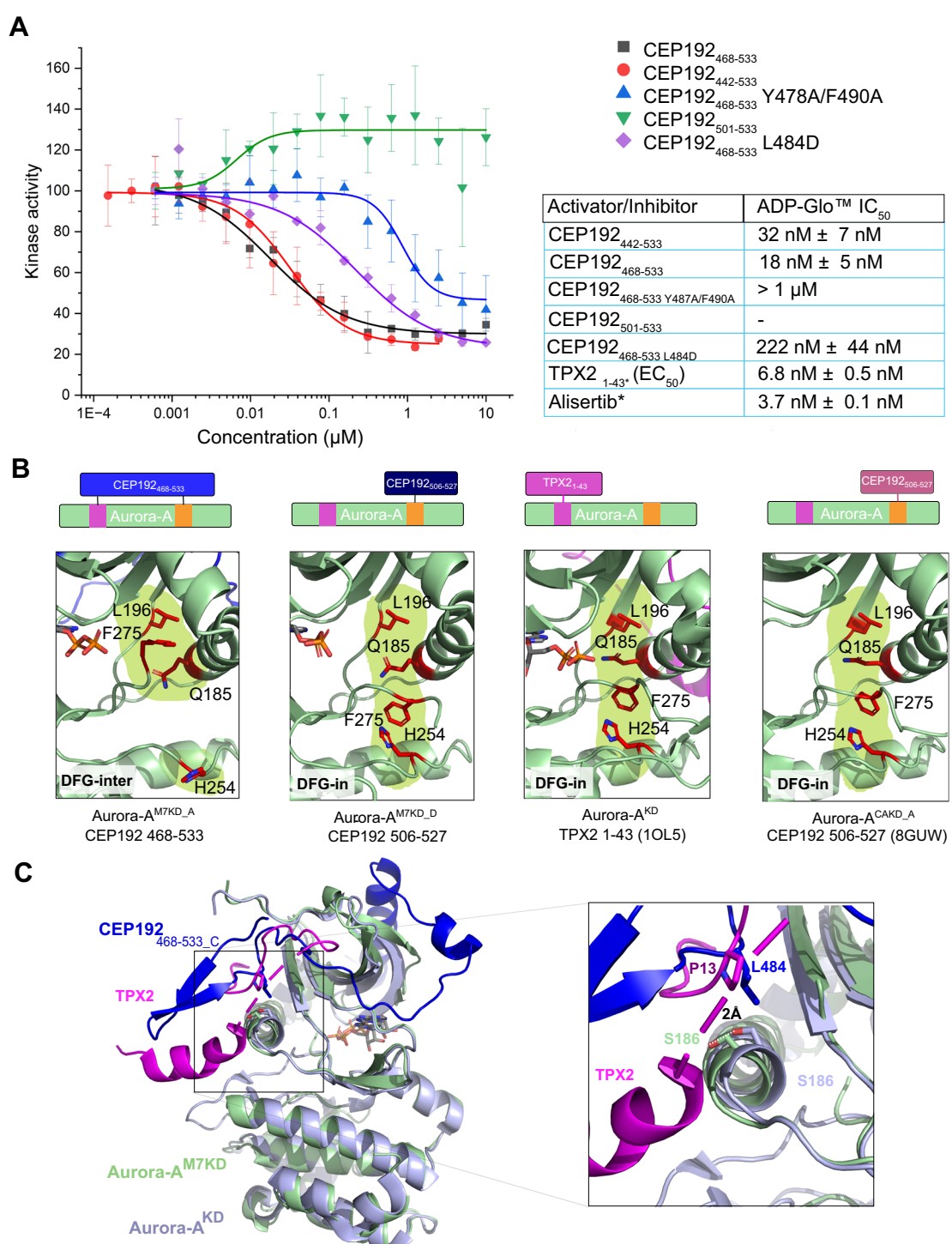

**A**

| Activator/Inhibitor | ADP-Glo™ $IC_{50}$ |
|---|---|
| $CEP192_{442-533}$ | 32 nM ± 7 nM |
| $CEP192_{468-533}$ | 18 nM ± 5 nM |
| $CEP192_{468-533\ Y487A/F490A}$ | > 1 µM |
| $CEP192_{501-533}$ | - |
| $CEP192_{468-533\ L484D}$ | 222 nM ± 44 nM |
| $TPX2_{1-43*}$ ($EC_{50}$) | 6.8 nM ± 0.5 nM |
| Alisertib* | 3.7 nM ± 0.1 nM |

Legend:
- $CEP192_{468-533}$
- $CEP192_{442-533}$
- $CEP192_{468-533}$ Y478A/F490A
- $CEP192_{501-533}$
- $CEP192_{468-533}$ L484D

**B**

Aurora-A$^{M7KD\_A}$
CEP192 468-533

Aurora-A$^{M7KD\_D}$
CEP192 506-527

Aurora-A$^{KD}$
TPX2 1-43 (1OL5)

Aurora-A$^{CAKD\_A}$
CEP192 506-527 (8GUW)

**C**

---

S6A) and confirmed all four clones to be functional null for p53 (RPE1$^{p53-/-}$, Appendix Fig. S6B). Centrosomal localisation of CEP192 lacking exon 11 (CEP192Δ11) was indistinguishable from that of wild-type CEP192 (Fig. 5C,D; Appendix Fig. S6C,D). While total centrosomal CEP192 intensities were comparable between RCon-1 and the RΔ11 clones, in RCon-2 CEP192 levels were elevated (Appendix Fig. S6D). Repeating these stainings with additional controls, such as an independently derived RPE1$^{p53-/-}$

line (C3) and the parental RPE-1 cell line confirmed RCon-2 to be an outlier, possibly reflecting clonal variations (Fig. 5D). TPX2 levels at spindle poles were indistinguishable across Rcon and RΔ11 clones (Fig. 5E).

Having confirmed that CEP192Δ11 localised normally, we tested if its binding to Aurora-A was impacted as expected. Proximity ligation assays (PLA) using CEP192 and Aurora-A antibodies enabled the detection of CEP192:Aurora-A complexes in cells with

◄ **Figure 4.  CEP192 inhibits Aurora-A kinase activity by binding to the site that competes with TPX2.**

(A) ADP-Glo assay to assess the effect of CEP192-WT and mutants binding to Aurora-A on the ATPase activity of phosphorylated Aurora-A$^{CAKD}$ with kemptide as substrate. The calculated IC/EC$_{50}$ values are summarised in the table. (B) Cartoon representation of the R-Spine assemblies in the two complexes of Aurora-A$^{M7KD}$ bound to CEP192 468–533 or 506–527 in the ASU, compared to the R-Spine in Aurora-A bound to TPX2 (PDB: 1OL5) and fused to CEP192 506–527 (PDB: 8GUW). The R-spine is not assembled when CEP192 468–533 forms extensive interactions with Aurora-A, but is assembled when CEP192 is truncated (CEP192 505–527 bound in chain F, or 506–527 fused to Aurora-A in 8GUW) or in the presence of TPX2. (C) Cartoon representation of the comparison of Aurora-A$^{M7KD}$ bound to CEP192$_{468-533}$ (green/bright blue) and Aurora-A/TPX2 (light blue/magenta) crystal structures in the vicinity of the kinase αC-β4 region. Leu484 from CEP192 (shown in blue) superimposes with Pro13 from TPX2 (shown in magenta), but penetrates deeper into the αC-β4 surface. A relative rotation of the αC-helix is needed to accommodate Leu484, including a 2.0 Å shift in the position of Ser186 in Aurora-A. Data information: Displayed data points in (A) and EC$_{50}$ values represent the average luminescence for each reaction condition with standard deviations of the mean as error bars ($n = 3$ independent experiment samples). Source data are available online for this figure.

good specificity (Appendix Fig. S6E). In control cells, bright clusters of PLA foci focussed around centrosomes and mitotic spindle poles, whereas foci were barely detectable in mitotic RΔ11 cells, consistent with a large decrease in CEP192:Aurora-A complexes (Fig. 5F,G). CEP192:Aurora-A PLA foci were also much reduced in UΔ11 cells although they remained detectable in many cells. However, residual foci appeared weak and scattered across the cytoplasm with no obvious focus around centrosomes and spindle poles (Fig. 5H,I). Controls of both cell types showed a few PLA foci outside the regions of the centrosome and spindle poles, indicating that some CEP192:Aurora-A complexes may form in the cytoplasm and/or move in and out of centrosomes.

Flow cytometry confirmed a relatively normal cell cycle distribution of RΔ11 clones with a moderate increase in the G2/M population and a corresponding reduction in the G1 population (Fig. 5J; Appendix Fig. S6F,G). By contrast, in the UΔ11 clones, G2/M populations ranged between 31 and 60% with a further 9–25% of cells showing greater than 4N ploidy (Fig. 5J; Appendix Fig. S6H). While 40% of UΔ11-1 cells had G1-like 2N ploidy, corresponding figures were 0.9 and 5.7% in UΔ11-2 and UΔ11-3 cells, respectively. By contrast, the majority of wild-type cells showed 2N ploidy (59%) with only 4% being over 4N. Therefore, tetraploidy is prevalent in all UΔ11 clones; over 90% of UΔ11-2 and UΔ11-3 and ~25% of UΔ11-1 cells are tetraploid. Indeed, the nuclear area of UΔ11 clones was significantly larger than that of wild-type cells (Appendix Fig. S6I). Because all three UΔ11 clones display increased ploidy, this is likely to be a consequence of *CEP192*(exon 11) deletion. By contrast, RΔ11 clones maintain their diploidy, suggesting that polyploidisation of UΔ11 cells is due to a genetic interaction between *CEP192*(exon 11) deletion and pre-existing sensitising mutations in U251 cells affecting components of the mitotic machinery. Since aberrant ploidy is expected to impact genome stability, centrosome numbers and mitotic spindle formation, whilst also generating considerable genetic heterogeneity, functional analyses were performed predominantly in the RPE1 clones. Indeed, polyploidy could explain the greater variability of CEP192:Aurora-A PLA foci intensities in UΔ11 vs. RΔ11 clones (Fig. 5H,I).

## Binding of Aurora-A to CEP192 is essential for its autophosphorylation and spindle recruitment

To determine the role of the CEP192:Aurora-A interaction in mitosis, we first examined levels and localisation of Aurora-A and Aurora-A-pT288 in UΔ11 and RΔ11 clones. Aurora-A levels were markedly reduced on mitotic spindles in all Δ11 clones (Fig. 6A–C); staining intensity at spindle poles was reduced by ~40% in UΔ11

and by ~70% in RΔ11 clones relative to controls (Fig. 6C). An even more striking decrease was observed in levels of pT288 Aurora-A with no detectable signal on the mitotic apparatus in any of the clones (Fig. 6A,B,D) despite some Aurora-A remaining on spindles. Mitotic cell lysates were immunoblotted to assess if deletion of *CEP192*(exon 11) also had a measurable impact on whole cellular, and not only spindle-bound, levels of total and autophosphorylated Aurora-A. There was no obvious difference in the levels of various mitotic regulators (e.g. TACC3 and PPP6C) or mitosis-specific phosphorylations (e.g. PPP1CA-T320) between control and RΔ11 lysates (Fig. 6E; Appendix Fig. S7A). RΔ11 lysates, however, displayed a reduction of over 85% in pT288 Aurora-A levels whether normalised against α-tubulin or total Aurora-A signal (Fig. 6E,F; Appendix Fig. S7B). A similar decrease in pT288 Aurora-A levels was seen in UΔ11 cells (Appendix Fig. S7C,D).

Since TPX2 recruits Aurora-A to spindle microtubules, a decrease in Aurora-A on spindles could indicate a defect in TPX2-dependent localisation of Aurora-A. Even though TPX2 levels are normal on RΔ11 mitotic spindles (Fig. 5C,E), PLA revealed a threefold reduction in Aurora-A:TPX2 complexes in mitotic cells (Fig. 6G,H). PLA intensity was also reduced in the polyploid UΔ11 clones despite comparison with a diploid control (Fig. 6I; Appendix Fig. S7E).

Our results suggest that loss of Aurora-A:CEP192 binding not only diminishes the pT288 Aurora-A pool but also precludes the formation of Aurora-A:TPX2 complexes and/or their concomitant recruitment to spindles. This impact on Aurora-A:TPX2 complexes was unexpected because purified TPX2 can bind Aurora-A and activate the kinase in vitro (Bayliss et al, 2003; Eyers et al, 2003). Our data, however, implies that in cells TPX2 preferentially binds a CEP192-dependent pool of Aurora-A, most likely carrying activation loop phosphorylation. This is consistent with the higher affinity of TPX2 for phosphorylated Aurora-A in vitro (McIntyre et al, 2017).

## Loss of Aurora-A:CEP192 interaction leads to mitotic spindle defects and chromosome segregation errors

Given that RΔ11 cells are essentially depleted of pT288 Aurora-A, the active form of Aurora-A kinase thought to drive mitotic spindle formation, we next investigated the impact of *CEP192*(exon 11) deletion on mitosis. In asynchronous fixed cells, we noted a modest increase of ~1.5–2-fold in the mitotic population when compared to controls (Fig. EV4A). Time-lapse microscopy of SiR-Hoechst-labelled cells revealed a significant delay in progression from nuclear envelope breakdown (NEBD) to anaphase onset in RΔ11 (Fig. 7A,B). Although RPE1 cells normally initiate anaphase within

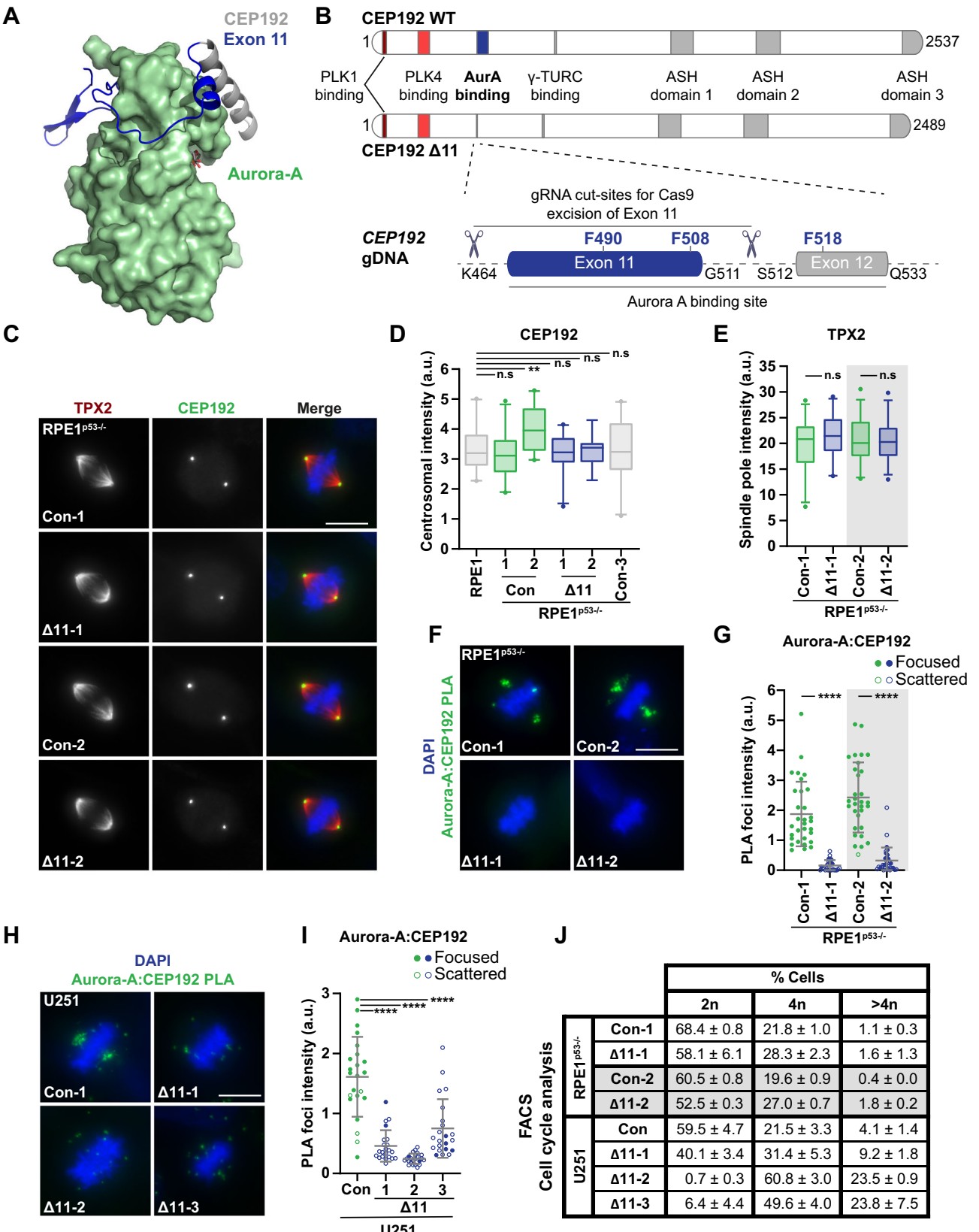

**Figure 5.    Targeted deletion of the Aurora-A binding interface of CEP192 in cells.**

(A) Crystal structure of the interaction between Aurora-A (green) and CEP192 (grey/blue). Residues contained within *CEP192* exon 11 are highlighted in blue.
(B) Schematic detailing the domain architecture of *CEP192* and the CRISPR/Cas9 approach used to remove exon 11 and generate CEP192(Δ11) mutants.
(C) Immunofluorescence images of methanol fixed control (Con) or Δ11 RPE1$^{p53-/-}$ cells. Antibodies against TPX2 and CEP192 are red and green in merged images, respectively, with DNA stained with DAPI (blue). (D) Box plot of CEP192 centrosomal signal intensity in multiple RPE1 and RPE1-derived (i.e. Con and Δ11) cell lines ($n = 2$, 10 cells/biological replicate). RPE1 indicates the parental p53$^{+/+}$ line. Con-3 is an additional RPE1$^{p53-/-}$ control cell line generated previously within the lab. Exact $p$ values (L-R): 0.6731, 0.0042, 0.9486, 0.8134, 0.841. (E) Box plot of TPX2 spindle pole signal intensity in RPE1$^{p53-/-}$ cells, with representative images shown in (C) ($n = 2$, 10 cells/biological replicate). Exact $p$ values (L-R): 0.2836, 0.982. (F) Proximity ligation assay (PLA) between Aurora-A and CEP192 specific antibodies in control and Δ11 RPE1$^{p53-/-}$ cells. PLA signal is green in merged images with DNA stained with DAPI (blue). (G) Scatter plot of the Aurora-A:CEP192 PLA signal intensity in mitotic cells from (F) ($n = 2$, 15 cells/biological replicate). Exact $p$ values (L-R): <0.0001, <0.0001. (H) Proximity ligation assay (PLA) between Aurora-A and CEP192 specific antibodies in control and Δ11 U251 cells. PLA signal is green in merged images with DNA stained with DAPI (blue). (I) Scatter plot of the Aurora-A:CEP192 PLA signal intensity in mitotic cells from (H) ($n = 2$, ≥10 cells/biological replicate). Exact $p$ values (L-R): <0.0001, <0.0001, <0.0001. (J) Table summarising cell cycle profiles, of propidium iodide-stained cells, obtained by flow cytometry ($n = 3$ biological replicates, mean ± S.D). Representative histograms can be found in figure Appendix Fig. S6F–H. Data information In (G) and (I), filled and hollow circles indicate PLA signal either being focused at the spindle pole or scattered throughout the cytoplasm, respectively. Box plots in (D–E) indicate the median and interquartile ranges (25th–75th percentile) with coloured whiskers representing 5th–95th percentile ranges. Grey bars in (G) and (I) indicate mean ± S.D. Grey shading in (E, G, J) denotes independently completed biological replicates. $p$ values are denoted as follows: ****$p < 0.0001$, **$p < 0.01$, n.s not significant (Mann–Whitney test). Scale bars in (C, F, H) represent 10 μm. Source data are available online for this figure.

20 min of NEBD, the mild genotoxic effect of SiR-Hoechst increases this to around 30 min (Rajendraprasad et al, 2023). RCon-1 and 2 reached anaphase after 36 and 30 min, respectively. These slightly longer timings may be due to the selection or divergence of our p53-deficient clones. Nonetheless, the two RΔ11 clones took nearly 50% longer (i.e. 48 and 52 min on average) to initiate anaphase, both showing a significant overall increase in mitotic duration.

Consistent with impaired spindle function, chromosome segregation errors were more frequent in RΔ11 cells, manifesting in a twofold increase of lagging chromatids during anaphase (Fig. 7C,D). Although mitotic spindle morphology was not majorly perturbed, RΔ11 spindles appeared shorter. While the latter could result from low Aurora-A activity (Bird and Hyman, 2008), the slower mitotic progression in RΔ11 cells could also increase the ratio of prometaphase vs metaphase cells, thus skewing spindle length distribution. To differentiate between these possibilities, spindle length was measured in a metaphase population obtained by treatment with the proteasome inhibitor, MG-132 (Fig. 7E,F). RΔ11 mitotic spindles were abnormally short under these conditions confirming the requirement for the CEP192:Aurora-A complex for normal spindle length. Indeed, a defect in spindle microtubule growth and/or stability is likely since the diameter of STLC-induced monopolar spindles was reduced in RΔ11-1 cells (Fig. EV4B,C). The occasional monopolar-like spindle is not caused by centriole loss because RΔ11 cells contain normal centriole numbers, demonstrating that deletion of exon 11 does not disrupt centrosome duplication (Fig. EV4D,E). Additionally, RΔ11 mitotic spindles exhibited a marked orientation defect with a twofold increase in cells unable to align their spindle parallel to the substratum (Fig. 7E,G). Time-lapse microscopy confirmed that the frequency of anaphases occurring at an angle was elevated in RΔ11 cells (Figs. 7B and EV4F).

The Aurora-A:TPX2 complex controls spindle orientation by limiting spindle pole association of Nuclear mitotic apparatus protein 1 (NuMA), hence enabling its cortical targeting (Gallini et al, 2016; Kotak et al, 2016; Polverino et al, 2021). Mitotic RΔ11 cells were co-stained with antibodies against NuMA and Eg5/Kinesin-5, a microtubule motor protein previously shown to depend on CEP192:Aurora-A for its spindle localisation (Joukov et al, 2014). While we observed a decrease in spindle levels of Eg5 upon Aurora-A inhibition, Eg5 intensity remained similar between control and RΔ11 spindles (Figs. 7H,I and EV4G). By contrast, NuMA levels at spindle poles appeared equally high in RΔ11 and Aurora-A inhibitor-treated cells (Figs. 7H,J and EV4G). An increase was also seen in UΔ11 cells (Fig. EV4H,I), pointing to a major role for Aurora-A:CEP192 complexes in generating a highly active Aurora-A:TPX2 pool that removes NuMA from spindle poles.

Collectively these results argue that binding of Aurora-A to CEP192, which occurs predominantly at centrosomes and results in autophosphorylation, is vital for normal mitotic spindle function. In particular, CEP192:Aurora-A mediates effective Aurora-A:TPX2 complex formation, hence promoting normal spindle length and orientation, whilst increasing the fidelity of chromosome segregation.

## CEP192:Aurora-A and TPX2:Aurora-A are both required for spindle recruitment of TACC3

Phosphorylation of TACC3 on S558 by Aurora-A is required for its association with clathrin heavy-chain and subsequent recruitment to the mitotic spindle (Cheeseman et al, 2011). Indeed, Aurora-A inhibition abrogated spindle recruitment of TACC3, whereas TACC3 levels decreased by ~35% at RΔ11 spindle poles indicative of a reduction rather than loss of kinase activity (Fig. EV5A,B). This fall in spindle-associated TACC3 can contribute to the short spindle phenotype of RΔ11 cells (Fig. 7F) (Gergely et al, 2003). In line with less TACC3 and Aurora-A localising to RΔ11 mitotic spindles, the intensity of PLA foci corresponding to Aurora-A:TACC3 complexes was much reduced in RΔ11 cells with remaining foci being scattered throughout the cytoplasm (Fig. EV5C,D). Remarkably, TPX2 depletion fully prevented both spindle recruitment of TACC3 and PLA foci formation between TACC3 and Aurora-A antibodies, uncovering an essential role for TPX2 in Aurora-A-driven spindle localisation of TACC3 (Fig. 7K–M). We also investigated Aurora-A-dependent phosphorylation of LATS2 in Aurora-A inhibitor-treated and RΔ11-1 cells; pS83 LATS2 levels were reduced by 50% in the former but unchanged in the latter (Fig. EV5E,F). These findings collectively suggest that Aurora-A kinase, even without T-loop-phosphorylation, can phosphorylate certain substrates like Lats2 and TACC3, most likely in a complex with its co-activator TPX2.

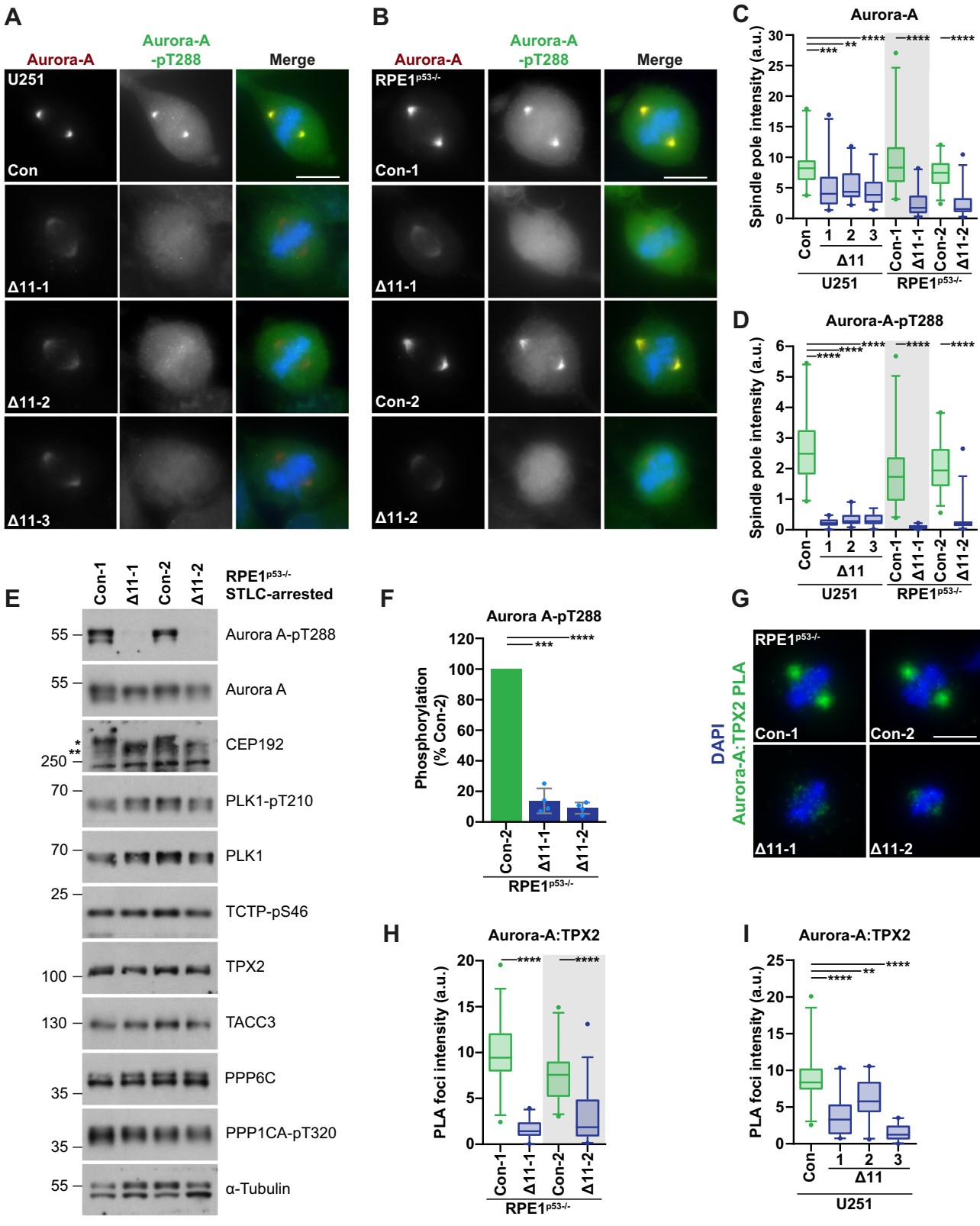

Figure 6. Binding of Aurora-A to CEP192 is essential for its autophosphorylation and spindle recruitment.

(A, B) Immunofluorescence images of methanol fixed control and Δ11 (A) U251 and (B) RPE1$^{p53-/-}$ cells. Antibodies against Aurora-A and Aurora-A-pT288 are red and green in merged images, respectively, with DNA stained with DAPI (blue). (C, D) Box plots of (C) Aurora-A and (D) Aurora-A-pT288 spindle pole signal intensity in U251 and RPE1$^{p53-/-}$ cells, with representative images shown in (A, B) ($n = 2$, ≥10 cells/ biological replicate). Exact $p$ values from (C) (L-R): 0.0005, 0.001, <0.0001, <0.0001, <0.0001. Exact $p$ values from (D) (L-R): <0.0001, <0.0001, <0.0001, <0.0001, <0.0001. (E) Western blot of control and Δ11 RPE1$^{p53-/-}$ cells synchronised in mitosis with 10 μM STLC (20 h). Antibodies against Aurora-A binding partners and several mitotic regulators are shown. α-tubulin serves as a loading control, while the mitotic phosphorylation PPP1CA-pT320 demonstrates equivalent mitotic arrest between samples. A band-shift between CEP192-WT and Δ11 proteins can be observed, these forms are marked with * and **, respectively. (F) Densitometric quantification of Aurora-A-pT288 signal from (E). Grey bars indicate mean ± S.D ($n = 4$ biological replicates). Exact $p$ values (L-R): 0.0002, <0.0001. (G) Proximity ligation assay (PLA) between Aurora-A and TPX2 specific antibodies in control and Δ11 RPE1$^{p53-/-}$ cells. PLA signal is green in merged images with DNA stained with DAPI (blue). (H) Box plot of the Aurora-A:TPX2 PLA signal intensity in mitotic RPE1$^{p53-/-}$ cells with representative images shown in (G) ($n = 2$, ≥15 cells/biological replicate). Exact $p$ values (L-R): <0.0001, <0.0001. (I) Box plot of the Aurora-A:TPX2 PLA signal intensity in mitotic U251 cells, with representative images shown in Appendix Fig. S7E ($n = 2$, ≥10 cells/ biological replicate). Exact $p$ values (L-R): <0.0001, 0.0048, <0.0001. Data information Box plots in (C, D and H, I) indicate the median and interquartile ranges (25th–75th percentile) with coloured whiskers representing 5th–95th percentile ranges. Grey shading in (C, D, H) denotes independently completed biological replicates. $p$ values are denoted as follows: ****$p < 0.0001$, ***$p < 0.001$, **$p < 0.01$ (C, D and H, I Mann–Whitney test, F Welch's $t$-test). Scale bars in (A, B, G) represent 10 μm. Source data are available online for this figure.

Aurora-A is also considered a key regulator of centrosome maturation (Joukov et al, 2014), and therefore, we expected mitotic RΔ11 centrosomes lacking Aurora-A to be depleted of PCM proteins such as γ-tubulin or pericentrin (PCNT). As with CEP192 levels, centrosomal γ-tubulin intensity in RCon-2 was elevated and confirmed to be an outlier when compared to further control cell lines (Fig. EV5G). Nevertheless, while Aurora-A inhibitors caused a significant drop in centrosomal γ-tubulin, γ-tubulin and PCNT levels were much less impacted by *CEP192*(exon 11) deletion (Fig. EV5A,E,H,I). Why might this be the case? The BORA:Aurora-A complex has been shown to activate PLK1 in G2, thus promoting both mitotic entry and centrosome maturation (Seki et al, 2008a; Seki et al, 2008b). It is therefore feasible that in RΔ11 (as in control cells), centrosomal PLK1, previously activated by BORA:Aurora-A, binds and phosphorylates CEP192 leading to recruitment of additional γ-tubulin ring complexes and hence PCM expansion (Alvarez-Rodrigo et al, 2019; Joukov et al, 2014; Meng et al, 2015; Ohta et al, 2021). The PLK1-binding domain is retained in CEP192(Δ11) and thus should interact with PLK1 just like wild-type CEP192 (Fig. 5A). Indeed, PLK1 levels at mitotic centrosomes were comparable between control and RΔ11 cells (Appendix Fig. S8A,B). Whereas levels of both Aurora-A-dependent T-loop phosphorylation of PLK1 (pT210) and PLK1-dependent phosphorylation of TCTP (pS46) (Cucchi et al, 2010), decreased in response to treatment with Aurora-A and PLK1-inhibitors (Appendix Fig. S8C–E), there was no change in these or in total PLK1 levels, in mitotic lysates of RΔ11 cells, indicating that mitotic PLK1 function is unperturbed (Fig. 6E; Appendix S8F,G).

## The interaction between Aurora-A and CEP192 is ancient and conserved

INCENP is the ancestral Aurora-binding protein, conserved in single-celled eukaryotes such as the Sli15 protein in *Saccharomyces cerevisiae*. INCENP binds to both Aurora-B and Aurora-C kinases, wrapping around the N-lobe of the kinase and also burying a large surface area (Elkins et al, 2012; Sessa et al, 2005). The interaction stimulates both Aurora-B/C activity and the phosphorylation of INCENP itself (Abdul Azeez et al, 2019). Like CEP192, INCENP interacts with part of the region equivalent to the TACC3-binding site of Aurora-A. While W845 in human INCENP is equivalent to

F508 in CEP192 and F525 in TACC3, INCENP does not have a helix that occupies the groove above the glycine-rich loop like CEP192 or TACC3 (Fig. 8A). As INCENP wraps around the N-lobe, the region between R847 and H861 is helical in comparison to the equivalent stretch in CEP192 that is disordered. In the F-pocket of the kinases, F869 in INCENP is equivalent to CEP192 F490. However, INCENP cannot be the ancestor of CEP192 or TPX2 because the main chain of INCENP runs in the opposite direction around the N-lobe of Aurora-B/C, and is perhaps an example of convergent evolution (Fig. 8A) (Elkins et al, 2012).

To explore the evolution of Aurora-A binding in CEP192, we searched for relatives of the human sequence in other eukaryotes. The key motifs are highly conserved in distantly related organisms such as corals and anemones (Fig. 8B, larger alignment Appendix Fig. S9). In the region of CEP192 that interacts with the F-pocket of Aurora-A, similar to TPX2, only the residue equivalent to F490 in human CEP192 is highly conserved, and L484 that packs close to the αC-β4 region in Aurora-A is somewhat conserved (Fig. 8B). In the other region of CEP192, that competes with TACC3, the aromatic F508 in human CEP192 is highly conserved (Fig. 8C). Additionally, the residues at positions $i$, $i+4$, $i+7$ and $i+11$ along the long helix in CEP192, between residues 510–529 (I518, E522, F525 and H529 in human CEP192), that interact with Aurora-A above the glycine-rich loop show strong conservation (Fig. 8C). From this analysis, we conclude that the interaction of CEP192 with Aurora-A via two sites is an ancient structural mechanism dating back at least 500 MY.

To test this concept and to gain further insights into the conserved nature of this interaction, AlphaFold2 multimer was used to model the complex between *Acropora millepora* (stony coral) orthologues of CEP192 and Aurora-A kinase domain (Fig. 8D). The sequence identity between the Aurora-A orthologues is 74%, comparing human to *Acropora*. By contrast, the sequence identity between CEP192 human region 468–533 and *Acropora* 781–838 is 26%. The coral CEP192 is predicted to wrap around the N-lobe of Aurora-A, with a conserved Phe overlaying with F490 in human CEP192 and pointing into the F-pocket, just as in the human protein complex. However, in the predicted complex of coral proteins, there is no β-hairpin in CEP192 and no interaction with the αC-β4 region of Aurora-A. Thus although the Aurora-A:CEP192 interaction is highly conserved, there are likely to be differences in their biochemical properties.

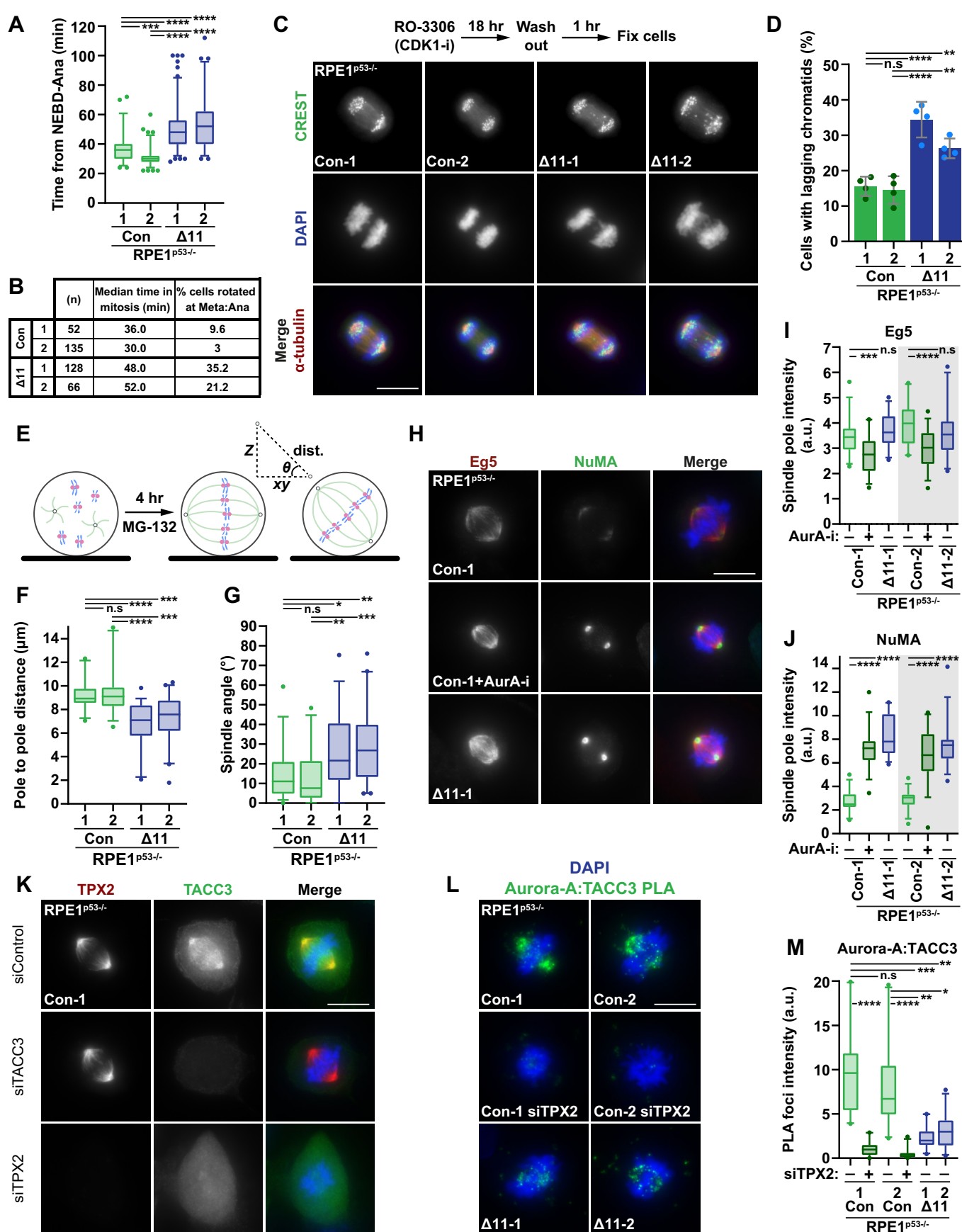

◄ **Figure 7. Loss of the CEP192:Aurora-A interaction leads to mitotic spindle defects and chromosome segregation errors.**

(A) Box plot of the time from NEBD-Ana (nuclear envelope breakdown to anaphase onset) in SiR-Hoechst-labelled control and Δ11 RPE1$^{p53-/-}$ cells from (>50 mitotic events per condition, collected during one imaging session). Exact $p$ values (bottom-top): <0.0001, <0.0001, 0.0004, <0.0001, <0.0001. (B) Table indicating the number of mitotic events imaged ($n$), the median time in mitosis and the % of cells rotated at the metaphase to anaphase transition for each condition in (A). (C) Immunofluorescence images of PTEMF-fixed control and Δ11 RPE1$^{p53-/-}$ cells in anaphase. Antibodies against α-tubulin and CREST are red and green in merged images, respectively, with DNA stained with DAPI (blue). Cells were treated with 6 μM RO-3306 (18 h) to synchronise in late-G2 and then released into mitosis. Cells were then incubated for 60 min to enrich for anaphase cells and fixed. (D) Bar graph indicating the percentage of control and Δ11 RPE1$^{p53-/-}$ cells, from (C), containing lagging chromatids in anaphase. Grey bars indicate mean ± S.D ($n = 4$ biological replicates). Exact $p$ values (bottom-top): <0.0001, 0.0038, 0.9973, <0.0001, 0.0078. (E) Schematic depicting metaphase arrest using MG-132 proteasome inhibition and how pole-to-pole distance and spindle angle were subsequently calculated. (F, G) Box plots of the (F) pole-to-pole distance and (G) spindle angle in RPE1$^{p53-/-}$ cells treated with a proteasome inhibitor, MG-132 (4 h), to enrich for metaphase cells ($n = 2$, ≥15 cells/biological replicate). Exact $p$ values from (F) (bottom-top): <0.0001, 0.0001, >0.9999, <0.0001, 0.0002. Exact $p$ values from (G) (bottom-top): 0.0024, 0.0002, >0.9999, 0.0297, 0.0036. (H) Immunofluorescence images of control-1 and Δ11-1 RPE1$^{p53-/-}$ cells treated with either DMSO control or Aurora-A inhibitor (30 min) prior to methanol fixation. Antibodies against Eg5 and NuMA are red and green in merged images, respectively, with DNA stained with DAPI (blue). Representative images of control-2 and Δ11-2 RPE1$^{p53-/-}$ cells can be found in EV4G. (I, J) Box plots of the (I) Eg5 spindle and (J) NuMA spindle pole signal intensity in RPE1$^{p53-/-}$ cells, with representative images shown in (H) and EV4G ($n = 2$, ≥15 cells/biological replicate). Exact $p$ values from (I) (L-R): 0.0004, 0.1439, <0.0001, 0.0578. Exact $p$ values from (J) (L-R): <0.0001, <0.0001, <0.0001, <0.0001. (K) Immunofluorescence images of methanol fixed control and Δ11 RPE1$^{p53-/-}$ cells treated with the indicated siRNA (48 h). Antibodies against TPX2 and TACC3 are red and green in merged images, respectively, with DNA stained with DAPI (blue). (L) Proximity ligation assay (PLA) between Aurora-A and TACC3 specific antibodies in either control or Δ11 RPE1$^{p53-/-}$ cells, treated with the indicated siRNA (48 h). PLA signal is green in merged images with DNA stained with DAPI (blue). (M) Box plot of the Aurora-A:TACC3 PLA signal intensity in mitotic RPE1$^{p53-/-}$ cells from (L) ($n = 2$, ≥10 cells/biological replicate). Exact $p$ values (bottom-top): <0.0001, <0.0001, 0.0026, 0.022, >0.9999, 0.0001, 0.0013. Data information Box plots in (A, F, G, I, J, M) indicate the median and interquartile ranges (25th–75th percentile) with coloured whiskers representing 5th–95th percentile ranges. Grey shading in (I, J) denotes independently completed biological replicates. $p$ values are denoted as follows: ****$p < 0.0001$, ***$p < 0.001$, **$p < 0.01$, *$p < 0.05$, n.s not significant (A, F, G, M Kruskal–Wallis test, D ANOVA, I, J Mann–Whitney test). Scale bars in (C, H, K, L) represent 10 μm. Source data are available online for this figure.

# Discussion

## CEP192 is a context-specific regulator of Aurora-A with a dual-site interaction mechanism

Aurora-A depends on binding partners to stimulate its activation via autophosphorylation, and so we have previously described it as an "incomplete" kinase (Burgess et al, 2015). The structural underpinnings of its low basal activity and rescue by binding partners such as TPX2 and TACC3 has been described in previous work from us and other laboratories (Bayliss et al, 2003; Dodson and Bayliss, 2012; Zorba et al, 2014). In the absence of an appropriate binding partner, Aurora-A adopts an inactive conformation, and the surface of the kinase has several hydrophobic pockets that provide binding sites for combinatorial interactions. CEP192 binds to two sites on Aurora-A with one in common with TACC3 and the other in common with TPX2. The key side chains of CEP192 involved in the TACC3 site interaction are the same as those observed in a previous structure of CEP192$_{506-527}$ fused to the Aurora-A kinase domain (Appendix Fig. S10A,B) (Park et al, 2023). However, there are several differences between our structure and the previous structure that are most likely due to the use of a separate, not fused, and longer fragment of CEP192 (aa 468–533). The long helix that packs over the glycine-rich loop contains one extra turn that positions His529 of CEP192 to interact with Gln168 of Aurora-A (Appendix Fig. S10B,C). There is an additional, short helix present in the structure, spanning residues 502-507, in which both Met504 and Asn505 of CEP192 interact with Aurora-A. The most striking difference is a second binding site with Aurora-A that was not present in the construct used in the previous structure. The contribution of the second site to the interaction in solution is supported by site-specific mutations and biophysical methods, such as NMR chemical shift perturbations and HDX-MS, that were not used in the previous study (Park et al, 2023). The

importance of these residues to the Aurora-A:CEP192 interaction in vivo is underscored by the phenotype of Δ11 cells where deleting CEP192 (aa 464–511) prevents both centrosomal accumulation and autophosphorylation of Aurora-A (Fig. 6A-D). Our structure also shows how CEP192 wraps around the N-lobe of Aurora-A to interact with the F-pocket in competition with TPX2 (Fig. 2G).

The F-pocket is a rare feature in protein kinases, but is conserved in the Aurora family, and the equivalent site forms part of the interface between Aurora-B and INCENP. It thus may contribute to the specificity of the molecular recognition of Aurora kinases by their most critical binding partners. Binding to the F-pocket had not been thought to affect Aurora-A activity but our results with CEP192 suggest otherwise. One side of the F-pocket is formed by the β4 strand, and TPX2/CEP192 both have a Phe that interacts with Y197, one residue C-terminal to the top R-spine position (R4, L196). TPX2 and CEP192 also position a hydrophobic side chain into the gap between αC and β4 at the top of the R-spine, but with different outcomes for Aurora-A activity. Whereas TPX2 binds via P13 and Aurora-A is activated, the bulkier L484 of CEP192 is incompatible with the short distance needed for an active conformation in this region, and the kinase is inhibited. The αC-β4 region of kinases is a hotspot for kinase regulation, and here we have demonstrated the bi-directional allosteric control of Aurora-A kinase by its native partners at this region of the kinase surface. The function of inhibition is unclear, but it could help to suppress the activity of the Aurora-A:CEP192 complexes we observed in the cytoplasm (Fig. 5F,H), while the high concentration of this complex at the centrosome enables the kinase to auto-phosphorylate as proposed before (Joukov et al, 2010). The two-site mechanism of CEP192 binding to Aurora-A via a (i) mildly activating, robust binding to the TACC3 pocket and (ii) an inhibitory, more dynamic interaction at the F-pocket offers a compelling hypothesis to how CEP192 can act as a context-specific inhibitor or activator of kinase activity.

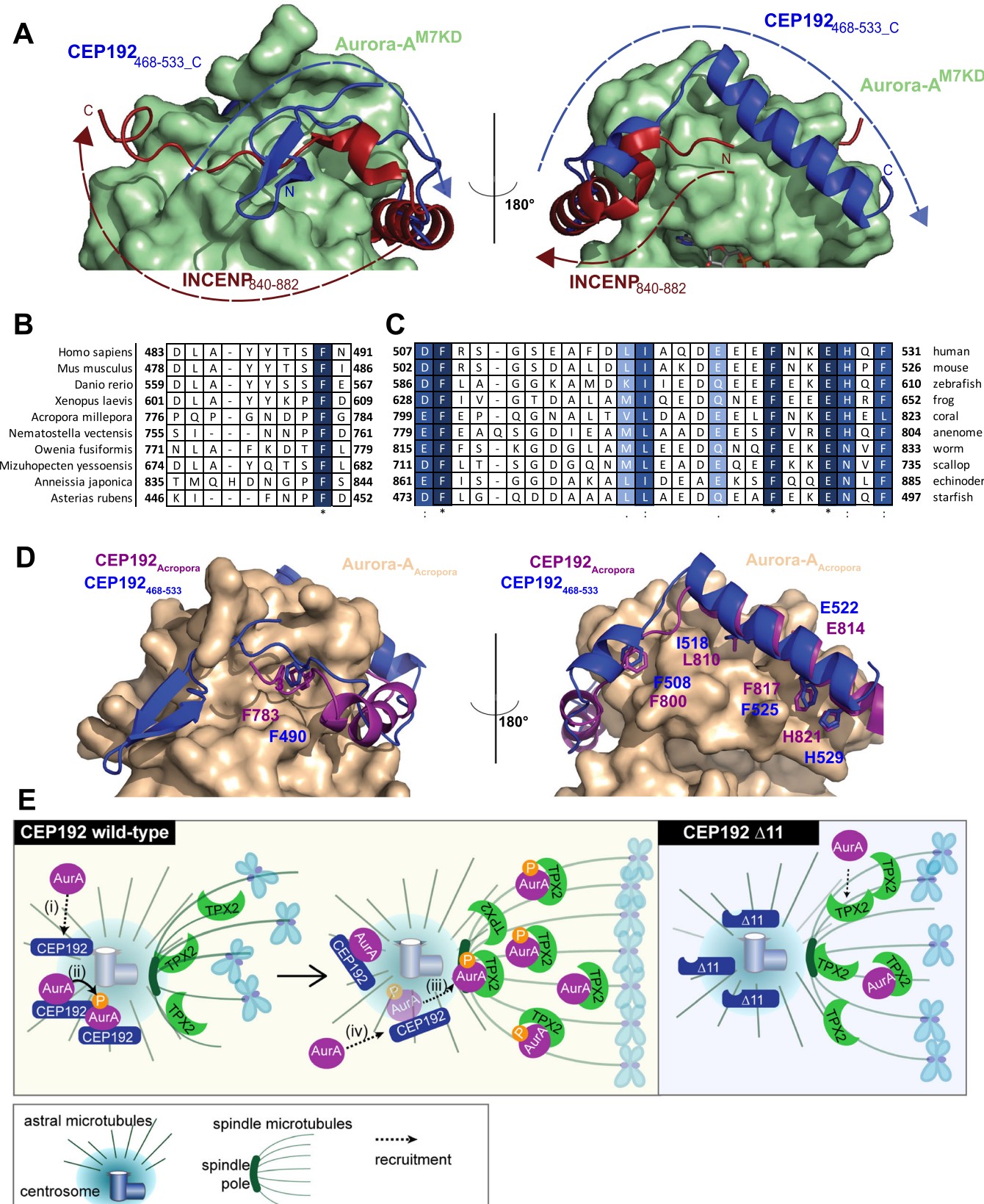

**Figure 8. The interaction between CEP192 and Aurora-A is highly conserved.**

(A) Comparison of CEP192₄₆₈₋₅₃₃ bound to Aurora-A^M7KD with human INCENP bound to Aurora-B. PDB 4AF3 was overlaid with the co-crystal structure of CEP192₄₆₈₋₅₃₃ (dark blue) bound to Aurora^M7KD (pale green). The main chain of INCENP (dark red) runs in the opposite direction around the N-lobe of the kinase domain. (B) Sequence alignment of CEP192 orthologues from divergent species identified from a BLAST search, in the region that binds the F-pocket on Aurora-A. The darker the residue, the higher degree of conservation. (C) Sequence alignment of CEP192 orthologues from divergent species identified from a BLAST search, in the region that binds the TACC3-binding site. The darker the residue, the higher degree of conservation. (D) Cartoon representation of the AlphaFold2 model of the *Acropora millepora* Aurora-A kinase domain (XP_029186576.2 115–381, wheat, Aurora-A_Acropora) and potential CEP192 orthologue (XP_044175215.1 781–838, purple, CEP192_Acropora). The human CEP192₄₆₈₋₅₃₃ is shown in dark blue for comparison. (E) Cartoon depicting the hub-and-spoke model of Aurora-A activation during mitosis. (i) In wild-type cells, Aurora-A is recruited to the centrosome through a bi-modal interaction with CEP192, where (ii) a high local-concentration of Aurora-A promotes autophosphorylation of the Aurora-A activation loop. (iii) Aurora-A phosphorylation increases the affinity of the Aurora-A:TPX2 interaction, facilitating the recruitment of Aurora-A to spindle microtubules and, therefore, proper spindle function and orientation. (iv) CEP192 can now interact with and activate further molecules of Aurora-A at the centrosome, ensuring optimal spatiotemporal control of Aurora-A activity during mitosis. In Δ11 cells, Aurora-A is neither recruited to the centrosome nor autophosphorylated. This prevents efficient formation of the Aurora-A:TPX2 complex at the spindle pole, where the reduced Aurora-A activity results in spindle abnormalities.

## Hierarchical relationship between different Aurora-A:co-activator complexes

Our results uncover a previously unrecognised hierarchy of Aurora-A:co-activator complex formation. In G2, CyclinA-CDK2-phosphorylated BORA acts a potent activator of Aurora-A and the resulting BORA:Aurora-A complex is vital for PLK1 activation and mitotic entry (Seki et al, 2008b; Tavernier et al, 2021). Concomitantly, CEP192 also generates a small centrosomal pool of T-loop-phosphorylated Aurora-A, but PLK1-driven degradation of BORA (Seki et al, 2008a) from late-G2 could increase the functional significance of this CEP192-dependent pool. Indeed, our findings suggest that during mitosis CEP192 is the main source of T-loop phosphorylated Aurora-A.

We postulate a model (Fig. 8E) where by concentrating Aurora-A at centrosomes, CEP192 generates an autophosphorylated pool of the kinase primed for complexing with TPX2. While we favour this model, it is also possible that TPX2 requires centrosomal Aurora-A, either phosphorylated or non-phosphorylated, for effective complex formation, and this pool is absent in Δ11 cells. In either case, TPX2 seems unable to stimulate autophosphorylation of the kinase. Once in a complex, TPX2 localises Aurora-A to spindle microtubules, whilst also shielding pT288 from dephosphorylation by the abundant protein phosphatase 1 (PP1) (Bayliss et al, 2003; Eyers et al, 2003). Such a scenario is consistent with our results as well as the literature. Upon loss of CEP192:Aurora-A binding, Aurora-A is no longer recruited to centrosomes thus failing to undergo concentration-dependent T-loop phosphorylation, which leads to reduced complex formation with TPX2. In TPX2-depleted cells, pT288 Aurora-A levels are very low because centrosome-generated pT288 Aurora-A is neither recruited to the spindle nor does it benefit from TPX2-dependent protection from PP1. Therefore, while CEP192 generates pT288 Aurora-A, its subsequent binding to TPX2 is vital for maintaining T-loop-phosphorylation and amplifying kinase activity.

In contrast to PP1, protein phosphatase-6 (PP6) specifically targets TPX2-bound Aurora-A for dephosphorylation (Zeng et al, 2010). PP6-driven loss of T-loop phosphorylation could lower binding affinity between TPX2 and Aurora-A and ultimately cause release of the kinase into the cytoplasm, making it available for centrosomal recruitment by CEP192.

Nevertheless, our work suggests a role for pT288 in driving the association of Aurora-A with TPX2, and thus spindle recruitment of the kinase. Intriguingly, once the TPX2:Aurora-A complex forms, T-loop phosphorylation seems dispensable for Aurora-A-dependent spindle recruitment of TACC3. Indeed, despite the loss

of pT288 in RΔ11 cells, TACC3 remained detectable on spindles along with PLA foci corresponding to Aurora-A:TACC3 complexes. In contrast, TPX2-depleted cells were devoid of spindle-bound TACC3. Therefore, TPX2 recruits TACC3 to the spindle, a process likely to involve TACC3 phosphorylation by Aurora-A:TPX2. While TACC3 itself is an Aurora-A co-activator (Burgess et al, 2018; Burgess et al, 2015), its interaction with Aurora-A in cells seems TPX2-dependent, suggesting that TACC3 might bind Aurora-A:TPX2. Such a mechanism is consistent with our previous work, where we showed that the Aurora-A activating domain of TPX2 (aa 1–43) facilitates the binding of Aurora-A to the substrate region of TACC3 (aa 555–563), and the three proteins form a stable complex (Burgess et al, 2015).

Due to the progressive dephosphorylation of Aurora-A:TPX2 by PP6, T-loop phosphorylated Aurora-A:TPX2 complexes, which could represent the most active kinase pool, are likely to be confined to the spindle pole region, the precise location of another important substrate, NuMA (Compton and Cleveland, 1993; Gallini et al, 2016; Kallajoki et al, 1993). While some phenotypes (i.e. monopolar spindle diameter in Fig. EV4B or LATS2 phosphorylation in Fig. EV5F) were milder upon *CEP192*(exon 11) deletion than Aurora-A inhibition, the increase in spindle pole-associated NuMA levels were identical under these conditions (Figs. 7H,J and EV4G–I). Hence, NuMA is a key target of the CEP192-dependent, T-loop-phosphorylated Aurora-A:TPX2 complexes. Perhaps NuMA is a poorer Aurora-A substrate than TACC3, which can act as an allosteric activator, and as a result, it may require maximal kinase activity only provided by CEP192-dependent autophosphorylation.

Altogether, our data suggests that CEP192 at the centrosome functions upstream of TPX2 in terms of generating T-loop phosphorylated Aurora-A, hence controlling levels, activity and localisation of the kinase. Downstream of CEP192, TPX2 and Aurora-A are both essential for spindle recruitment of TACC3 and therefore for the formation of the conserved TACC3:CHC:ch-TOG complex (Booth et al, 2011).

## Spatiotemporal control of Aurora-A activity by CEP192 is important for effective spindle assembly and genome stability

The newly discovered hierarchy of Aurora-A:co-activator complex formation provides tight spatiotemporal control of kinase activity during mitosis. T-loop-phosphorylated Aurora-A is produced almost exclusively at centrosomes, while the high affinity of TPX2 for this kinase pool ensures that active Aurora-A:TPX2 complexes are present only on the

spindle apparatus. The mitotic centrosome, therefore, generates peak Aurora-A activity in its vicinity. Nevertheless, the Aurora-A:CEP192 partnership is likely to be equally important in cells that lack centrosomes. Indeed, CEP192 is vital for spindle assembly both in mouse oocytes that naturally lack centrioles and in cells depleted of centrioles by artificial means (Chinen et al, 2021; Lee et al, 2018; Wang et al, 2017). In these cases, CEP192 forms an integral part of the PCM matrix at the mitotic spindle poles, where it should still be able to concentrate and activate Aurora-A (Chinen et al, 2021; Wang et al, 2017).

Previous studies demonstrated that PP6 activity is inhibited by polo-like kinase 1 (PLK1) (Kettenbach et al, 2018). Since Aurora-A activates PLK1 (Macurek et al, 2008; Seki et al, 2008b), and both Aurora-A and PLK1 interact with CEP192, it is tempting to speculate that high PLK1 activity in the vicinity of CEP192 could prevent PP6 from targeting pT288 for dephosphorylation. Whether the centrosomal pool of PLK1 concerned is CEP192-associated or not, such a mechanism would facilitate the build-up of a high concentration of autophosphorylated Aurora-A at centrosomes.

The hierarchical association of Aurora-A with co-activators, combined with phosphatase activities, serves to generate a gradient of kinase activity with peak activity at spindle pole and centrosomes. In PP6 knockout cells, where Aurora-A:TPX2 complexes do not get dephosphorylated and recycled, overactive Aurora-A causes severe chromosome segregation defects (Hammond et al, 2013; Sobajima et al, 2023). We find that weak kinase activity resulting from loss of the Aurora-A:CEP192 interaction also impacts genomic stability, manifesting in chromosome segregation errors in RΔ11 and polyploidisation of UΔ11 cells. Because polyploidy is not a feature of RΔ11 cells, it is tempting to speculate that cancer-related genetic changes render cytokinesis in U251 cells susceptible to disruptions of Aurora-A activity, thereby resulting in polyploidisation over cellular generations. The fine-tuning of Aurora-A activity is thus crucial to prevent genome instability in proliferating cells. This mechanism is based on the interaction of partners *via* different combinations of pockets on the surface of Aurora-A. Most of these pockets are also used by INCENP in the regulation of Aurora-B/C, an interaction preserved from the ancestral Aurora kinase found in single-celled eukaryotes. Thus, the gene duplication event that generated a second Aurora kinase provided an opening for evolutionary processes to shape a new set of proteins which not only complete the kinase domain of Aurora-A but also exert spatiotemporal control over its kinase activity, processes vital for normal mitotic spindle function.

# Methods

### Reagents and tools table

| Reagent/Resource | Reference or source | Identifier or catalogue number |
| --- | --- | --- |
| **Experimental models** | | |
| B834 competent *E.coli* cells | Novagen | |
| DH5α competent *E.coli* cells | NEB | |
| BL21(DE3)RIL competent *E.coli* cells | Novagen | |
| HEK293T (*Homo sapiens*) | ATCC | RRID:CVCL_0063 |

| Reagent/Resource | Reference or source | Identifier or catalogue number |
| --- | --- | --- |
| HeLa (*Homo sapiens*) | ATCC | RRID:CVCL_0058 |
| hTERT-RPE1 (*Homo sapiens*) | ATCC | RRID:CVCL_4388 |
| hTERT-RPE1^p53-/- Con-1 and -2 (*Homo sapiens*) | This study. | |
| hTERT-RPE1^p53-/- Δ11-1 and -2 (*Homo sapiens*) | This study. | |
| U251 (*Homo sapiens*) | ATCC | RRID:CVCL_0021 |
| U251 Δ11-1, -2 and -3 (*Homo sapiens*) | This study. | |
| U2OS (*Homo sapiens*) | ATCC | RRID:CVCL_0042 |
| **Recombinant DNA** | | |
| pet28a Inhibitory monobody 2 (with TEV cleavable His-SUMO tag) | Genscript Zorba et al, (2019) | |
| pgex6P1 stop (GST only) | This study | |
| pgex6P1 human CEP192 1–96 | This study. | |
| pgex6P1 human CEP192 96–266 | This study. | |
| pgex6P1 human CEP192 266–442 | This study. | |
| pgex6P1 human CEP192 405–649 | This study. | |
| pgex6P1 human CEP192 686–1000 | This study. | |
| petSUMO human CEP192 442–533 | This study. | |
| petSUMO human CEP192 468–533 | This study. | |
| pCDF Lambda phosphatase | This study. | |
| petSUMO human CEP192 468–533 Y487A/F490A | This study. | |
| petSUMO human CEP192 468–533 F490D/F508D /I518D | This study. | |
| petSUMO human CEP192 468–533 F508D/I518D/F525D | This study. | |
| petSUMO human CEP192 468–533 L484D | This study. | |
| pet30TEV human Aurora-A 122–403 C290A/C393A (pet30TEV vector has TEV cleavable 6xHistag) | Burgess et al, (2015) | |
| pet30TEV human Aurora-A 122–403 D274N/C290A/ C393A | This study | |

| Reagent/Resource | Reference or source | Identifier or catalogue number |
|---|---|---|
| pet30TEV human Aurora-A 122–403 C290A/C393A/ D274N /N332 A/ Q335A/T347A/ D350A | This study. | |
| pet30TEV human Aurora-A 1–403 | This study. | |
| pet30TEV human Aurora-A 122–403 F165D | This study. | |
| pet30TEV human Aurora-A 122–403 R205A | This study. | |
| pet30TEV human Aurora-A 122–403 R151A | This study. | |
| pet30TEV human TPX2 1–43 | This study. | |
| pet30TEV human Aurora-A 122–403 R205A F165D | This study. | |
| pSpCas9-2A-GFP-CEP192 Exon11_5'_Guide 1 | This study. gRNA sequence in Appendix Table S4. | Original plasmid from addgene Cat# 48138. |
| pSpCas9-2A-GFP-CEP192 Exon11_3'_Guide 1 | This study. gRNA sequence in Appendix Table S4. | Original plasmid from addgene Cat# 48138. |
| pSpCas9-2A-GFP-CEP192 Exon11_3'_Guide 2 | This study. gRNA sequence in Appendix Table S4. | Original plasmid from addgene Cat# 48138. |
| p53 CRISPR/Cas9 KO Plasmid | SantaCruz Biotechnology | Cat #sc-4146469 |
| **Antibodies** | | |
| GST polyclonal (Goat) | Cytiva | Cat #27-4577-01, RRID: AB_771432 |
| 6xHis monoclonal (Mouse) | Takara Bio | Cat #631212, RRID: AB_2721905 |
| Aurora-A monoclonal (Rabbit, 1:500 dilution) | CST | Cat #4718S, RRID: AB_2061482 |
| Aurora-A monoclonal (Mouse, 1:500, PLA 1:1500) | Merck | Cat #A1231, RRID: AB_796191 |
| Aurora-A-pT288 monoclonal (Rabbit, 1:250) | CST | Cat #3079S, RRID: AB_2061481 |
| Aurora-A-pT288, Aurora-B-pT232, Aurora-C-pT198 monoclonal (Rabbit) | CST | Cat #2914S, RRID: AB_2061631 |
| Bora monoclonal (Rabbit, 1:500) | CST | Cat #12109S, RRID: AB_2797821 |
| CDK1-pY15 monoclonal (Rabbit, 1:500) | CST | Cat #4539S, RRID: AB_560953 |
| CEP192 polyclonal (Rabbit, 1:1000, PLA 1:1500) | Bethyl Laboratories | Cat #A302-324A, RRID: AB_1850234 |

| Reagent/Resource | Reference or source | Identifier or catalogue number |
|---|---|---|
| Eg5 monoclonal (Mouse, 1:500) | AbCam | Cat #ab51976, RRID: AB_941398 |
| LATS2-pS83 monoclonal (Mouse, 1:500) | Caltag Medsystems | Cat #CY-M1020, RRID: AB_593025 |
| NuMA monoclonal (Rabbit, 1:500) | AbCam | Cat #ab109262, RRID: AB_10863599 |
| p21 monoclonal (Mouse, 1:1000) | SantaCruz Biotechnology | Cat #sc-6246, RRID: AB_628073 |
| p53 (DO-1) monoclonal (Mouse, 1:1000) | Merck | Cat #OP43, RRID: AB_213403 |
| PCNT polyclonal (Rabbit, 1:500) | AbCam | Cat #ab4448, RRID: AB_304461 |
| PLK1 monoclonal (Mouse, 1:500) | SantaCruz Biotechnology | Cat #sc17783, RRID: AB_628157 |
| PLK1-pT210 polyclonal (Rabbit, 1:500) | CST | Cat #5472S, RRID:AB_10698594 |
| PPP1CA-pT320 monoclonal (Rabbit, 1:2000) | AbCam | Cat #ab62334, RRID: AB_956236 |
| PPP6C monoclonal (Rabbit, 1:500) | AbCam | Cat #ab131335, RRID: AB_11155732 |
| PRC1-pT481 monoclonal (Rabbit, 1:4000) | AbCam | Cat #ab62366 (EP1514Y), RRID: AB_944969 |
| TACC3 (Rabbit, 1:500, PLA 1:1500) | Gergely et al, (2000) | |
| TCTP-pS46 polyclonal (Rabbit, 1:500) | CST | Cat #5251S, RRID: AB_10547143 |
| TPX2 polyclonal (Rabbit, PLA 1:1500) | Novus Biologicals | Cat #500-179, RRID: AB_527246 |
| TPX2 monoclonal (Mouse, 1:500) | AbCam | Cat #ab32795, RRID: AB_778561 |
| α-Tubulin monoclonal (Mouse, 1:5000) | Merck | Cat #T9026, DM1A, RRID: AB_477593 |
| γ-tubulin monoclonal (Mouse, 1:500) | Merck | Cat #T6557, RRID: AB_477584 |
| Histone H3-pS10 monoclonal (Rabbit, 1:2000) | CST | Cat #53348S, RRID: AB_2799431 |
| Histone H3-pS10Alexa Flour Conjugate 488 monoclonal (Rabbit, 1:1000) | CST | Cat #3465S, RRID: AB_10695860 |
| Anti-Mouse IgG, HRP-conjugated (1:2000) | Cytiva | Cat #NXA931-1ML; RRID: AB_772209 |
| Anti-Rabbit IgG, HRP-conjugated (1:2000) | Cytiva | Cat #NA934, RRID:AB_772206 |
| Donkey anti-Mouse IgG (H + L) Highly Cross-Adsorbed Secondary Antibody, Alexa Fluor™ 555 (1:1000) | Thermo Scientific | Cat #A-31570, RRID: AB_2536180 |

| Reagent/Resource | Reference or source | Identifier or catalogue number |
|---|---|---|
| Donkey anti-Rabbit IgG (H + L) Highly Cross-Adsorbed Secondary Antibody, Alexa Fluor™ 488 (1:1000) | Thermo Scientific | Cat #A-21206, RRID: AB_2535792 |
| **Oligonucleotides and other sequence-based reagents** | | |
| PCR primers | This study | Table S4 |
| siBORA (Silencer Select) | Thermo Scientific | Cat #s36521 |
| siCEP192 (Silencer Select) | Thermo Scientific | Cat #s226819 |
| siTACC3_1 (Silencer Select) | Thermo Scientific | Cat #s20470 |
| siTACC3_2 (Silencer Select) | Thermo Scientific | Cat #s20471 |
| siTPX2 (Silencer Select) | Thermo Scientific | Cat #s22747 |
| siNegative Control No.1 (siControl) (Silencer Select) | Thermo Scientific | Cat #4390843 |
| **Chemicals, Enzymes and other reagents** | | |
| Sodium chloride | Fisher | Cat #BP358-212 |
| Tris | Melford | Cat #T600-40 |
| TCEP | Fluorochem | Cat #M02624 |
| Tween 20 | Millipore | Cat #655204 |
| Magnesium chloride hexahydrate | Sigma | Cat #63068 |
| Glycerol | Fisher | Cat #G/0650/17 |
| cOmplete™, Mini, EDTA-free Protease Inhibitor Cocktail | Roche | Cat #11836170001 |
| Hepes | Sigma | Cat #H4034 |
| DTT | Melford | Cat #D11000 |
| TBS | Thermo Scientific | Cat #J62938K7 |
| Kanamycin | Sigma | Cat #K4000 |
| Spectinomycin | Thermo Scientific | Cat #J61810.14 |
| Chloramphenicol | Sigma | Cat #C0378 |
| Imidazole | Thermo Scientific | Cat #122025000 |
| IPTG | Protein Arc | Cat #GEN-S-02122 |
| FAM-TPX2 7-43 | This study. | |
| FAM-CEP192 501–533 | This study. | |
| FAM-TACC3 522-536 | This study. | |
| ADP | Thermo Scientific Acros | Cat #164670050 |
| Kemptide | Cambridge Bioscience Ltd | Cat #4095917.0005 |
| Ampicillin | Toku-e | Cat #A042 |
| L-Glutathione reduced | Sigma | Cat #G4251 |
| Deuterium oxide | Goss Scientific | Cat #DLM-4 |
| Ammonium chloride ($^{15}$N, 99%) | Goss Scientific | Cat #NLM-467 |

| Reagent/Resource | Reference or source | Identifier or catalogue number |
|---|---|---|
| D-Glucose ($^{13}$C6, 99%) | Goss Scientific | Cat #CLM-1396 |
| Sodium hydrogen phosphate | Acros | Cat #271750025 |
| Potassium dihydrogen phosphate | Sigma | Cat #P3786 |
| PBS (10x) | Melford | Cat #P32060 |
| BME vitamin solution (x100) | Sigma-Aldrich/Merck | Cat #B6891 |
| Magnesium sulfate | Sigma | Cat #M7506 |
| Calcium chloride | Sigma | Cat #21097 |
| Iron sulfate heptahydrate | Sigma | Cat #215422 |
| Nickel chloride | Honeywell | Cat #223387 |
| EDTA | Fisher | Cat #BP120 |
| Sodium hydroxide | SLS | Cat #HE3370 |
| Alisertib | Generon | Cat #A10004-10 |
| All blue standard | Bio-Rad | Cat #1610373 |
| Blue Pre-stained Protein Standard | NEB | Cat #P7718S |
| Rink amide ProTide resin | CEM corp | Cat #R002-C |
| DMF | Fisher Scientific Ltd | Cat #10284140 |
| Piperidine | Thermo Fisher Scientific | Cat #A12442.0 F |
| DIC | Merck | Cat #38370-500 ML |
| OXYMA | Fluorochem Ltd | Cat #F043278-500G |
| 5(6)-Carboxyfluorescein | Fisher Scientific Ltd | Cat #10516081 |
| Triisopropylsilane | Fluorochem Ltd | Cat #S17975-100G |
| DODT | Merck | Cat #465178-100 ML |
| TFA | Fluorochem Ltd | Cat #F008708-1L |
| Fmoc-Ala-OH | Fluorochem Ltd | Cat #M03347-100G |
| Fmoc-Arg(Pbf)-OH | Fluorochem Ltd | Cat #M03398-100G |
| Fmoc-Asn(Trt)-OH | Fluorochem Ltd | Cat #M03352-100G |
| Fmoc-Asp(OtBu)-OH | Fluorochem Ltd | Cat #M03404-100G |
| Fmoc-Gln(Trt)-OH | Fluorochem Ltd | Cat #M03356-100G |
| Fmoc-Glu(OtBu)-OH | Fluorochem Ltd | Cat #M03409-100G |
| Fmoc-Gly-OH | Fluorochem Ltd | Cat #M03361-100G |
| Fmoc-His(Trt)-OH | Fluorochem Ltd | Cat #M03415-100G |
| Fmoc-Ile-OH | Fluorochem Ltd | Cat #M03362-100G |

| Reagent/Resource | Reference or source | Identifier or catalogue number |
| --- | --- | --- |
| Fmoc-Leu-OH | Fluorochem Ltd | Cat #M03365-100G |
| Fmoc-Lys(Boc)-OH | Fluorochem Ltd | Cat #M03419-100G |
| Fmoc-Met-OH | Fluorochem Ltd | Cat #M03368-100G |
| Fmoc-Phe-OH | Fluorochem Ltd | Cat #M03370-100G |
| Fmoc-Pro-OH | Fluorochem Ltd | Cat #M03372-100G |
| Fmoc-Ser(tBu)-OH | Fluorochem Ltd | Cat #M03382-100G |
| Fmoc-Thr(tBu)-OH | Fluorochem Ltd | Cat #M03389-100G |
| Fmoc-Trp(Boc)-OH | Fluorochem Ltd | Cat #M03376-100G |
| Fmoc-Tyr(tBu)-OH | Fluorochem Ltd | Cat #M03428-100G |
| Fmoc-6-Aminohexanoic acid | Fluorochem Ltd | Cat #F045380-25G |
| Okadaic acid | Enzo | Cat # ALX-350-003-C100 |
| S-Trityl-L-cysteine (STLC) | Bio-Techne | Cat #2191-50 |
| MLN8237 (Aurora-A-i) | MedChemExpress | Cat #HY-10971 |
| ZM447439 (Aurora-B-i) | MedChemExpress | Cat #HY-10128 |
| BI2536 (PLK1-i) | MedChemExpress | Cat #HY-50698 |
| RO-3306 (CDK1-i) | Tocris | Cat #4181 |
| MG-132 (Proteasome-i) | Merck | Cat #47490 |
| Palbociclib (CDK4/6-i) | MedChemExpress | Cat #HY-50767 |
| Nutlin-3a (MDM2-i) | SelleckChem | Cat #S8059-SEL |
| SiR-DNA kit | Spirochrome | Cat #SC007 |
| Protease inhibitor cocktail | Merck | Cat #P8340 |
| Phosphatase inhibitor cocktail 3 | Merck | Cat #P0044 |
| Oligofectamine transfection reagent | Thermo Scientific | Cat #12252011 |
| Lipofectamine RNAiMAX transfection reagent | Thermo Scientific | Cat #13778075 |
| FuGENE HD transfection reagent | Promega | Cat #E2311 |
| Opti-MEM Reduced Serum media | Thermo Scientific | Cat #31985047 |
| DMEM/F12, GlutaMAX supplement | Gibco | Cat #31331093 |
| DMEM, low glucose, pyruvate | Gibco | Cat #31885049 |

| Reagent/Resource | Reference or source | Identifier or catalogue number |
| --- | --- | --- |
| Foetal Bovine Serum (Heat inactivated) | Merck | Cat #F9665 |
| 16% Formaldehyde (methanol-free) | Thermo Scientific | Cat #28908 |
| Methanol (for fixing cells) | Merck | Cat #154903 |
| Prolong Diamond Antifade mountant | Thermo Scientific | Cat #P36961 |
| **Software** | | |
| Origin | OriginLab https://www.originlab.com/ | |
| Ccp4 | https://www.ccp4.ac.uk/ | |
| AlphaFold2 | https://github.com/sokrypton/ColabFold | |
| Coot | https://www2.mrc-lmb.cam.ac.uk/personal/pemsley/coot/ | |
| Pymol | Schrodinger https://www.pymol.org/ | |
| Molprobity | http://molprobity.biochem.duke.edu/ | |
| PDBePISA | https://www.ebi.ac.uk/pdbe/pisa/ | |
| KinCoRe | http://dunbrack.fccc.edu/kincore/home | |
| Phenix | https://phenix-online.org/ | |
| PSI-Blast | https://blast.ncbi.nlm.nih.gov/Blast.cgi | |
| MAFFT | https://mafft.cbrc.jp/alignment/software/ | |
| GUIDANCE2 | https://taux.evolseq.net/guidance/ | |
| Uniprot | https://www.uniprot.org/ | |
| NMRpipe/NMRDraw | https://doi.org/10.1007/BF00197809 | |
| CCPNMR Analysis v2.5 | https://doi.org/10.1002/prot.20449 | |
| GraphPad Prism | www.graphpad.com | |
| PLGS (v3.0.2) | www.waters.com | |
| DynamX (v3.0.0) | www.waters.com | |
| Deuteros 2.0 | https://github.com/andymlau/Deuteros_2.0 (Lau et al, 2021) | |
| Fiji ImageJ.2 | https://imagej.net/software/fiji/ | |
| **Other** | | |
| Hidex Sense plate reader | Hidex | |
| ITC200 microcalorimeter | MicroCal | |
| Akta pure protein purification system | Cytiva | |
| 5 ml HisTrap HP | Cytiva | Cat #17524802 |

| Reagent/Resource | Reference or source | Identifier or catalogue number |
| --- | --- | --- |
| Amicon Ultra Centrifuge Filters 10 kDa MWCO | Millipore | Cat #UFC901024 |
| Q5 Mutagenesis kit | NEB | Cat #E0554S |
| Quikchange SDM kit | Aligent | Cat #200519 |
| ADP-Glo Kit | Promega | Cat #V9101 |
| HiLoad Superdex 200 pg SEC column | Cytiva | Cat #28989335 |
| Superose 12 10/300 GL | Cytiva | Cat #GE17-5173-01 |
| Automated HDX robot | LEAP Technologies, Fort Lauderdale, FL, USA | |
| M-Class Acquity LC and HDX manager | Waters Ltd., Wilmslow, Manchester, UK | |
| agarose immobilised pepsin | Thermo Fisher Scientific | |
| VanGuard Pre-column Acquity UPLC BEH C18 | Waters Ltd., Wilmslow, Manchester, UK | |
| C18 column | Waters Ltd., Wilmslow, Manchester, UK | |
| Synapt G2Si mass spectrometer | Waters Ltd., Wilmslow, Manchester, UK | |
| Pur-A-lyzer mini dialysis kit | Sigma | Cat #PURD35050 |
| SnakeSkin™ Dialysis Tubing | Thermo Fisher Scientific | Cat # 68100 |
| Low volume non-binding 384-well black plate | Greiner | Cat #784900 |
| Morpheus HT screen | Molecular Dimensions | Cat #MD1-47 |
| Glutathione Sepharose FF | Sigma | Cat #GE17-5132-03 |
| Low volume non-binding 384-well white plate | Greiner | Cat #784904 |
| 5 mm NMR tube | Norell | |
| 5 mm Shigemi NMR tube | Shigemi | |
| 750 MHz Oxford Instruments Spectrometer, 5 mm Bruker TCI cryoprobe, Bruker Avance III HD console | Bruker | |
| Mosquito Crystal | SPT labtech | |
| Liberty Blue pep. synthesiser | CEM corp. | |
| Agilent 1260 infinity HPLC | Agilent | |
| Agilent 1290 Infinity II HPLC | Agilent | |
| Kinetex EVO 5 µm C18 100 Å 21.2 ×250 mm RP column | Phenomenex | |

| Reagent/Resource | Reference or source | Identifier or catalogue number |
| --- | --- | --- |
| maXis II™ Impact QToF | Bruker | |
| Alex Fluor™ 647 Antibody labelling kit | Thermo Scientific | Cat #A20186 |
| Pierce ECL Western Blotting Substrate | Thermo Scientific | Cat #32106 |
| Westar Supernova HRP detection substrate | Geneflow | Cat #K1-0068 |
| NaveniFlex Cell MR Red PLA kit | Navinci | Cat #NC.MR.100 Red |

## Methods and protocols

### Cell culture

The following cell lines were used for this study: HeLa, HEK293T, U-2 OS (ATCC), U251 MG (ECACC) and hTERT-RPE1 (kind gift by Prof Narita (CRUK Cambridge Institute, UK). All lines tested negative for mycoplasma, and STR genotyping was used to confirm the identities of HEK293T, U251 MG and hTERT-RPE1 cells. HEK293T, HeLa, U2OS and U251 cells were grown in Dulbecco's modified Eagle medium (DMEM) (Gibco, 31885) supplemented with 10% heat-inactivated foetal bovine serum (FBS) (Gibco), while hTERT-RPE1 cells were grown in DMEM:F12 Glutamax (Gibco, 31331), 10% FBS. All cells were incubated at 37 °C, 5% $CO_2$.

### Drug treatments

For western blotting analysis of mitotic lysates, cells were synchronised in mitosis with 10 µM S-Trityl-L-cysteine (STLC, Bio-Techne, 2191-50), an Eg5 inhibitor, for 20 h prior to collecting by mitotic shake-off. For measuring monopolar spindle diameter and counting centriolar foci by immunofluorescence, cells were treated with 10 µM STLC for 1 h and 4 h prior to fixation, respectively.

As noted in the figures, the following drugs were incubated with cells for 30 min at 37 °C: Aurora-B-i ZM447439 2 µM [10 mM], PLK1-I BI2536 1 µM [10 mM] and Aurora-A-i MLN8237 [5 mM] was used at 250 and 500 nM in U251 and RPE1 cells, respectively. Drugs were resuspended in DMSO to the stock concentrations indicated in square brackets. BI2536 (HY-50698), MLN8237 (HY-10971) and ZM447439 (HY-10128) were all purchased from MedChemExpress. For western blotting, a 20-min pre-treatment with proteasome inhibitor (20 µM MG-132, MerckMillipore, 47490 [20 mM]) was required prior to the addition of DMSO control or kinase inhibitors. This prevented mitotic exit as a consequence of checkpoint override following Aurora-A and Aurora-B inhibitor addition.

Additionally, CDK1-inhibitor RO-3306 was purchased from Tocris (4181 [10 mM]) and CDK4/6-inhibitor palbociclib was purchased from MedChemExpress (HY-50767 [1 mM]).

### Assessing chromosome segregation errors

$8 \times 10^4$ RPE1[p53−/−] cells were seeded in a 24-well plate, allowed to attach and then incubated with 6 µM RO-3306 (18 h) to synchronise cells in late-G2. To release cells into mitosis, wells

were washed twice with 1x PBS and three times with DMEM/F12–10% FBS medium, all with gentle swirling. Both 1x PBS and DMEM:F12, 10% FBS were pre-equilibrated to 37 °C, 5% $CO_2$ prior to washing steps. Plates were then re-incubated for 60 min to enrich for anaphase cells, prior to PTEMF-fixation (see immunofluorescence below). To determine the percentage of cells with lagging chromatids in anaphase (Fig. 7C,D), cells were stained with antibodies against CREST and α-tubulin in addition to DAPI. Anaphase cells were defined as those fixed following the initiation of chromosome separation and prior to midbody formation and, therefore, included in the analysis. Lagging chromatids were specified as observable CREST and/or DAPI staining toward the cell centre, distinct from the main body of segregating chromatin.

### Measuring pole-to-pole distance and spindle angle in metaphase cells

$2.5 \times 10^5$ RPE1$^{p53-/-}$ cells were seeded in a six-well plate and grown for 24 h. Cells were then treated with proteasome inhibitor, 20 μM MG-132 (4 h), to prevent mitotic exit and enrich for metaphase cells. To determine the pole-to-pole distance of mitotic cells and the spindle angle relative to the coverslip (Fig. 7E–G), centrosomes were stained with antibodies against PCNT. The straight-line distance between the centre of the two centrosomes in both *xy* and z was measured. The following formulas were then used:

$$pole\ to\ pole\ distance\ \mu m = \sqrt{(xy^2 + z^2)}$$

$$spindle\ angle° = \frac{180}{\pi}\left(\tan^{-1}\left(\frac{z}{xy}\right)\right)$$

### Generation of CEP192 Δ11 CRISPR clones

Guide RNAs (gRNAs) targeting *Homo sapiens CEP192 exon 11* (ENSEMBL ENST00000506447) were designed using the CRISPR design tool within the Benchling user interface. gRNA oligonucleotides were designed with overhangs containing the Bbs1 restriction site (gRNA sequences in Appendix Table S4). The oligonucleotide pairs for *CEP192* exon 11 gRNA (sense and antisense) were phosphorylated using the T4 Polynucleotide Kinase (NEB), according to the manufacturer's instructions, and annealed. The pX458 plasmid (#48138, Addgene) was digested with the BbsI restriction enzyme (NEB) and ligated together with the oligonucleotide duplexes. gRNA-Cas9 pX458 vector pairs were transfected into U251 and RPE1 cells using FuGENE HD (Promega), according to the manufacturer's instructions. In each case, CEP192 Exon 11_5′_1 was transfected alongside either CEP192 Exon 11_3′_1 or 2, in a 1:1 ratio. Additionally, RPE1 cells were co-transfected with guides targeting p53 (SantaCruz, sc-416469). Forty-eight hours after transfection, GFP-positive cells were single-sorted into 96 well plates by FACS, using BD FACSAria IIU (BD Biosciences). Single clones were expanded and screened by PCR to identify *CEP192* Δ11 mutation-containing colonies (primer sequences in Appendix Table S4). PCR products were cloned into the pJET1.2/blunt vector and sequenced to verify the intended mutation. CEP192 wild-type control clones were chosen from sorted GFP-positive cells which did not contain the required CEP192 mutation. p53 status of RPE1 clones was determined through 24 h treatment with 2.5 μM Nutlin-3a (SelleckChem, S8059), followed by western blotting.

### Cell cycle analysis by flow cytometry in U251 and RPE1 cells

U251 and hTERT-RPE1 cells were grown to ~75% confluence in a 10 cm dish, washed once gently with PBS and trypsinized. Cells were collected in 10 ml of their respective media and counted. $1 \times 10^6$ cells were added to a 15 ml falcon and washed in PBS by centrifuging for 5 min, $500 \times g$ at room temperature. The cell pellet was resuspended in 1 ml ice-cold PBS and added dropwise to 9 ml of ice-cold 70% ethanol. Cells were fixed for at least 24 h at 4 °C prior to subsequent processing. Cells were then pelleted and washed in 1x PBS, both spins for 5 min, $500 \times g$ at 4 °C. DNA content was then stained by incubating in propidium iodide buffer (20 mg/ml propidium iodide, 2 mg/ml RNAse A, 0.1% Triton X-100 (v/v) in PBS) for 15 min at 37 °C. Stained samples were then immediately analysed using a BD FACSAria Fusion cell analyser. Histogram plots were generated using FloJo V10.9.0.

### Cell lysis and western blotting

Cells were lysed on ice for 30 min in lysis buffer (50 mM tris pH 8.0, 300 mM NaCl, 0.2%(v/v) NP-40, 10%(v/v) glycerol, protease inhibitor cocktail C, phosphatase inhibitor cocktail 3 (Sigma, 1:200 dilution), 100 nM okadaic acid (Enzo, ALX-350-003-C100)). Lysates were then centrifuged at $14{,}000 \times g$, 15 min, 4 °C, the supernatants were transferred to a fresh tube, and the pellet was discarded. Protein concentrations were measured by Bradford assay using Protein Assay Dye Reagent Concentrate (Bio-Rad Laboratories, 5000006). NuPage LDS sample buffer, supplemented with NuPAGE sample reducing agent (Invitrogen) were added prior to boiling for 10 min at 90 °C. Equal amounts of protein, between 8–20 μg/lane, were loaded into pre-cast 4–12% Bis-Tris gels (Invitrogen, NP0322BOX) and proteins separated in 1x MOPS buffer (Invitrogen) and transferred to nitrocellulose membrane (0.45 μM). Nitrocellulose was then blocked in 5% milk (wt/v) in TBS, 0.1% Tween (v/v) (TBST) for 1 h. Primary and secondary antibodies were diluted in 5% milk, TBST and incubated overnight at 4 °C or for 1 h at room temperature, respectively. Species-specific secondary antibodies conjugated to horseradish peroxidase (HRP) were used (anti-Mouse, Cytiva NA931V; anti-Rabbit, Cytiva NA934; anti-Sheep, Jackson 713-035-003). Membranes were washed for 1 h in TBST, changing the buffer five times, after each antibody incubation. All western blots were revealed using ECL (Pierce, 32106) on films or, where required for weaker signals Westar Supernova (Cyanagen, XLS3-0100).

### siRNA depletion

siRNA depletion was carried out by reverse transfection using oligofectamine (Thermo Scientific, 12252011) and lipofectamine RNAiMax (Thermo Scientific, 13778075), according to the manufacturer's instructions, for U251 and RPE1 cells, respectively. For immunofluorescence, $1 \times 10^5$ cells were seeded in six-well plates and simultaneously transfected with 60 pmol siRNA duplex and 3 μl transfection reagent diluted in 200 μl Opti-MEM serum-free medium. Growth medium was added to a final volume of 2 ml. For western blotting, $1 \times 10^6$ cells were seeded in 10 cm dishes and transfected with 300 pmol siRNA duplex and 15 μl transfection reagent diluted in 1000 μl Opti-MEM. Growth medium was added to a final volume of 10 ml. Cells were incubated with siRNA for 48 h prior to fixation or harvesting. As a negative control (siControl) Silencer select negative control No.1 siRNA was transfected as above. For TACC3 depletion,

where two siRNA duplexes were used, 40 pmol of each was transfected per well.

The following siRNA duplexes were used (Ambion):
siBora (s36521): 5′-GGAUAUGGUUGAUCCUAUAtt-3′
siCEP192 (S226819): 5′-CAAACAACUUGGAAAUCGAtt-3′
siTACC3_1 (used in combination with 2) (s20470): 5′-CACCUCGACUGGGACAAAAtt-3′
siTACC3_2 (s20471): 5′-GAGCGGACCUGUAAAACUAtt-3′
siTPX2 (s22747): 5′-CGGUAGUGAUAAAAGCUCAtt-3′

### Immunofluorescence

For siRNA depletion, $1 \times 10^5$ cells were seeded in six-well plates and transfected as described above for 48 h prior to fixation. For all other experiments, $2.5 \times 10^5$ cells were seeded in six-well plates and grown for 24 h before fixation. All cells were grown on 12 mm diameter glass coverslips (VWR, 631–1577). Cells in Fig. EV4A were fixed using 4% (wt/v) paraformaldehyde (PFA) in PBS for 12 min, followed by 10 min quenching in 50 mM $NH_4Cl$ in PBS and 5 min of permeabilization with 0.2% (v/v) Triton X-100 in PBS, all at room temperature. For centriole staining in Fig. EV4E, cells were fixed in pre-chilled methanol for 10 min at −20 °C, incubated in PBS for 10 min and then 5 min in 0.5% Triton-X-100, 0.5% Tween, 0.05% SDS in PBS (v/v). To observe centromere and kinetochore staining in Fig. 7C and Appendix Fig. S8A, cells were fixed with PTEMF buffer (20 mM PIPES pH 6.8, 0.2% (v/v) Triton X-100, 10 mM EGTA, 1 mM $MgCl_2$, 4% PFA (wt/v) in PBS) for 12 min at room temperature. For all other experiments, cells were fixed in pre-chilled methanol for 5 min at −20 °C. Following fixation, coverslips were blocked in 2% BSA (wt/v), 0.1% Tween (v/v) in PBS, this and all antibody incubation steps were carried out for 1 h at room temperature, unless otherwise stated. Primary and secondary antibody (Life technologies) dilutions were made in 0.1% Tween, PBS, with secondary staining being conducted in the dark. Coverslips were gently washed with three x 2 ml 0.2% Tween, PBS between antibody incubation steps. 500 ng/ml DAPI (Sigma, D9542) was included during the secondary antibody incubation to stain DNA. α-tubulin antibody (Merck, T9026) was directly coupled to 647 fluorophore (Life technologies, A20186) and, when required, incubated with coverslips for 90 min following secondary antibody staining. Coverslips were mounted onto glass slides (SuperFrost Ultra Plus, Thermo Scientific) with ProLong Diamond Antifade Mountant (Invitrogen), then dried overnight at room temperature and stored at 4 °C.

A standard upright microscope system (BX61, Olympus) was used to image coverslips using GFP/Alexa Fluor 488, Cy3/Alexa Fluor 555, Cy5/Alexa Fluor 647 and DAPI filter sets (Chroma Technology Corp.). Images were taken and Z-projected using MetaMorph 7.5 imaging software (Molecular Devices) and a $2048 \times 2048$-pixel complementary metal oxide semiconductor camera (PrimΣ, Photometrics). Illumination was provided by an LED light source (pE300, CoolLED Illumination Systems). Z-stacks were obtained at a spacing of 0.4 µM throughout the volume of the cell.

### Live cell imaging

$6 \times 10^4$ RPE1$^{p53−/−}$ cells were seeded in a glass-bottomed 24-well plate (Greiner, 662892), allowed to adhere and then treated with CDK4/6-inhibitor 500 nM palbociclib (24 h). After 24 h, the G1 arrested cells were washed three times with 1x PBS and three times

with DMEM:F12, 10% FBS medium, all with gentle swirling. Both 1x PBS and DMEM/F12–10% FBS were pre-equilibrated to 37 °C, 5% $CO_2$ prior to washing steps. Washing should take ~10 min, after which cells were incubated at 5% $CO_2$, 37 °C for 14 hr prior to imaging. For the final 2 h of incubation, the medium was replaced with 250 nM siR-Hoechst (Spirochrome, SC007) diluted in DMEM/F12,10% FBS. This was then removed and exchanged for siR-Hoechst-free medium immediately prior to imaging.

Live cell imaging to determine the time from nuclear envelope breakdown to anaphase onset (NEBD-Ana) was performed on an Olympus ScanR widefield, high-content imaging system fitted with an IX83 inverted motorised frame with Z-drift control, sCMOS Hamamatsu Orca Fusion B Camera (6.5 µm pixel size on-chip, $2304 \times 2304$ pixels array size, 14.976 mm × 14.976 mm FOV) and Semrock DAPI/FITC/Cy3/Cy5 Quad LED filter set. An Olympus UPLXAPO 20x, NA 0.80, WD 0.6 mm, Air/Dry objective was used. The light source was Lumencor SPECTRA X Light Engine with the following LEDs: Red 640/30 231 mW (100% power, 200 ms exposure) and White (30% power, 10 ms exposure). Identical laser and exposure settings were used across all conditions. Each field of view was imaged once every 2 min, first acquiring the 640 nm channel with a Z-stack of three planes with 5 µm spacing, followed by transillumination acquiring a single Z-plane. Four fields of view were acquired, over a 3 h 20 min period, capturing ≥50 mitotic events per condition. Cells were incubated in a humidity chamber at 37 °C, 5% $CO_2$ for the duration of imaging. SCANR Acquisition software (Version 3.5.0) was used to capture images.

### Proximity ligation assay

Cells were seeded and fixed as above, then a proximity ligation assay (PLA) was carried out according to the manufacturer's instructions with the supplied buffers (Navinci, NaveniFlex Cell MR Red, NC.MR.100 Red). All incubations are for 1 h at 37 °C, unless otherwise stated. Briefly, each coverslip was blocked in 40 µl blocking buffer before incubation with primary antibodies diluted in 80 µl of primary antibody diluent. Slides were washed $3 \times 5$ min in TBST (TBS, 0.05% Tween) with gentle agitation. Navenibodies Mouse 1 and Rabbit 2 were diluted 1:40 in navenibody diluent (40 µl per coverslip) and incubated with coverslips, prior to washing as above. Buffer 1 (1:5) and enzyme 1 (1:40) were diluted in water (40 µl per coverslip) and incubated with coverslips for 30 min, prior to washing $2 \times 5$ min TBST. Buffer 2 red and enzyme 2 were diluted as above and incubated with coverslips for 90 min, followed by $1 \times 2$ min wash in TBS. Coverslips were then incubated in 1 µg/ml DAPI in PBS for 5 min at room temperature then washed $2 \times 10$ min in TBS and $1 \times 15$ min 0.1xTBS. Slides were mounted and imaged as described for immunofluorescence.

### Image analysis

Graphs were produced and statistical analysis performed in GraphPad Prism. All figures were produced using Adobe Photoshop and Illustrator (Adobe CC). Images were processed and analysed in FIJI/ImageJ as described below.

For assessing the percentage of Histone-H3-pS10 (Fig. EV4A) positive cells, the total cell number was determined using thresholding (default mode) to highlight all DAPI-stained nuclei followed by counting using the analyse particles functions (>50 pixels², circularity 0–100). This was repeated for the Histone-H3-pS10

channel and a percentage calculated. Thresholding settings were maintained between conditions of the same experiment.

The diameter of a monopolar spindle (Fig. EV4B,C) was calculated using the following formula:

$$diameter = 2\sqrt{\left(\frac{area}{\pi}\right)}$$

Here "area" refers to the area of DAPI-stained DNA, determined by thresholding (default mode) the region of interest. The same settings were used between conditions of the same experiment. The diameter calculated, therefore, refers to that of a circle with equivalent area to the DAPI signal measured previously. The area of the nuclear DAPI signal (Appendix Fig. S6I) was determined by thresholding (as above). Area measurements were multiplied by (camera pixel size)$^2$, to convert values from pixels$^2$ to μm$^2$. All fluorescence intensity and area measurements were obtained from maximum intensity z-projections. For the quantification of Aurora-A-pT288 and Aurora-A, the signal area surrounding the spindle pole was identified by thresholding and the intensity was measured. For Aurora-A-pT288, where no observable signal was present following Aurora-A inhibition or CEP192 exon 11 deletion, the corresponding LATS2-pS83 or Aurora-A signal was used as an area mask and intensity was measured. For each spindle pole measurement, this ROI was moved to a nearby area of the cell, not covering the spindle/spindle pole, and a background intensity measurement was obtained. This background was subtracted from each spindle pole measurement to give a corrected value, which was then averaged across all poles giving a single value per cell.

For CEP192, LATS2-pS83, PCNT, PLK1 and γ-tubulin intensity, a 20-pixel diameter circle was centred over each centrosome and the intensity measured. As above, a background intensity measurement was then taken and subtracted and the corrected value was averaged across both centrosomes per cell. A 35-pixel diameter circle was used for NuMA spindle pole intensities, a 55-pixel diameter circle for TACC3 and TPX2 and a 75-pixel diameter for Eg5 spindle intensities. For these three measurements, the circle was aligned to the edge of each spindle pole using a co-stained marker, prior to measuring intensity as above.

To measure the area of PLA foci, signals were selected using the default thresholding function and the intensity of all foci summed within the boundary of the cell. The same thresholding settings were used between conditions of the proximity ligation assay.

Densitometric quantification of western blots was also performed in ImageJ. Intensity was measured in a rectangular region of interest (ROI) around each band. The same size ROI was used within experiments. An adjacent region was also measured to account for background intensity.

### Cloning

A codon-optimised construct encoding human CEP192 1-1000 with an N-terminal GST tag and the inhibitory monobody with an N-terminal Hexa-His-SUMO tag (Mb2) were obtained from GenScript (Zorba et al, 2019). The optimised CEP192 was used as a template for amplification of regions of CEP192 for subcloning using restriction enzymes into a vector with a TEV protease-cleavable N-terminal Hexa-His-SUMO tag (petSUMO) or an N-terminal GST tag (pGEX-6P1). Site-directed mutagenesis was achieved using Quikchange protocols (Aligent) and Q5 mutagenesis (NEB).

### Expression and purification of recombinant proteins

Human CEP192 442–533, 468–533 or point mutants in the petSUMO vector were transformed into *Escherichia coli* B834 RIL cells. The protein was overexpressed in LB, with growth at 37 °C until the OD at 600 nm reached 0.6–0.8. Expression was then induced with 0.5 mM IPTG overnight at 20 °C. The pelleted cells were resuspended in 10 ml of ice-cold lysis buffer per litre of grow (50 mM tris pH 7.5, 500 mM NaCl, 20 mM imidazole, 10% glycerol, 1 mM TCEP, one EDTA-free protease inhibitor tablet per 50 ml of buffer). The resuspended cells were sonicated at 60% amplitude for 10 s on, 20 s off, 5 min total. The soluble lysate was collected at 30,000 × g for 45 min in a JA 17 rotor (Beckman Coulter). After filtering (0.45 μm) the solution was loaded onto a HisTrap HP (Cytiva). Any bound protein was eluted in a gradient of lysis buffer including 500 mM Imidazole. The His-SUMO tag was then cleaved overnight using TEV protease in dialysis at 4 °C into 50 mM tris pH 7.5, 250 mM NaCl, 10% glycerol, 1 mM TCEP. The following day, the cleaved material was rebound to the HisTrap HP six times to allow the His-SUMO to rebind. The flow through and one wash of 3 CV with dialysis buffer were concentrated down in 5 kDa cut-off concentrators (Amicon) and loaded onto a HiLoad 16/600 Superdex 200 column (Cytiva) equilibrated in dialysis buffer. Fractions were checked on an SDS-PAGE gel before the protein was concentrated and stored in aliquots at −80 °C.

Human Aurora-A kinase domain 122–403 C290A C393A and mutants in an N-terminal His-tagged vector (pet30TEV) was transformed into *E. coli* RIL cells alongside the pCDF vector encoding lambda phosphatase to generate dephosphorylated protein or into *E. coli* B834 cells for phosphorylated kinase domain. The protein was overexpressed in LB, with growth at 37 °C until the O.D. at 600 nm reached 0.6–0.8. Expression was then induced with 0.5 mM IPTG overnight at 20 °C. The pelleted cells were resuspended in 10 ml of ice-cold lysis buffer per litre of grow (50 mM tris pH 7.5, 250 mM NaCl, 20 mM imidazole, 10% glycerol, 5 mM magnesium chloride, one EDTA-free protease inhibitor tablet per 50 ml of buffer). The resuspended cells were sonicated at 60% amplitude for 10 s on, 20 s off, 5 min total. The soluble lysate was collected at 30,000 × g for 45 min in a JA 17 rotor (Beckman Coulter). After filtering (0.45 μm) the solution was loaded onto a 5 ml HisTrap HP (Cytiva) equilibrated in lysis buffer. Any bound protein was eluted using a gradient of lysis buffer containing 500 mM Imidazole. The His-tag was then cleaved overnight using TEV protease in dialysis at 4 °C into 50 mM tris pH 7.5, 250 mM NaCl, 10% glycerol, 5 mM magnesium chloride, 1 mM TCEP. After dialysis the cleaved protein was rebound to the HisTrap HP equilibrated in dialysis buffer. The Aurora still interacted with the HisTrap after cleavage so a gradient of 500 mM imidazole was used to elute off the tag-free protein. The tag-free Aurora was concentrated down (10 kDa cut-off concentrator, Amicon) and loaded onto a HiLoad 16/600 Superdex 200 column (Cytiva) equilibrated with 50 mM tris pH 7.5, 250 mM NaCl, 10% glycerol, 5 mM magnesium chloride, 1 mM TCEP. In the final step Aurora-A was concentrated down again and flash-frozen before storage at −80 °C.

The His-SUMO-tagged inhibitory monobody was transformed into *E. coli* B834 RIL cells for overexpression. After growth at 37 °C, until the O.D. at 600 nm reached 0.6–0.8, expression was induced overnight at 20 °C with 0.5 mM IPTG. The cells were pelleted at 4500 rpm before being resuspended in 10 ml per litre of lysis buffer

(50 mM tris pH 7.5, 10% glycerol, 500 mM NaCl, 20 mM imidazole, 0.5 mM TCEP and protease inhibitors). The cells were lysed with sonication for 10 s on, 20 s off, 5 min total at 60% amplitude. The soluble lysate was collected at 30,000 × $g$ for 45 min in a JA 17 rotor (Beckman Coulter). This was filtered and loaded onto a HisTrap FF column (Cytiva) equilibrated in a lysis buffer. Any bound protein was eluted using a gradient of lysis buffer with 500 mM imidazole. One large peak eluted which was checked on a gel and shown to contain the monobody. The His-SUMO tag was cleaved overnight in dialysis in the presence of TEV protease (50 mM tris pH 7.5, 250 mM NaCl, 0.5 mM TCEP, 10% glycerol). The following day, the cleaved material was rebound to the HisTrap six times to allow the His-SUMO to rebind. The flow through and one wash of 3 CV with dialysis buffer were concentrated down and loaded onto a HiLoad 16/600 Superdex 200 column (Cytiva) equilibrated in dialysis buffer. Protein-containing fractions were checked on a gel before being concentrated and stored in aliquots at −80 °C.

## HDX

HDX-MS experiments were carried out using an automated HDX robot (LEAP Technologies, Fort Lauderdale, FL, USA) coupled with an M-Class Acquity LC and HDX manager (Waters Ltd., Wilmslow, Manchester, UK). CEP192 442–533 and Aurora-A 122–403 (C290A C393A, unphosphorylated) were diluted to 10 µM in equilibration buffer (20 mM tris, 200 mM NaCl, 1 mM magnesium chloride, pH 7.5) prior to analysis. About 5 µl of sample was added to 95 µl of deuterated buffer (20 mM tris, 200 mM NaCl, 1 mM magnesium chloride, pD 7.5) and incubated at 4 °C for 0.5, 2, 10 or 30 min. Following the labelling reaction, samples were quenched by adding 75 µl of the labelled solution to 75 µl quench buffer (50 mM potassium phosphate, 0.05% DDM pH 2.2) giving a final quench pH ~2.5. About 50 µl of quenched sample were passed through a home-packed pepsin column using agarose immobilised pepsin (Thermo Fisher Scientific) at 40 µl min⁻¹ (20 °C) and a VanGuard Pre-column Acquity UPLC BEH C18 (1.7 µm, 2.1 mm × 5 mm, Waters Ltd., Wilmslow, Manchester, UK) for 3 min in 0.3% formic acid in water. The resulting peptic peptides were transferred to a C18 column (75 µm × 150 mm, Waters Ltd., Wilmslow, Manchester, UK) and separated by gradient elution of 0–40% MeCN (0.1% v/v formic acid) in H₂O (0.3% v/v formic acid) over 7 min at 40 µl min⁻¹. Trapping and gradient elution of peptides was performed at 0 °C. The HDX system was interfaced with a Synapt G2Si mass spectrometer (Waters Ltd., Wilmslow, Manchester, UK). HDMSE and dynamic range extension modes (Data Independent Analysis (DIA) coupled with IMS separation) were used to separate peptides prior to CID fragmentation in the transfer cell. HDX data were analyzed using PLGS (v3.0.2) and DynamX (v3.0.0) software supplied with the mass spectrometer. Restrictions for identified peptides in DynamX were as follows: minimum intensity: 10,000, minimum products per MS/MS spectrum: 3, minimum products per amino acid: 0.3, maximum sequence length: 18, maximum ppm error: 10, file threshold: 4/6. Following manual curation of the data, summary plots were generated using Deuteros 2.0 (Lau et al, 2021). These summary plots are available in the source data.

## NMR

¹⁵N- or ¹⁵N/¹³C-labelled CEP192 was produced in Rosetta *Escherichia coli* in 250 mL of minimal medium supplemented with

¹⁵NH₄Cl and D-glucose or D-¹³C-glucose, as a His-SUMO-CEP192 fusion (full details listed above).

Spectra for assignment of ¹⁵N/¹³C-labelled CEP192 sample (100 µM, 20 mM (K/H)₃PO₄, 150 mM NaCl, 0.5 mM TCEP, 2% glycerol, 5% D₂O) were recorded on a 750 MHz Oxford Instruments Spectrometer equipped with a 5 mm Bruker TCI cryoprobe and a Bruker Avance III HD console, with data acquisition achieved using Topspin. ¹H–¹⁵N HSQC spectra were recorded at 5 °C intervals between 5 and 25 °C. There was a general loss in peak intensity at higher temperatures, so assignment spectra were recorded at 10 °C. Peak positions were manually tracked to transfer assignments across temperatures. The following triple-resonance assignment spectra were recorded HNCO, HNcaCO, HNCA, HNCoCA, HNCACB and HNcocaCB (typically recorded with 25% NUS). HBHAcoNH (25% NUS) and ¹H–¹³C HSQC spectra were recorded for Hα and Hβ proton assignment. Processing of spectra used NMRPipe/NMRDraw (Delaglio et al, 1995), peak assignments and further analysis used CCPNmr Analysis v2.5 (Vranken et al, 2005). These spectra allowed the ab initio assignment of all but two backbone H–N correlations in the ¹H–¹⁵N HSQC spectrum. Secondary shifts for Cα, Cβ, CO and Hα nuclei were calculated using random coil values ($\delta_{RC}$) available via the website: (https://www1.bio.ku.dk/english/research/bms/sbinlab/randomchemicalshifts2/) (Hendus-Altenburger et al, 2019). Secondary shift $\Delta\delta$ = observed shift ($\delta$) – random coil shift ($\delta_{RC}$),

For Aurora-A titration experiments, small (~5 µL) aliquots of concentrated Aurora-A kinase domain 122–403 C290A C393A (550 µM) were added to ~40 µM ¹⁵N or ¹⁵N/¹³C-labelled solutions of CEP192 at 25 and 20 °C, respectively, and ¹H–¹⁵N HSQC spectra were recorded. The ratio of peak intensities was calculated for each backbone H–N correlation upon increasing Aurora-A:CEP192 molar ratios. Some peaks, particularly in the N-terminal half of the protein, could not be accurately tracked due to loss in intensity and/or peak overlap. Where appropriate, ¹H–¹³C-HSQC spectra were recorded before addition, and after final addition, of Aurora-A. The assignment covering CEP192 442–533 has been deposited to the BMRB under deposition 52044.

## ITC

Thermodynamic information was established using the ITC200 microcalorimeter (MicroCal, Northampton, MA). Proteins were dialysed overnight into 20 mM tris pH 7.5, 200 mM NaCl, 10% glycerol, 5 mM magnesium chloride (CEP192 constructs/mutants and Aurora-A 122–403 mutants). Experiments were carried out at 25 °C. The syringe contained 70 µl of 250 µM Aurora-A 122–403 (D274N C290A C393A or mutant). About 2 µl injections were applied every 180 s. The cell contained 205 µl of 25 µM CEP192 construct/mutant. Microcal Origin software version 7.0 was used to determine the dissociation constants ($K_d$). All measurements were repeated at least twice.

## Fluorescence anisotropy direct binding and competition

Assays were performed in 25 mM HEPES pH 7.5, 150 mM NaCl, 2 mM DTT, 5 mM MgCl₂, 0.01% Tween 20. For CEP192 and TPX2 competition assays 10 µM Aurora-A kinase domain (phosphorylated or unphosphorylated C290A C393A 122–403) was preincubated on ice with 100 nM FAM-TPX2 7-43 or 100 nM FAM-CEP192 501–533. For TACC3 competition, due to weaker binding, 25 µM Aurora-A kinase domain (phosphorylated or

unphosphorylated C290A C393A 122–403) was preincubated with 100 nM FAM-TACC3 522-536. Serial dilutions of competitor (CEP192) were performed in triplicate into a low volume 384-well black plate, before the premixed Aurora-A and tracer were added. A control row included no tracer. Plates were left incubating for 1 h at room temperature before the fluorescence anisotropy was measured using a HIDEX plate reader, with excitation at 485 nm and emission at 535 nm. For direct binding experiments, the assay was performed as above, but the Aurora-A (122–403 C290A C393A or point mutants) was diluted across the plate from high to low concentration, before the addition of 50 nM tracer (FAM-CEP192 501–533) to three replicate rows. Buffer only was added to the control row.

Fluorescence anisotropy data were processed using Microsoft Excel to calculate intensity and anisotropy using the equation listed in (Yeo et al, 2013). The $IC_{50}$ and $K_d$ values were calculated using Origin (OriginLab) where the data fitted to a sigmoid and are summarised in Appendix Table S3.

### Co-expression of CEP192 and Aurora-A for crystallisation

Human CEP192 468–533 was cloned in a vector with a TEV protease-cleavable His-SUMO tag. Human Aurora-A kinase domain (122–403, N332A Q335A T347A D350A C290A C393A D274N, Aurora-A^{M7KD}) with no tag was cloned into the pCDF vector with spectinomycin resistance. Both vectors were co-transformed into *E. coli* B834 RIL cells and a 6 litre grow was set up. Expression was induced overnight with 0.5 mM IPTG at 20 °C. Cells were pelleted and lysed in 10 ml of lysis buffer per litre of grow (250 mM NaCl, 50 mM tris pH 7.5, 10% glycerol, 5 mM magnesium chloride, 20 mM imidazole, EDTA protease inhibitor tablets). Cells were lysed by sonication (10 s on, 20 s off on ice, nine repeats, 60% amplitude). The soluble was collected at $30,000 \times g$ for 45 min in a JA 17 rotor (Beckman Coulter) before filtering and loading onto a HisTrap HP (Cytiva) equilibrated in lysis buffer. Bound protein was eluted using a gradient of lysis buffer containing 500 mM imidazole. Dialysis was set up overnight at 4 °C to remove the His-SUMO tag from the complex with TEV protease (250 mM NaCl, 50 mM tris pH 7.5, 10% glycerol, 5 mM magnesium chloride, 1 mM TCEP). The dialysed protein was rebound to a HisTrap HP column equilibrated in dialysis buffer. The cleaved protein complex eluted early in an imidazole gradient followed by the His-SUMO and uncut protein. Fractions containing the complex were pooled, concentrated and loaded onto a HiLoad 16/600 Superdex 200 column (Cytiva) equilibrated in 250 mM NaCl, 50 mM tris pH 7.5, 10% glycerol, 5 mM magnesium chloride, 1 mM TCEP. The purity of the final complex was assessed on a gel before fractions were concentrated down in a 10 kDa cut-off concentrator and flash-frozen before storage at −80 °C.

### Crystallisation and data collection

The purified inhibitory monobody was incubated with the CEP192_{468-533}:Aurora-A^{M7KD} complex with a 1.2:1 molar ratio. After 1 h on ice the complex was separated on a HiLoad 16/600 Superdex 200 column (Cytiva) equilibrated in 250 mM NaCl, 50 mM tris pH 7.5, 10% glycerol, 0.5 mM TCEP. The protein eluted as a three-way complex. The material was concentrated down to 27 mg/ml in a 30 kDa cut-off concentrator (Amicon). ADP was added to a final concentration of 5 mM and left to bind for 1 h on ice. The complex was screened against a range of commercial crystallisation matrices. Drops were laid down at 1:1, 1:1.5 and 1.5:1 ratios of complex:precipitant in MRC sitting drop plates using a Mosquito LCP crystallisation robot (STP labtech) and incubated at 18 °C. Rod-shaped crystals were produced using Morpheus condition A2 (0.06 M Divalents, Buffer system 1 pH 6.5, 30% precipitant mix 2 (40% ethylene glycol, 20% PEG 8000)) as the precipitant after 5 days. Crystals were flash-frozen in liquid nitrogen. Diffraction data were collected from a single crystal at Diamond Light Source (Oxford, UK) on beamline I04. Autoprocessed data from xia2 "3-daii" pipeline at Diamond Light Source were used for structure determination (Winter, 2010). Molecular replacement was performed in PHASER (McCoy et al, 2007) using the Aurora-A 122–403 bound to Monobody structure as a model (PDB 6CPG; (Pitsawong et al, 2018)). Clear difference density was observed for CEP192 468–533, and this was modelled in using Coot (Emsley et al, 2010). Subsequent rounds of iterative refinement were performed using Phenix (Liebschner et al, 2019) and Coot (data collection and refinement statistics can be found in Appendix Table S1). MolProbity was used to determine structure quality (Williams et al, 2018). Key contributions to the interaction between Aurora-A and CEP192 were identified by analysis of the structure using PDBePISA (Proteins, Interfaces, Structures and Assemblies, (Krissinel and Henrick, 2007)). Structures of Aurora-A^{M7KD} bound to either CEP192_{468-533} or _{506-527} were individually submitted to the KinCoRe server (Modi and Dunbrack, 2022) to determine the conformation of the kinase (Appendix Table S2).

### Pull-down of Aurora-A by GST-tagged CEP192

GST-tagged CEP192 constructs were overexpressed in LB and purified on glutathione resin using standard methods. Tagged proteins were eluted in 100 mM TRIS pH 7.5, 500 mM NaCl, 1 mM DTT, 10% Glycerol, 10 mM reduced glutathione, before dialysis in elution buffer without glutathione and storage at −80 °C. About 250 µg of GST-tagged CEP192 constructs or GST as a control were immobilised on 50 µl of glutathione resin (GE healthcare) per sample equilibrated in pull-down buffer (50 mM tris pH 7.5, 300 mM NaCl, 1 mM DTT, 5 mM magnesium chloride, 10% glycerol, 0.1% Tween 20). After 60 min of immobilisation the beads were washed three times in 900 µl of binding buffer to remove any excess protein. About 35 µg of His-tagged Aurora-A 122–403 or full-length protein (1–403) was added to samples of immobilised truncation or GST control. These were left to bind for 60 min on the roller. The beads were pelleted and washed three times in 900 µl of binding buffer. The resin was resuspended in 100 µl of SDS loading dye. After separation on an SDS-PAGE gel, these were transformed onto nitrocellulose membranes before probing with anti-GST and anti-His primary antibodies.

### Analytical size exclusion chromatography

A Superose 12 10/300 column (Cytiva) was equilibrated in 50 mM tris pH 7.5, 250 mM NaCl, 10% glycerol, 5 mM magnesium chloride, 5 mM beta-mercaptoethanol. Aurora-A 122–403 C290A C393A and CEP192 442–533 or Aurora-A 122–403 D274N C290A C393A N332A D335A T347A D350A, CEP192 468–533 and the inhibitory monobody were mixed at 1:1 molar ratio in a total volume of 500 µl, alongside samples of just each protein alone. These were run through the column one by one and the resulting traces overlaid.

### ADP-Glo

An ADP-Glo assay kit was used to test the effect of CEP192 binding on the ATPase activity of Aurora-A kinase. CEP192 468–533 WT, 442–533 WT, mutants, 501–533 peptide, TPX2 1–43 or Alisertib (Generon) were titrated into 10 nM Aurora-A 122–403 C290A C393A phosphorylated protein in a buffer of 40 mM tris pH 7.5, 150 mM NaCl, 10 mM magnesium chloride, 1 mM DTT, 0.1 mg/ml BSA, 0.01% Tween 20. Reactions were initiated by the addition of 10 µM ATP and 100 µM kemptide as substrate (Cambridge Bioscience Ltd). The reactions were incubated at room temperature for 60 min before 5 µl was transferred into 5 ul of ADP-Glo Reagent (Promega) in a 384-well low volume low binding plate (Greiner Bio-one). After a further 40 min at room temperature, 10 µl of kinase detection reagent was added to each well, and the reaction was allowed to incubate for an additional 30 min. Luminescence was measured on a HIDEX Sense plate reader (HIDEX) using a 1 s integration time. Control data was subtracted from the luminescence and three repeats were averaged before being fitted to a sigmoidal plot using Origin (OriginLab), where $IC_{50}$ and $EC_{50}$ values were calculated.

### Evolutionary biology

Orthologous sequences were identified using iterative PSI-BLAST searches on the nr database using the UniProt sequence Q8TEP8 CEP192 468–533 as a template. Diverse orthologues were identified by constraining the searches to use just a particular taxonomic group. The multiple sequence alignment of CEP192 was generated using MAFFT and using GUIDANCE2 to generate a confidence score for the alignment with 100 bootstrap repeats (Katoh et al, 2019; Sela et al, 2015).

### AlphaFold2 modelling

The Aurora-A kinase orthologue from *Acropora millepora* was identified via a BLAST search of human Aurora-A (Uniprot O14965) against the *A. millepora* genome. The CEP192 orthologue from *A. millepora* was identified via a BLAST search of human CEP192 468–533 (Uniprot Q8TEP8) against the *A. millepora* genome. AF2-multimer (Bryant et al, 2022; Liu et al, 2023; Mirdita et al, 2022) was used through the Google Colab notebook to model the complexes between the CEP192 orthologue (XP_044175215.1 781–838) and Aurora-A kinase domain orthologue (XP_029186576.2 115-381). Superposition of AlphaFold2 predictions on known structures was performed using the align command in PyMOL (The PyMOL Molecular Graphics System, Version 2.3.5 Schrödinger, LLC).

### Peptide synthesis

Peptide synthesis was performed using Liberty Blue peptide synthesiser (CEM Corporation) with microwave heating at 0.1 mmol scale on Rink Amide ProTide resin (loading 0.19 mmol/g). Standard preprogrammed coupling and deprotection cycles were applied. The deprotection was achieved using 20% piperidine in DMF with microwave heating at 90 °C for 100 s, followed by three DMF washing steps. The couplings were performed using 5 eq. of Fmoc-protected amino acid, 5 eq. of *N,N'*-diisopropylcarbodiimide (DIC) and 5 eq. of 2-cyano-2-(hydroxyimino)acetate (Oxyma) in DMF with microwave heating at 90 °C for 3 min, followed by two DMF washing steps. Single couplings were applied

to the first 15 residues while two coupling cycles were used for residue 16 and further.

The resin with synthesised peptide was then transferred to an SPS tube and labelled using 3 eq. of 5(6)-Carboxyfluorescein, 3 eq. of DIC and 3 eq. of Oxyma in DMF for 16 h, followed by washing with 10 ml of 20% piperidine in DMF two times for 5 min and three times with 10 ml DMF. After further washing three times with 10 ml dichloromethane and two times with 10 ml of diethylether the resin was dried under vacuum for 30 min. To deprotect the side chains and cleave the peptide from the resin, the resin was incubated on a rotator for 3 h with 10 ml of cleavage mix (92.5% trifluoroacetic acid, 2.5% water, 2.5% triisopropylsilane, 2.5% 2,2'-(Ethylenedioxy)diethanethiol (DODT)) and filtered. The filtrate was concentrated to ca 1 ml under a stream of nitrogen, and the peptide was precipitated by the addition of 10 ml of ice-cold diethylether and isolated by centrifugation (6000 rpm for 5 min). The precipitate was resuspended in 10 ml of ice-cold diethylether and isolated by repeating the centrifugation step. After decanting the diethylether, the precipitate was allowed to dry for 30 min, dissolved in 5 ml of 1% (v/v) acetic acid and freeze-dried.

### Peptide purification

Peptides were dissolved in 4–10 ml of 1:1 mixture of acetonitrile and water and purified using an Agilent 1260 infinity system equipped with a UV detector and fraction collector on Kinetex EVO 5 µm C18 100 Å 21.2 × 250 mm reverse phase column. About 1–4 ml of the peptide solution was injected and 25 min gradient of 20–40% acetonitrile in water with 0.1% formic acid additive was run at 10 ml/min. The fractions containing peptides were pooled and freeze-dried.

The identity of the peptides was confirmed by high-resolution mass spectrometry on Bruker maXis II™ Impact QToF spectrometer using electrospray ionisation. The purity was determined by analytical HPLC on Agilent 1290 Infinity II system using Ascentis peptide column and gradient 5–95% of acetonitrile in water with 0.1% trifluoroacetic acid additive at 0.5 ml/min for 10 min. The HR-QToF(ESI)MS analysis and analytical HPLC traces for peptides used in this study are available in the source data.

## Data availability

The crystal structure has been deposited to the Protein Data Bank with accession code 8PR7 (https://www.rcsb.org/structure/8PR7). The assignment covering human CEP192 442–533 has been deposited to the BMRB under deposition 52044 (https://bmrb.io/data_library/summary/?bmrbId=52044). Source data for images presented in the figures has been deposited to the BioImage Archive under the accession number: S-BIAD1268 (https://www.ebi.ac.uk/biostudies/BioImages/studies?query=S-BIAD1268).

The source data of this paper are collected in the following database record: biostudies:S-SCDT-10_1038-S44318-024-00240-z.

## Peer review information

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

## Acknowledgements

This research was supported by the Biotechnology and Biological Sciences Research Council (BBSRC) (BB/V003577/1) and a BBSRC Flexible Talent Mobility Accounts grant to facilitate collaboration with the University of Georgia (BB/X01763X/1). We thank Diana Gimenez-Ibanez for peptide production, Iain Manfield for ITC (WT094232MA 094232/Z/10/Z), James Ault for HDX-MS and Arnout Kalverda for NMR.

We thank the beamline scientists of Diamond i04 and the Sir William Dunn School of Pathology FACS facility scientists for their assistance with data collection. We thank Francis Barr for the use of their upright microscope system. We thank Niloufer Irani and the Micron Advanced Bioimaging Facility (supported by Wellcome Strategic Awards 091911/B/10/Z and 107457/Z/15/Z) for their training and assistance in acquiring live cell microscopy data.

## Author contributions

**James Holder**: Conceptualisation; Formal analysis; Validation; Investigation; Visualisation; Methodology; Writing—original draft; Writing—review and editing. **Jennifer A Miles**: Conceptualisation; Formal analysis; Validation; Investigation; Visualisation; Methodology; Writing—original draft; Writing—review and editing. **Matthew Batchelor**: Investigation; Methodology. **Harrison Popple**: Investigation. **Martin Walko**: Resources; Investigation. **Wayland Yeung**: Investigation; Methodology. **Natarajan Kannan**: Supervision; Writing—original draft. **Andrew J Wilson**: Resources; Funding acquisition; Investigation. **Richard Bayliss**: Conceptualisation; Supervision; Funding acquisition; Investigation; Writing—original draft; Writing—review and editing. **Fanni Gergely**: Conceptualisation; Supervision; Funding acquisition; Investigation; Writing—original draft; Writing—review and editing.

Source data underlying figure panels in this paper may have individual authorship assigned. Where available, figure panel/source data authorship is listed in the following database record: biostudies:S-SCDT-10_1038-S44318-024-00240-z.

## Disclosure and competing interests statement

The authors declare no competing interests.

# Expanded View Figures

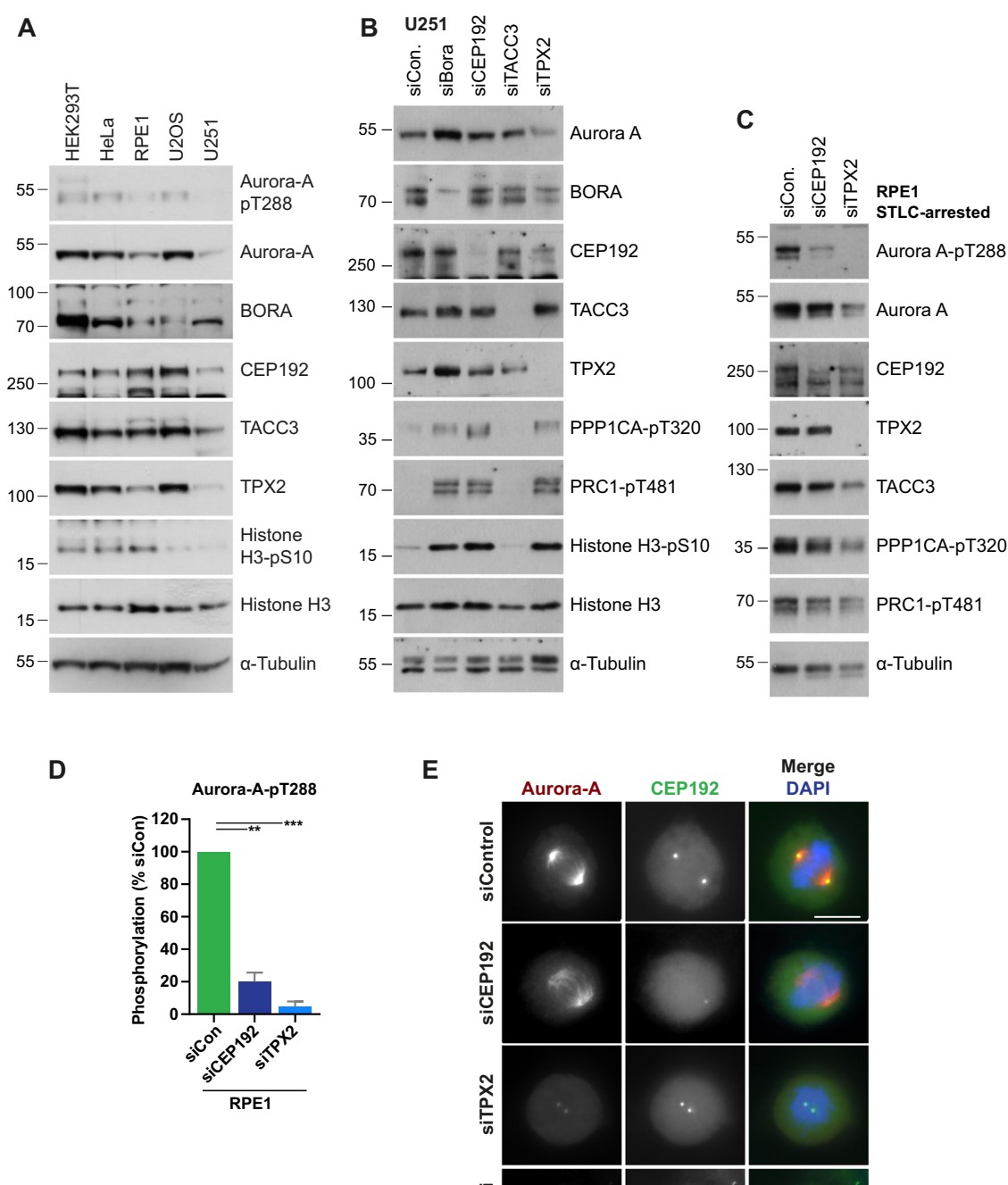

**Figure EV1. Aurora-A colocalizes with CEP192 in the absence of TPX2 or Aurora-A activity.**

(A) Western blot analysis of Aurora-A and its binding partners in a panel of asynchronous cell lines. (B) Western blot analysis of asynchronous U251 cells treated with the indicated siRNA (48 h). Antibodies against the mitotic phosphorylations PPP1CA-pT320, PRC1-pT481 and Histone H3-pS10 highlight the enrichment of mitotic cells following depletion of certain Aurora-A co-activators. (C) Western blot analysis of RPE1 cells treated with the indicated siRNA (48 h total) and arrested in mitosis with STLC (20 h). (D) Densitometric quantification of Aurora-A-pT288 signal from (C). Grey bars indicate mean ± S.D (*n* = 3 biological replicates). Exact *p* values (L-R): 0.0016, 0.0003. (E) Immunofluorescence images of U251 cells treated with siRNA as in (B), prior to 30 min incubation with DMSO control or Aurora-A inhibitor and then methanol fixation. Antibodies against Aurora-A and CEP192 are red and green in merged images, respectively, with DNA stained with DAPI (blue). Data information In (D), *p* values are denoted as follows: ***\*\*\**p* < 0.001, \*\**p* < 0.01, (Welch's *t*-test). The scale bar in (E) represents 10 μm. Source data are available online for this figure.

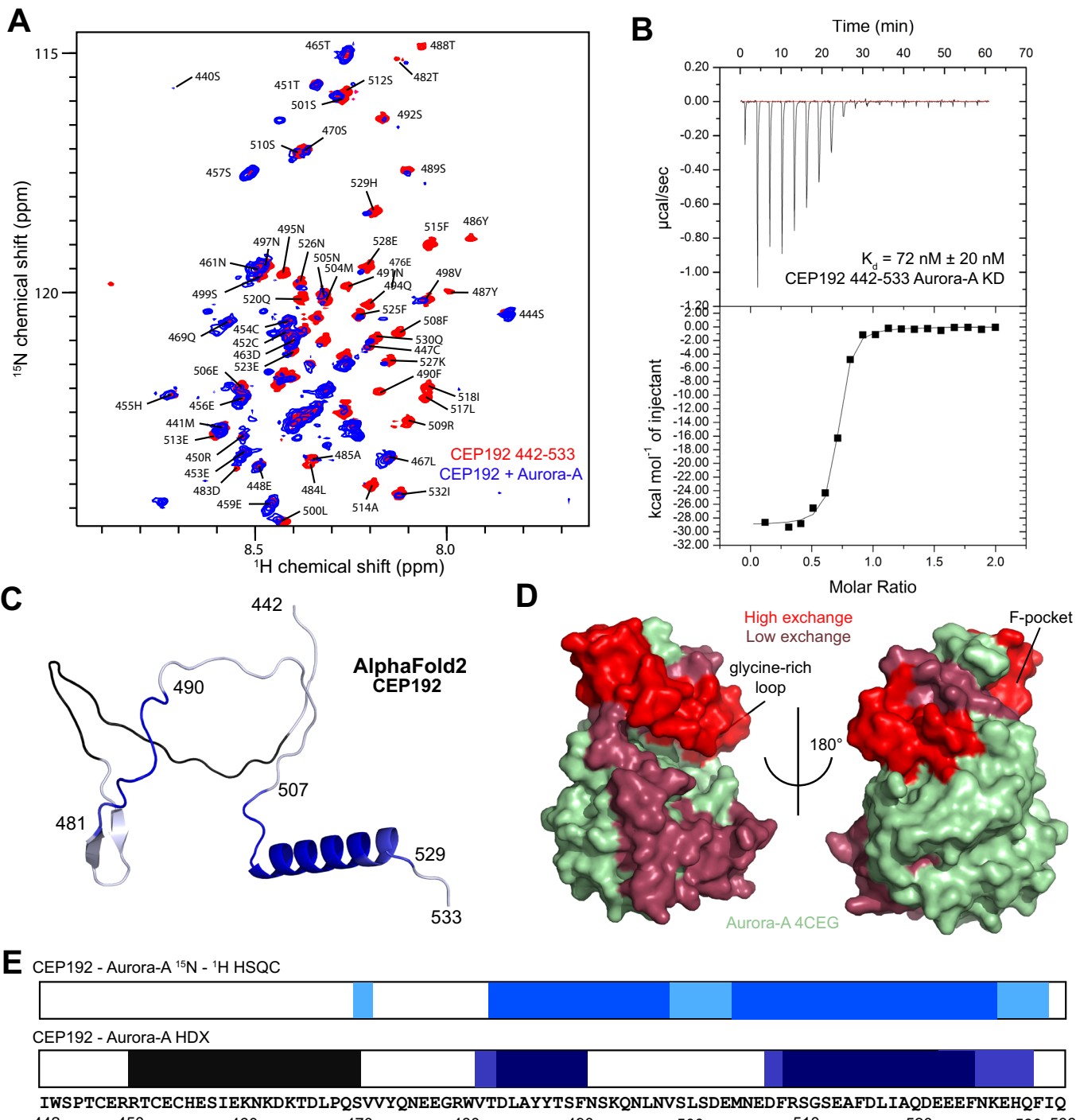

**Figure EV2.   CEP192 binds Aurora-A kinase domain with an extensive and high-affinity interaction.**

(A) $^{1}$H-$^{15}$N HSQC recorded on *Homo sapiens* CEP192 442–533 in the absence (red) and presence (blue) of Aurora-A 122–403 C290A C393A. A significant number of CEP192 peaks disappeared in the presence of Aurora-A, and following assignment, these residues were mapped between 470 to 533. (B) Isothermal titration calorimetry experiment showing titration of CEP192 442–533 into Aurora-A kinase domain (122–403 C290A C393A D274N). The measured $K_d$ was 72 nM, with a molar ratio of 0.67 from two experimental repeats. (C) Mapping of the HDX-MS data from CEP192 442–533 in the presence of Aurora-A 122–403 C290A C393A onto an AlphaFold2 model of human CEP192 442–533. The dark blue shows regions with a greater than 10% difference in uptake, with lighter blue for differences in uptake between 5 and 10%. Regions where there was no change are shown in grey, with the region in black where no peptides were identified. (D) Mapping of the HDX-MS experiment as (C), mapped onto the surface of human Aurora-A$^{CAKD}$ (PDB: 4CEG, shown in light green). The darker the red, the greater the difference in uptake when CEP192 468–533 is present. (E) Summary of the results of the interaction mapping data from NMR and HDX-MS on CEP192 442–533. The residues in the $^{1}$H-$^{15}$N-HSQC with the highest peak loss are shown in the top section in blue, with dark blue indicating peaks that completely disappeared and light blue indicating peaks that decreased and shifted significantly. The protected residues in the CEP192 HDX-MS are shown in the bottom section in blue, with peptides where uptake differed by over 10% shown in dark blue and uptake differences between 5 and 10% shown in light blue. The region in black shows where no peptides were identified. Source data are available online for this figure.

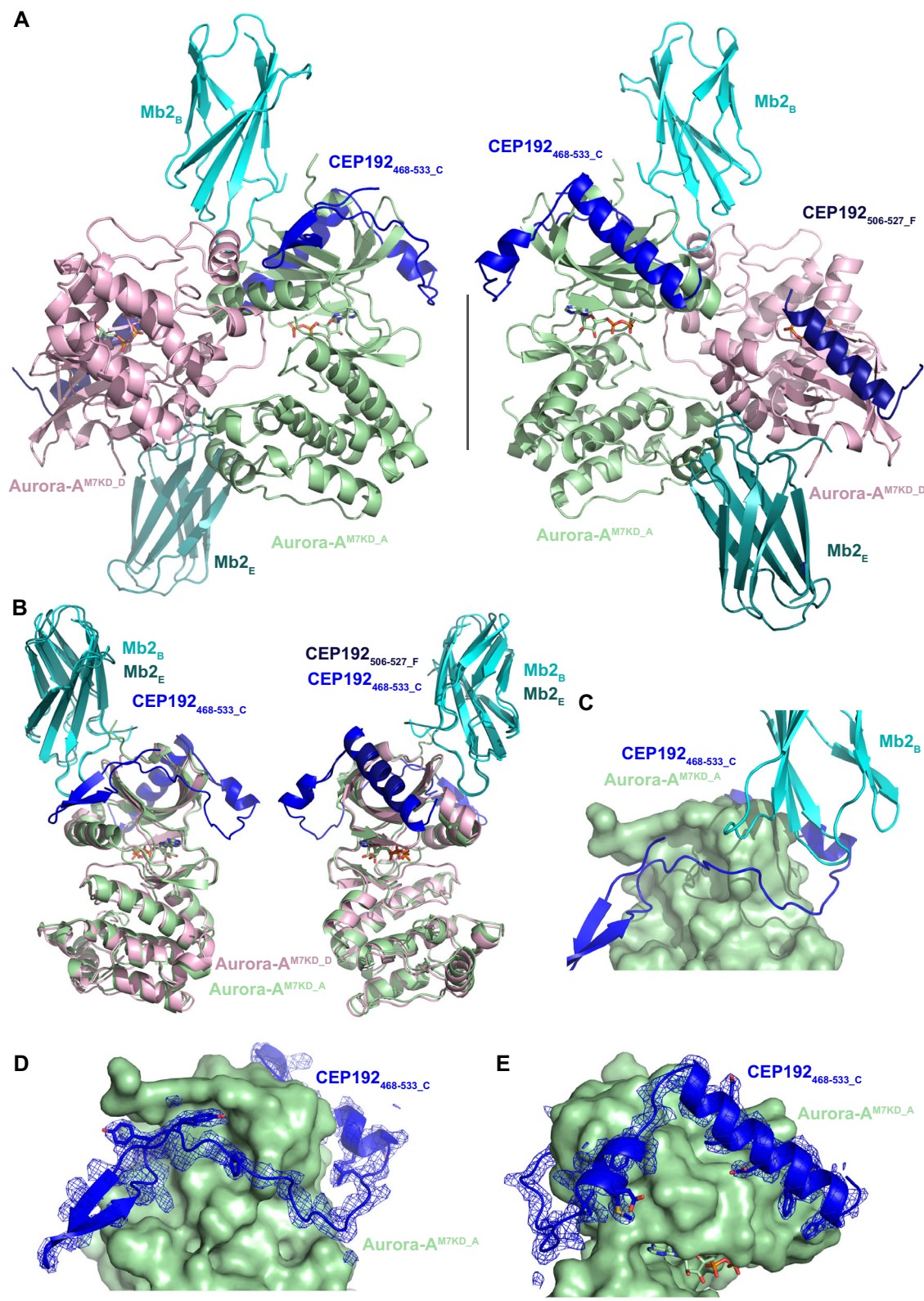

◀ **Figure EV3.    Contents of the asymmetric unit of the crystal structure of Aurora-A$^{M7KD}$ bound to CEP192$_{468\text{-}533}$ and the inhibitory monobody (Mb2).**

(A) Cartoon representation of the asymmetric unit of the crystal structure of Aurora-A$^{M7KD}$ bound to CEP19$_{468\text{-}533}$ and the inhibitory monobody. There are two copies of Aurora-A$^{M7KD}$ (Chain A in light green, Chain D in light pink), 2 copies of the inhibitory monobody (Chains B and E in cyan) and 2 copies of CEP192$_{468\text{-}533}$ (Chains C and F in blue and dark blue). Only part of the CEP192 was visible in chain F (residues 506–526). (B) Overlay of the cartoon representations of the two Aurora-A$^{M7KD}$ copies in the asymmetric unit. Chain A is shown in light green, with chain D in light pink. There are no significant differences between the two copies in the asymmetric unit. (C) Cartoon representation of the complex between CEP192$_{468\text{-}533\_C}$ (dark blue) and Aurora-A$^{M7KD\_A}$ (light green) with a symmetry-related copy Mb2 (teal). The interface was analysed on PDBePISA, giving an interface score of 0.00 suggesting that this is merely a crystal contact and not a biologically relevant interface. (D) Representation of the electron density around CEP192$_{468\text{-}533}$ (dark blue) when bound to Aurora-A$^{M7KD}$ (light green). The mesh represents a 2mFo-DFc map contoured at 1.2σ. (E) A second view of the electron density around CEP192$_{468\text{-}533}$ (dark blue) when bound to Aurora-A$^{M7KD}$ (light green) to show the αS/αL region. The mesh represents a 2mFo-DFc map contoured at 1.2σ. Data information: Regions modelled for the different chains were 124–275 and 289–389 (chain A, Aurora-A$^{M7KD}$); 3–93 (chain B, Mb2); 468–531 (chain C, CEP192$_{468\_533}$); 126–277 and 290–388 (chain D, Aurora-A$^{M7KD}$); 3–93 (chain E, Mb2); 506–527 (chain F, CEP192$_{468\_533}$).

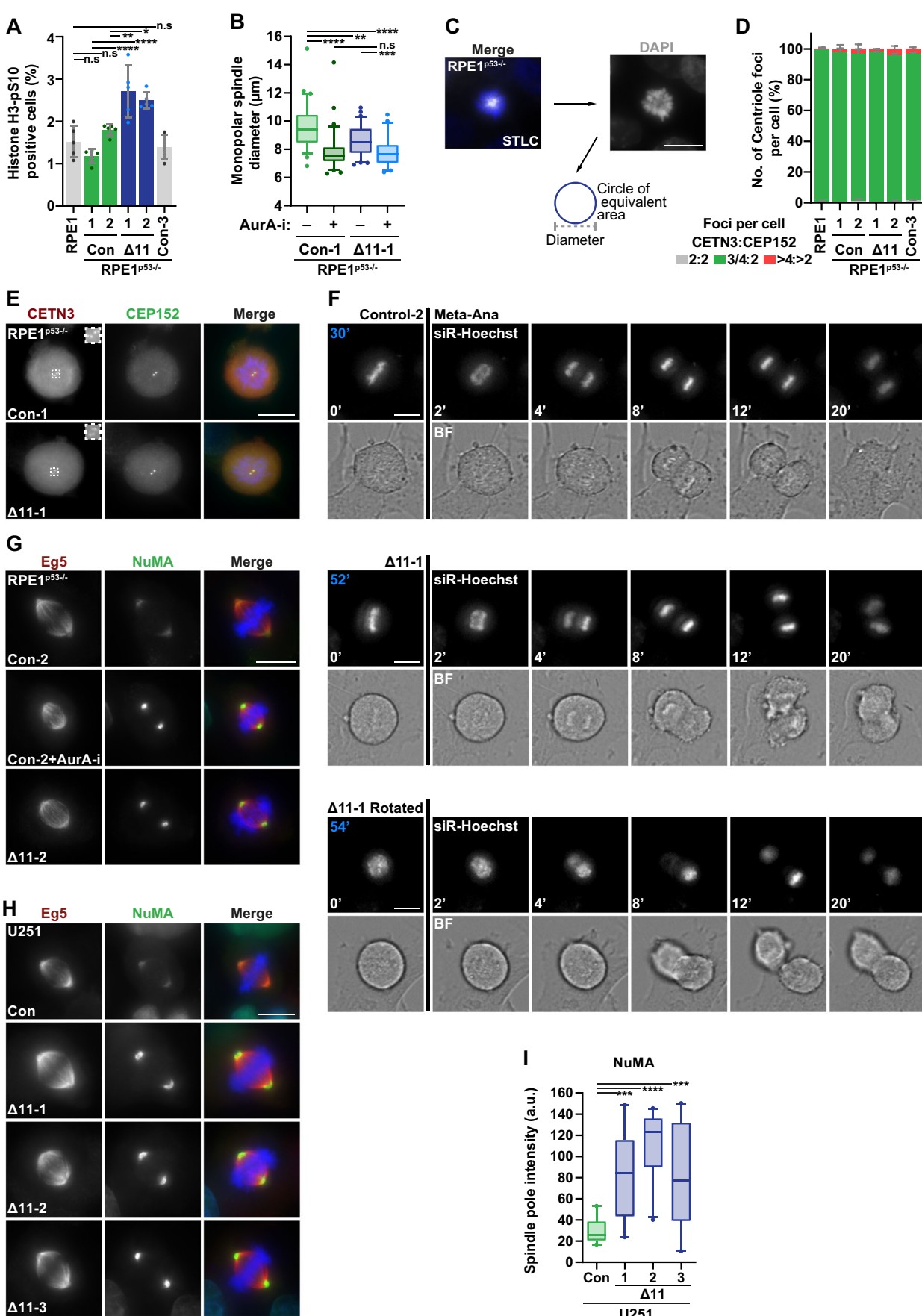

◀ **Figure EV4. The Aurora-A:CEP192 complex is required to establish proper mitotic spindle length and orientation.**

(A) Bar chart showing the percentage of Histone H3-pS10 positive cells in multiple RPE1 and RPE1-derived cell lines ($n = 5$ biological replicates). Exact $p$ values from (bottom-top): 0.4834, 0.6124, <0.0001, <0.0001, 0.0026, 0.0264, 0.6927. (B) Box plot of monopolar spindle diameter in control and Δ11-1 RPE1$^{p53-/-}$ cells treated with either DMSO control or Aurora-A inhibitor for 30 min prior to methanol fixation ($n = 3$, >15 cells/biological replicate). Exact $p$ values from (bottom-top): 0.001, >0.9999, <0.0001, 0.0048, <0.0001. (C) Schematic detailing how monopolar spindle diameter, plotted in (B), was calculated based on each DAPI-stained DNA signal area (blue). (D) Bar chart showing the number of CETN3 and CEP152 foci in multiple STLC-arrested RPE1 and RPE1-derived cell lines ($n = 3$, ≥50 cells/biological replicate). (E) Representative immunofluorescence images of control and Δ11 RPE1$^{p53-/-}$ cells from (D). Antibodies against CETN3 and CEP152 are red and green in merged images, respectively, with DNA stained with DAPI (blue). Enlarged inserts centred on centriolar foci have sides 2 μm in length. (F) Representative images of control and Δ11 RPE1$^{p53-/-}$ progressing through anaphase, siR-Hoechst and brightfield (BF) channels are shown for each image. The black line indicates the metaphase-anaphase transition (Meta-Ana). Numbers in white indicate time relative to Meta-Ana, while numbers in blue indicate the time from NEBD-Ana (nuclear envelope breakdown to anaphase onset) for that cell (see also Fig. 7A-B). For Δ11-1 rotated, note how one of the daughter cells moves out of focus, indicating an out-of-plane division. (G, H) Immunofluorescence images of methanol fixed control and Δ11 (G) RPE1$^{p53-/-}$ and (H) U251 cells. Antibodies against Eg5 and NuMA are red and green in merged images, respectively, with DNA stained with DAPI (blue). Quantification of (G) is found in Fig. 7I, J. (I) Box plot of NuMA spindle pole signal intensity in U251 cells, with representative images shown in (H) ($n = 2$, ≥10 cells/biological replicate). Exact $p$ values (L-R): <0.0001, <0.0001, <0.0001. Data information In (A) and (D) RPE1 indicates the parental p53$^{+/+}$ line from which other clones were derived. Con-3 is an additional RPE1$^{p53-/-}$ control cell line generated previously within the lab. Grey bars in (A) and (D) indicate mean ± S.D. Box plots in (B) and (I) indicate the median and interquartile ranges (25th–75th percentile) with coloured whiskers representing 5th–95th percentile ranges. $p$ values are denoted as follows: ****$p < 0.0001$, ***$p < 0.001$, **$p < 0.01$, *$p < 0.05$, n.s not significant (A ANOVA, B, I Mann–Whitney test). Scale bars in (C) and (E–H) represent 10 μm. Source data are available online for this figure.

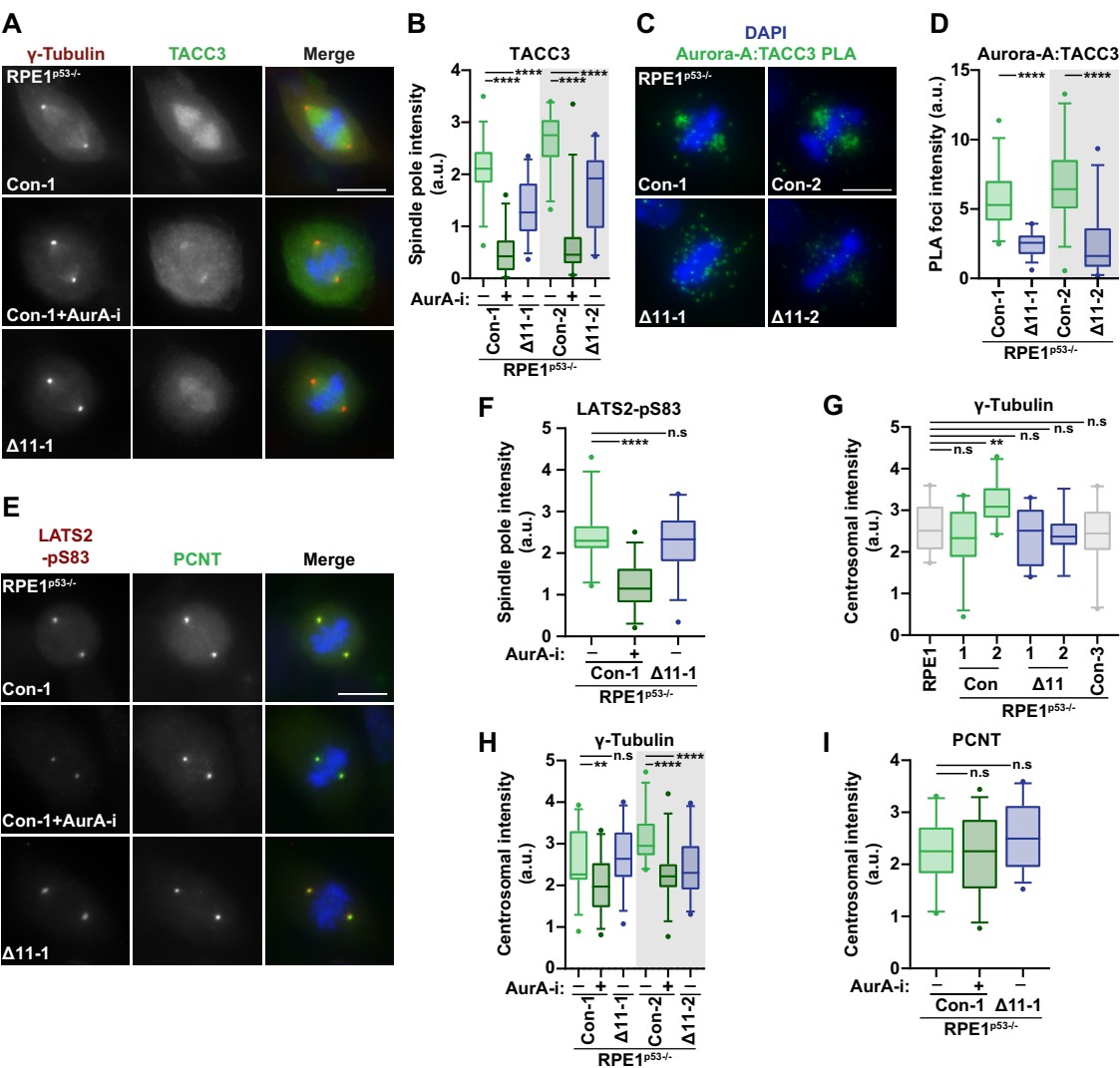

**Figure EV5. Loss of the interaction between Aurora-A and CEP192 impairs TACC3 spindle recruitment.**

(A) Immunofluorescence images of control and Δ11 RPE1$^{p53-/-}$ cells treated with either DMSO control or Aurora-A inhibitor (30 min) prior to methanol fixation. Antibodies against γ-tubulin and TACC3 are red and green in merged images, respectively, with DNA stained with DAPI (blue). (B) Box plot TACC3 spindle signal intensity in RPE1$^{p53-/-}$ cells, with representative images shown in (A) ($n = 2$, ≥15 cells/biological replicate). Exact $p$ values (L-R): <0.0001, <0.0001, <0.0001, <0.0001. (C) Proximity ligation assay (PLA) between Aurora-A and TACC3 specific antibodies in control and Δ11 RPE1$^{p53-/-}$ cells. PLA signal is green in merged images with DNA stained with DAPI (blue). (D) Box plot of Aurora-A:TACC3 PLA signal intensity in mitotic RPE1$^{p53-/-}$ cells, with representative images shown in (B) ($n = 2$, ≥15 cells/biological replicate). Exact $p$ values (L-R): <0.0001, <0.0001. (E) Immunofluorescence images of control and Δ11 RPE1$^{p53-/-}$ cells treated as in (A). Antibodies against LATS2-pS83 and PCNT are red and green in merged images, respectively, with DNA stained with DAPI (blue). (F) Box plot of LATS2-pS83 spindle pole signal intensity in RPE1$^{p53-/-}$ cells, with representative images shown in (E) ($n = 2$, ≥15 cells/biological replicate). Exact $p$ values (L-R): <0.0001, 0.6749. (G) Box plots of γ-tubulin centrosomal signal intensity in multiple RPE1 and RPE1-derived cell lines ($n = 2$, 10 cells/biological replicate). RPE1 indicates the parental p53$^{+/+}$ line from which other clones were derived. Con-3 is an additional RPE1$^{p53-/-}$ control cell line generated previously within the lab. Exact $p$ values (L-R): 0.1865, 0.0022, 0.4935, 0.3363, 0.6783. (H) Box plot of γ-tubulin centrosomal signal intensity in RPE1$^{p53-/-}$ cells treated as in (A), with representative images also shown in (A) ($n = 2$, ≥15 cells/biological replicate). Exact $p$ values (L-R): 0.0017, 0.3941, <0.0001, <0.0001. (I) Box plot of PCNT centrosomal signal intensity in RPE1$^{p53-/-}$ cells treated as in (A), with representative images shown in (E) ($n = 2$, ≥15 cells/biological replicate). Exact $p$ values (L-R): 0.7364, 0.0919. Data information Box plots in (B, D) and (F–I) indicate the median and interquartile ranges (25th–75th percentile) with coloured whiskers representing 5th–95th percentile ranges. Grey shading in (B, D, H) denotes independently completed biological replicates. $p$ values are denoted as follows: ****$p < 0.0001$, **$p < 0.01$, n.s not significant (Mann–Whitney test). Scale bars in (A, C, E) represent 10 μm. Source data are available online for this figure.

