## [Peer Review File · The EMBO Journal]

CEP192 localises mitotic Aurora-A activity by priming its interaction with TPX2

Fanni Gergely, Richard Bayliss, James Holder, Jennifer Miles, Matthew Batchelor, Harrison Popple, Martin Walko, Wayland Yeung, Natarajan Kannan, and Andrew Wilson

Corresponding author(s): Fanni Gergely (fanni.gergely@bioch.ox.ac.uk) , Richard Bayliss (r.w.bayliss@leeds.ac.uk)

Review Timeline:

Submission Date:	16th Jan 24
Editorial Decision:	9th Feb 24
Revision Received:	15th Jul 24
Editorial Decision:	6th Aug 24
Revision Received:	27th Aug 24
Accepted:	3rd Sep 24

Editor: Hartmut Vodermaier

Transaction Report:

Dr. Fanni Gergely
University of Oxford
Department of Biochemistry
South Parks Road
Oxford OX1 3QU
United Kingdom

9th Feb 2024

Re: EMBOJ-2024-116662
CEP192 drives spatio-temporal control of mitotic Aurora-A

Dear Fanni,

Thank you again for submitting your study on spatiotemporal Aurora A control to The EMBO Journal. I sent it to three expert referees, who have now returned the reports copied below. As you will see, all referees acknowledge the interest and potential importance of your findings, as well as the extensive analyses. Nevertheless, they do raise a number of conclusiveness and presentational issues that should be addressed prior to publication.

I would in this light be happy to consider a revision further for The EMBO Journal, and invite you to start preparing such a revised version. Since it is our policy to consider only a single round of major revision, it would in this case be very helpful to pre-discuss how the referees' points may best be clarified - I would therefore encourage you to get back to me with a tentative response letter/revision plan already during the early stages of the revision work, and I'd be happy to use this as a basis for follow-up discussions e.g. via a video call. I should add that we could also offer extension of the default three-months revision period if needed, with our 'scooping protection' (meaning that competing work appearing elsewhere in the meantime will not affect our considerations of your study) remaining of course valid also throughout this extension.

Detailed information on preparing, formatting and uploading a revised manuscript can be found below and in our Guide to Authors. Thank you again for the opportunity to consider this work for The EMBO Journal, and I look forward to hearing from you in due time.

With kind regards,

Hartmut

4) Each main and each Expanded View (EV) figure should be uploaded as individual production-quality files (preferably in .eps,

.tif, .jpg formats). For suggestions on figure preparation/layout, please refer to our Figure Preparation Guidelines: <http://bit.ly/EMBOPressFigurePreparationGuideline>

9) Digital image enhancement is acceptable practice, as long as it accurately represents the original data and conforms to community standards. If a figure has been subjected to significant electronic manipulation, this must be clearly noted in the figure legend and/or the 'Materials and Methods' section. The editors reserve the right to request original versions of figures and the original images that were used to assemble the figure. Finally, we generally encourage uploading of numerical as well as gel/blot image source data; for details see: embopress.org/page/journal/14602075/authorguide#sourcedata

At EMBO Press, we ask authors to provide source data for the main manuscript figures. Our source data coordinator will contact you to discuss which figure panels we would need source data for and will also provide you with helpful tips on how to upload and organize the files.

In the interest of ensuring the conceptual advance provided by the work, we recommend submitting a revision within 3 months (9th May 2024). Please discuss the revision progress ahead of this time with the editor if you require more time to complete the revisions. Use the link below to submit your revision:

Link Not Available

Referee #1:

Holder and Miles et al investigates Aurora A binding to CEP192 and builds a model on how this interaction leads to a handover to TPX2. Different structural methods are logically combined with gene-editing and cell biological assays, making this a very solid and interesting manuscript. The experiments are in my opinion well performed and the manuscript is well written. The spatio-temporal control of Aurora A activity is important for how mitosis is regulated. I am very positive to the manuscript but have some suggestions for minor changes.

I miss a description of how Bora would fit into the model? The Aurora A-Bora complex is present independent of phosphorylation changes and could potentially compete with the other interactions described here (Lössl ACS 2016; Tavernier Nat comm 2021). A large part of Bora is degraded when PLk1 activity reaches the cytoplasm prior to mitosis, but a subset is stabilized in mitosis and overexpression or non-degradable versions of Bora have been suggested to impact on Aurora A-Tpx2 activity (Chan et al, Chromosoma 2008; Seki et al, JCB 2008; Feine et al cell cycle 2014; Bruinsma et al JCS 2014). I think a possible contribution of the degradation of a large part of the pool of Bora is relevant to mention in addition to the present discussion on CENP152, TPX2 and TACC3.

Fig 5H and EV5F. The FACS in EV5F does not include 8N population (just outside of figure). Were the gates set so that 8N cells are included in the statistics? Although it would not change any major conclusion, otherwise figure 5H is not correct and the wide range in polyploid populations could possibly be attributed to different PI intensity (which is apparent in EV5F).

Fig EV1. There seem to be a difference in size in Bora blot in A and B?

Fig 2C. Same color of text in fig for high and low exchange?

Fig 2C and E. It is not indicated what color black represents.

Fig EV2C. Are these colors really blue and red as stated in text?

Referee #2:

In this manuscript, Holder and colleagues investigate the structural determinants of the complex between the Aurora-A kinase and its main centrosomal partner, Cep192. By in vitro assays, crystallography and in silico approaches they identify and characterize two distinct regions of binding within the previously identified Aurora-A binding domain. They then explore the specific function of Aurora-A bound Cep192 by CRISPR-CAS9-mediated deletion of the Cep192 region involved in the binding. Cellular experiments confirm the importance of the Aurora-A/Cep192 axis in determining correct spindle length, organization and orientation and indicate the interaction as a pre-requisite for the formation of the Aurora-A/TPX2 complex on spindle microtubules, proposing Cep192 as a main determinant of spatio-temporal regulation of mitotic Aurora-A. Overall the study provides interesting novel observations going from structural characterization to cellular data. Still, I feel that prior to publication a number of experiments should be performed to support conclusions and better place the work in the context of the existing literature, as detailed below.

Major points:

1. Throughout the manuscript, previously published data should be mentioned/discussed more accurately. "As expected" or "in line with previous reports" should be accompanied by citations. The mention of the structure resembling the α L region of CEP192 506-527 fused to the N-terminus of Aurora-A, with reference to Park et al., 2023, seems to be a critical point in the text. It's suggested that the work by Park is cited earlier in the paper for better contextualization. Additionally, a more in-depth comparison with the structure and data from Park's work is recommended to provide a clearer understanding of the similarities and differences between the two studies. This could enhance the comprehensibility and relevance of the current discussion in relation to the prior research.
2. Experiments in Figure 1 and EV1 constitute a premise to the subsequent structural investigation, but in my opinion lack novelty. Selection of the 2 cell lines is based on a WB in asynchronous cultures, that in my experience is not very reliable for cell-cycle regulated proteins, and without providing a normalized quantification; mitotic extracts would have been more informative. The 2 lines are then not really compared, therefore I find this characterization unnecessary. Results mostly recapitulate what has been shown in previous papers and could largely be moved into Supplementary. Showing results in the non-transformed RPE cells may add novelty. Given the authors' intention to highlight the non-redundancy of TPX2 and Cep192, it would be interesting to see AurkA and p-AurkA levels and localization also in the Cep192i/TPX2i condition shown in EV1B.
3. Page 6: authors attempt to validate Cep192-468-533 as the minimal region essential for interaction with Aurora-A by using ITC to determine binding affinity. Which is the rationale behind the selection of Cep192-468-533? The values obtained from ITC with Cep192-468-533 (71.4 nM in Fig S2 A, but 74nM in the text) are notably different from those obtained with fluorescence polarization (FP) in another phase of the study with Cep192-505-529 (Fig S2D, 1.4 μ M). How is this difference explained? FP analysis on Cep192-468-533 and ITC on Cep192-505-529 could strengthen the data. Importantly, a significant difference exists on a key point, i.e. by ITC the 442-533 fragment binds with a 70 nM Kd while similar fragments in Park et al. display μ M Kds. Can authors explain/comment on this?
Crystals diffracted at 2.76 Å were formed only in the presence of Mb2, but the latter is not shown in any Figure. Could the Mb2 presence interfere with Cep192 binding to Aurora? Are there any contacts between Mb2 and Cep192?
4. AurkA kinase assays suggest that Cep192 may act as an inhibitor of AurkA substrate phosphorylation. Given the overall activating function of Cep192 on AurkA in cells, the relevance of this inhibitory complex should be better investigated/discussed.
5. I find data with U251 Δ 11 cell lines confusing. WB blot and PLA characterization are missing. Based on FACS analysis results they are not analyzed further. On these bases I find U251 Δ 11 data are not informative in the actual form and should not be included.
6. Related to the previous point, the manuscript does not include analysis of chromosome segregation defects or genome instability in the Δ 11 cell lines. Based on the available data no conclusions should be drawn on genome stability (abstract, legends titles, discussion).
7. I have a concern about the choice to co-target p53 in the RPE cell lines. It has been shown that p53 has mitotic functions, in particular at centrosomes (Contadini et al., 2019). Did authors check the status of mitotic centrosomes in the WT-p53^{-/-} cell line vs controls? And, based on this and on p53/AurkA co-regulation, how can authors exclude a contribution of lack of p53 to the

observed phenotypes?

8. Authors describe two RPE- Δ 11 lines but most shown results are in one of them (Δ 11-1) only. Main results should be shown in both lines.

9. Is mitotic entry and progression normal in the Δ 11 cell lines? Are phenotypes measured in the same mitotic stages? In particular, are the pole-to-pole distance and spindle orientation measured in aligned metaphases only?

10. Mitotic phenotypes. It is not obvious why the spindle diameter is measured on DAPI and not tubulin staining. Parallel to tubulin area, tubulin intensity would also be informative. More accurate measurements would be required to draw conclusions on MT length. Spindle orientation in RPE metaphase cells should not be much higher than 10{degree sign}. Having set a first class of 0-30{degree sign} is therefore misleading. Smaller intervals should be shown. Additionally, given that the pole-to-pole distance varies and this can influence the calculation of the spindle angle, raw data of z-distances should be provided to unambiguously show that orientation is affected.

11. Why is PLA measured by area? Intensity (or number of spots) should also be shown, together with a negative control for each interaction (particularly for the "diffused" signals).

12. For a more robust demonstration of the conclusions drawn on the complexes with Cep192, TPX2 and TACC3 in the Δ 11 cell lines it is important to perform AurkA immunoprecipitation experiments.

13. The conclusion that "the residual spindle localisation of Aurora-A in R Δ 11-1 cells is still dependent on TPX2 because TPX2 depletion in these knockouts further diminishes the Aurora-A signal" should be supported by quantitative analysis.

14. Authors propose that the lack of effects on centrosome maturation are explained by the fact that "Deletion of CEP192(exon 11) should not impact the interaction between CEP192 and PLK1, and...centrosomal PLK1, previously activated by BORA:Aurora-A, facilitates PCM expansion". This needs support by experimental data by quantifying Plk1 localization and phosphorylation, and possibly by assessing AurkA levels and activation in G2 cells.

15. pTACC3 should be tested to support the TACC3 proposed mechanism and an explanation for lack of p-Lats2 decrease should be provided. Eg5 is shown but not quantified. Since existing data indicate a decrease of centrosomal Eg5 in Cep192i, it would be interesting to know how it is in the Δ 11 cells.

16. Results in Δ 11 cells are very similar to those previously obtained by RNAi/reconstitution or by transfection of dominant negative forms of Cep192. Interestingly, authors show that TPX2 interaction is also decreased, and propose that Cep192-induced AurkA autophosphorylation is required for TPX2 efficient binding. On the other hand, since Cep192 is required for localizing AurkA to the mitotic centrosome, it could also be that the decrease that they observe at spindle poles reflect this effect and that this then results in less AurkA/TPX2 interaction. More experimental data should be provided to support authors' model. For example, how is total AurkA and AurkA/TPX2 PLA in WT + Aurki condition? How does a kinase dead mutant behave?

Minor points:

- Page 5: the lack of a clear and concise explanation of the rationale behind the transition from CEP192-405-649 (Appendix Figure S1B) to CEP192-442-533 in the experimental design and analysis makes it challenging for readers to grasp the logical flow of the text. Clarification on this transition would significantly enhance the overall clarity and coherence of the text.
- Fig.1A. Histone H3-pS10, PPP1CA-pT320, PRC1-pT481 are included and not mentioned, as AurBi and Plki controls. Cep192 signal is not clear. Bora is missing and total AurkA is not quantified. Both 1A and EV1B indicate fluctuations of analyzed proteins in the RNAi samples that may be relevant but are not commented (e.g, EV1B, Cep192 decrease in TPX2i and Bora fluctuations, or in 1A TACC3 levels).
- panels in Figures should be ordered in the order they appear in the text;
- sometimes Δ 11-R1 is used instead of R 11-1
- cep192 Δ 11 migration is not completely consistent in the WBs shown in Expanded View figures.

Referee #3:

This manuscript by Holder, Miles, and colleagues reports that the centrosomal protein CEP192 plays a crucial and unexpected role in the accumulation of the activation loop-phosphorylated (and therefore active) form of the Aurora A kinase. Aurora A localises predominantly to the spindle poles and to the spindle during mitosis. It is known to achieve these different localisations by interacting with specific binding partners. TPX2 and TACC3 are spindle-associated partners of Aurora A, while CEP192 is a PCM component surrounding the centrosome. Yet another partner, Bora, is predominantly cytosolic. The authors performed a very thorough biochemical and biophysical characterisation of the CEP192-Aurora A complex. The main conclusions from this part of the work, which extends through Figures 2-4, are that 1) CEP192 binds competitively with the other above-mentioned factors (TPX2 and TACC3); 2) It may act as an activator under specific conditions, but in its most fundamental, high-affinity binding state, it seems to act as an inhibitor of Aurora A kinase activity. In the rest of the manuscript, described in Figures 1 and

5-6, the authors present data that CEP192 appears to act at the apex of an activation pathway for Aurora A and is essential for the accumulation of the active form of Aurora A, identified by phosphorylation on T288 (P-T288). The authors reach this conclusion not only through more classical RNAi-based depletion experiments, but also by generating cell lines expressing a mutant CEP192 deleted of the Aurora A binding region. Collectively, these experiments revealed that CEP192 is required for cells to achieve a normal spindle lengths and spindle orientation relative to a substrate, and more generally to maintain the correct ploidy.

Overall, the results presented in this manuscript support the concept that CEP192 is a master regulator of Aurora A activation. This conclusion emerges with clarity from the presented experiments and is surprising and exciting. The mechanistic part of the work and the cell biological analyses do not bring about a fully coherent picture yet. The reason for this is that there is no clear explanation for why a potential biochemical inhibitor of Aurora A, like CEP192, is also absolutely necessary for Aurora A activation. This should not be seen as a major limitation in our opinion, as the main conclusions of this very data-rich study are sufficiently novel to guarantee the general interest of the paper. So we are supportive of the work and would like to see it published. However, we would also like the authors to consider the following points (in no specific order):

-The presentation of the manuscript could be improved significantly. We encourage the authors to consider the following five main points:

a) The Abstract is not sufficiently informative and should be rewritten to communicate the papers's main points. E.g. which "spindle-associated partners"? The following sentence repeats the concept that the binders are competitors, but forgets to highlight the interesting apparent contradiction that CEP192 is biochemically an inhibitor, which is not to be read anywhere in the abstract. The apparent paradox with the biological observations is not highlighted. We would recommend avoiding using "license" in the absence of clear evidence of a licensing step. CEP192 is necessary for the accumulation of phosphorylated Aurora A. How it does it, remains unclear. The final sentence seems a cut-and-paste from troves of other papers where authors hope to awake the editor's interest by invoking genome stability. We would recommend giving a more specific and informative conclusion.

b) The description of the structural work feels endless. Sorry, we don't want to sound insulting and we are aware of the massive amount of work that went into Figure 2-4, but that remains a poor reason to launch a lot of apparently redundant data onto readers. In essence, we think that the crystal structure (obtained through elegant thinking on how to eliminate unwanted crystal contacts) is just enough for the main paper, and the rest (HDX, NMR, AF2 predictions), being confirmatory, should be presented in the extended or supplementary data. I.e. we would only present Figures 3 and 4 and would GREATLY streamline the description of the other approaches, which should be presented rapidly and as confirmatory of the structure.

c) A major point of the paper is that CEP192 operates upstream of TPX2, but the evidence supporting this notion is limited if at all existing. Depletion of TPX2 leads to an equally dramatic disappearance of Aurora A activation loop phosphorylation. Without clearer mechanistic evidence, this conclusion seems unsupported.

d) We feel that the paper would make for an easier read if it began with the structure and biochemistry, and moved on to describe the surprising observation that CEP192 seems to be an activator, not an inhibitor. Figure 5 seems like a logical follow up to Figure 1, so we would suggest that the authors use Figure 1 as the first figure of the biological characterisation after the description of the structure. A most exciting result presented in Figure EV6A should be brought to the main figures unless the authors had a concern with it justifying its relation in the Extended result.

e) A comparative discussion of the results from the Park et al. Science Advances 2023 paper would add clarity. There are similarities and differences between the papers, and the authors have to try to rationalize them for their audience.

-Figure 1A, the depletion of CEP192 in this particular blot does not seem striking. It is much clearer in EV1.

-Panel 2E is described earlier than most other panels in the figure

-Hard to understand the logic of colour assignments in panels 2C-E. Colours for high and low exchange are too similar and hard to distinguish. Second, we assume there is a third category of rates of exchange that remain unaltered in the bound and unbound state. Shouldn't those be indicated as well with an appropriate colour?

-Figure 3A: Please indicate N and C-termini at least for CEP192 and name the main secondary structure elements (e.g. alphaS, alpha L)

-Figure 4G is too small and packed to be effective at making the important point it is trying to make. It is too out of context (which part of the structure is being shown?) for most readers to interpret. One of the structures of Aurora A-CEP could be removed, just TPX2 vs one of the CEP structures could be retained. An associated problem is that this figure only makes sense after discussion of panel H, as it provides an explanation for the paradoxical effects of the two parts of CEP192 that interact with Aurora A. This is an exciting point, but it needs to be presented in a way that readers can appreciate it. In 4H, why not show all Aurora A with the same color, rather than changing color for every panel? This would apply also to the CEP192 species.

Overall, this is an important figure. Maybe consider splitting figure 4 after panel F and create a new figure, which will be possible if Figure 2 is moved into the Supplementary.

- Figure 4C-E and every other sensor-gram, when multiple mutations are present, please separate them with a hyphen or slash
- Page 3, first line "...found at the centromeres and at the kinetochores"; In the following line replace "the spindle assembly checkpoint" with "a pathway" (reason being that Aurora B is credited to work in error correction, whereas its role in the SAC is still considered by many less well established)
- Page 4, the paragraph starting "Consistent with previous studies..." is somewhat confusing. First, it is unclear to us that the authors really quantified spindle localization. "Despite more Aurora-A..." We assume the authors mean "relative to TPX2" but it is not immediately clear and we would suggest being explicit.
- Page 6. Please rewrite: "The longer (CEP192-442-533) and minimal (CEP192-468-533) CEP192 constructs....Aurora-AM3KD: 72 nM and 74 nM, respectively (Figure 2B and AF S2A)
- Consider adding "Thus, in summary..." to the sentence starting "Using three tag-free..."
- Page 7: "Hot-spot" of what? Mutation?
- Page 7: Is the predicted "salvation energy" dimensionless? What is the source of the reported number, 2.37?
- Page 7, last paragraph, second line "(Figure 3C)" We think it should read Figure 3D
- Page 8, first paragraph: "Kd values > 5 μ M" should come together with a value for the control.
- Page 8, first paragraph: in "and also showed", "and" should be removed.
- Page 10, first paragraph: "had no significant effect on the kinase activity"; but the activity appears to increase.
- Figure 2 legend "over by over"
- Figure 5I-J, maybe easier to use grayscale? The two blue tones are not easily recognised in printouts
- Figure 7A: green for INCENP and green for Aurora A makes color resolution really poor. Add tags to C- and N-termini and consider replacing the arrows.
- Figure EV1B: does TPX2 control the stability of CEP192?
- "Deletion of CEP192(exon11) should not impact the interaction between CEP192 and PLK1." This is based on the idea that PLK1 binds to a different region, but Joukov et al. 2014 suggested co-dependence of PLK1 and Aurora A on CEP192. This point is somewhat tangential to the paper, yet quite relevant and the authors could have elected to test PLK1 localization. In the absence of a direct test, they could nevertheless refer to the observations of Joukov et al.
- Figure 5A-B: The scheme could be made clearer. Why is CEP192 depicted in blue and grey in panel A, green and grey in panel B (grey in scheme of excision sites). Why not use color consistently? "PLK1 and 4 binding" implies there is a single bipartite binding site for both, or individual binding sites currently displayed in red and dark red?
- Figure 5C-E is a control and could be moved to EV or SF figures.
- Figure 4F reports effects of CEP192 on Aurora A activity. CEP192 (468-533) is shown to be inhibitory (IC50 18 nM compared to alisertib 3 nM), while the region 500-533 is slightly activatory (claimed to have no significant effect). This analysis would benefit from a comparison of the authors' results with those of Park et al. 2023, who did not identify inhibitory activity with a longer fragment of CEP193. The authors claim that their results are consistent with those of Joukov et al. 2010. If we correctly understood Joukov et al. 2010, they obtained results with a dominant-negative approach where overexpression of the Aurora A binding region of CENP193 caused Aurora A displacement from the poles, and failed activation due to decreased local concentration. It is difficult to compare these results with those from a biochemical assay in vitro.
- Figure 4F: Another concern regarding this figure is that the authors use Aurora A at 10 nM concentration. Hard to see how TPX2 could cause 50 % max activation at 1 nM concentration. At this concentration, even assuming the highest possible binding affinity, only 10 % of the 10 nM kinase can be saturated (1 nM of 10 nM; half-maximal is expected at 5 with infinitely high affinity for a 1:1 complex). Furthermore, half maximal activation at 1 nM implies TPX2 binds Aurora A at this concentration, which would contradict the authors' claim that CEP192 is the tightest binding partner of Aurora A (70 nM).

-Figure 3: Aurora A M7KD interacts with Kd 1.4 μ M to FAM-CEP192 peptide, while CAKD construct Kd was 420 nM. The author state it is similar, but it is a 3-fold difference. Furthermore, another binding domain, Mb2, was added to improve diffraction quality. To understand this system quantitative, it would be necessary to test if there is any difference in affinity with and without Mb2. The authors should not worry about this, but should probably be more cautious stating that the approach is entirely neutral towards the binding interaction.

We thank all three Reviewers for their constructive criticism. We hope they will all agree that by addressing their points the manuscript has been made clearer and more impactful.

Referee #1:

Holder and Miles et al investigates Aurora A binding to CEP192 and builds a model on how this interaction leads to a handover to TPX2. Different structural methods are logically combined with gene-editing and cell biological assays, making this a very solid and interesting manuscript. The experiments are in my opinion well performed and the manuscript is well written. The spatio-temporal control of Aurora A activity is important for how mitosis is regulated. I am very positive to the manuscript but have some suggestions for minor changes.

We are pleased that the Reviewer finds our manuscript interesting, the experiments well performed and the data solid.

I miss a description of how Bora would fit into the model? The Aurora A-Bora complex is present independent of phosphorylation changes and could potentially compete with the other interactions described here (Lössl ACS 2016; Tavernier Nat comm 2021). A large part of Bora is degraded when PLk1 activity reaches the cytoplasm prior to mitosis, but a subset is stabilized in mitosis and overexpression or non-degradable versions of Bora have been suggested to impact on Aurora A-Tpx2 activity (Chan et al, Chromosoma 2008; Seki et al, JCB 2008; Feine et al cell cycle 2014; Bruinsma et al JCS 2014). I think a possible contribution of the degradation of a large part of the pool of Bora is relevant to mention in addition to the present discussion on CENP152, TPX2 and TACC3.

This is an excellent point. In our revision, we have included additional description of BORA in the Results and Discussion sections (lines 103-106, 627-632).

Fig 5H and EV5F. The FACS in EV5F does not include 8N population (just outside of figure). Were the gates set so that 8N cells are included in the statistics? Although it would not change any major conclusion, otherwise figure 5H is not correct and the wide range in polyploid populations could possibly be attributed to different PI intensity (which is apparent in EV5F).

There were no gates set so data in table (Figure 5J now) includes all >4N cells including 8N (i.e. 4N-G2/M) cells. The latter do not seem to form a tight peak but are visible on the FACS profile. We have also analysed the nuclear area in fixed cells to provide an independent measure of polyploidy (Figure EV4I).

Fig EV1. There seem to be a difference in size in Bora blot in A and B?

The blots have been misaligned, we have rectified this.

Fig 2C. Same color of text in fig for high and low exchange?

We have rectified these in the resubmitted manuscript and this is now Figure EV2.

Fig 2C and E. It is not indicated what color black represents.

The black in Figures 2C and 2E represents a region in the CEP192 sequence where there was no coverage in the mass spectrometry analysis. We have rectified this omission in the resubmitted manuscript and this is now Figure EV2.

Fig EV2C. Are these colors really blue and red as stated in text?

The colours/legends have been updated and this is now Appendix Figure S2.

Referee #2:

In this manuscript, Holder and colleagues investigate the structural determinants of the complex between the Aurora-A kinase and its main centrosomal partner, Cep192. By in vitro assays, crystallography and in silico approaches they identify and characterize two distinct regions of binding within the previously identified Aurora-A binding domain. They then explore the specific function of Aurora-A bound Cep192 by CRISPR-CAS9-mediated deletion of the Cep192 region involved in the binding. Cellular experiments confirm the importance of the Aurora-A/Cep192 axis in determining correct spindle length, organization and orientation and indicate the interaction as a pre-requisite for the formation of the Aurora-A/TPX2 complex on spindle microtubules, proposing Cep192 as a main determinant of spatio-temporal regulation of mitotic Aurora-A. Overall the study provides interesting novel observations going from structural characterization to cellular data. Still, I feel that prior to publication a number of experiments should be performed to support conclusions and better place the work in the context of the existing literature, as detailed below.

We are pleased that the Reviewer finds our study interesting and our findings novel, and appreciates the combination of structural and cellular characterisation provided in the manuscript.

Major points:

1. Throughout the manuscript, previously published data should be mentioned/discussed more accurately. "As expected" or "in line with previous reports" should be accompanied by citations. The mention of the structure resembling the α L region of CEP192 506-527 fused to the N-terminus of Aurora-A, with reference to Park et al., 2023, seems to be a critical point in the text. It's suggested that the work by Park is cited earlier in the paper for better contextualization. Additionally, a more in-depth comparison with the structure and data from Park's work is recommended to provide a clearer understanding of the similarities and differences between the two studies. This could enhance the comprehensibility and relevance of the current discussion in relation to the prior research.

As requested, we have included citations in the text wherever we refer to previously published work. We have referred to the Park et al. study earlier in the text (lines 153-155, 160-162, 304-308), and we also provide a more in-depth description and discussion of this study in light of our findings (lines 581-591). We would like to stress that our study was initiated 3 years ago, and our experimental design is based on our own structure, which we obtained well before this publication appeared. Therefore, we are happy to highlight the similarities and differences between our findings but our study cannot be considered a logical follow-up of theirs.

2. Experiments in Figure 1 and EV1 constitute a premise to the subsequent structural investigation, but in my opinion lack novelty. Selection of the 2 cell lines is based on a WB in asynchronous cultures, that in my experience is not very reliable for cell-cycle regulated proteins, and without providing a normalized quantification; mitotic extracts would have been more informative. The 2 lines are then not really compared, therefore I find this characterization unnecessary. Results mostly recapitulate what has been shown in previous papers and could largely be moved into Supplementary. Showing results in the non-transformed RPE cells may add novelty. Given the authors' intention to highlight the non-redundancy of TPX2 and Cep192, it would be interesting to see AurkA and p-AurkA levels and localization also in the Cep192i/TPX2i condition shown in EV1B.

We appreciate the reviewer's comments but decided to leave in Figures 1 and EV1 for the reasons outlined below. While the individual findings may not be that novel, we don't think this data is redundant as when starting our study we could not find publications where all co-factors and their effects on Aurora-A kinase and T loop phosphorylation were examined side-by-side whether in mitotic or asynchronous cells. Also there was no such data available from U251 cells at all. Results in Figs 1 and EV1 therefore serve to benchmark the effects of co-activator depletion against subsequent removal of CEP192:Aurora-A binding region in two cell lines. Furthermore, while we agree that comparing protein expression can be problematic in asynchronous cell lines, the cell lines examined have comparable cell cycle and mitosis durations (20-24 hours and ~40-60 minutes, respectively) and thus assessing relative expression of Aurora-A and its co-factors (all subject to cell

cycle control) in these cell lines is still useful. As suggested, we have now added western blots assessing the impact of siRNA depletion of co-factors in RPE1 cells (Fig. EV1C and D).

As for combined depletion of CEP192 and TPX2, this was very toxic in both U251 and RPE1 cells, unsurprising given that both proteins are essential for proliferation. Immunofluorescence microscopy revealed that the surviving cells invariably contained residual CEP192 at their centrosomes, indicative of incomplete depletion and/or selection (incomplete depletion of CEP192 is now mentioned in our main text-line 113, Fig EV1E). Since depletion of either CEP192 or TPX2 caused a marked loss of p-T288 signal already we decided not to persevere with these experiments and instead provide new western blots without co-depletion in our revised paper. Importantly, in Figures 6F and EV5A-B we show that P-T288 signal is >90% reduced in five independently derived Δ 11 clones (3 U251 and 2 RPE1 clones) providing ample evidence for the critical and non-redundant role of CEP192 in Aurora-A auto-phosphorylation, while confirming the well-established role of TPX2 in this process (e.g. Fig. 1A or 1C).

3. Page 6: authors attempt to validate Cep192-468-533 as the minimal region essential for interaction with Aurora-A by using ITC to determine binding affinity. Which is the rationale behind the selection of Cep192-468-533 ? The values obtained from ITC with Cep192-468-533 (71.4 nM in Fig S2 A, but 74nM in the text) are notably different from those obtained with fluorescence polarization (FP) in another phase of the study with Cep192-505-529 (Fig S2D, 1.4 μ M). How is this difference explained? FP analysis on Cep192-468-533 and ITC on Cep192-505-529 could strengthen the data. Importantly, a significant difference exists on a key point, i.e. by ITC the 442-533 fragment binds with a 70 nM Kd while similar fragments in Park et al. display μ M Kds. Can authors explain/comment on this?

The longer construct comprising 468-533 was chosen following the NMR and HDX analysis suggesting that the residues in this region are important for the direct interaction with Aurora-A (Figure 2). We are sorry if this was not clear before, but the revised manuscript has a more concise description of this data improves clarity, and we made an explicit statement (lines 162-163). The nanomolar binding was seen when either the 468-533 or 442-533 constructs of CEP192 were tested in ITC with the Aurora-A kinase domain (Now Figure 3C and EV2B, 80.6 nM and 72 nM). This affinity is higher than when just 501-533 of CEP192 is used as the longer constructs of CEP192 wrap around Aurora-A kinase domain, covering a large surface area (Kd for FAM-501-533 to Aurora-A is 1.17 μ M in Appendix Figure S3C).

The constructs used by Park et al in their ITC analysis contained an 40,300 Da MBP tag on CEP192 constructs, which were at most 21,722 Da (CEP192 440-631, ITC in Figure 1B of Park et al. 2023), which may have obstructed part of the binding site or cause steric hinderance. There is also a significant difference in buffer conditions used in these experiments, where we find that 10% glycerol is vital in the majority of our studies using Aurora-A. To add to this, our FP-analysis with the fluorescently labelled peptides based on CEP192 gave a very similar Kd to that seen by Park et al 2023 (this manuscript FAM-CEP192 501-533 Kd is 1.17 μ M Figure S3C, in black compared to Park et al. (2023) Figure 3B FAM-CEP192 506-536 0.7 μ M). And when we mutated residues in CEP192 that overlap with the TPX2 binding site (CEP192 468-533 Y487AF490A) we see a reduction in the affinity measured in the ITC (Kd of 2 μ M, Figure 3D) which matches well with the affinities measured by Park et al.

Crystals diffracted at 2.76 Å were formed only in the presence of Mb2, but the latter is not shown in any Figure. Could the Mb2 presence interfere with Cep192 binding to Aurora? Are there any contacts between Mb2 and Cep192?

The structure including Mb2 is shown in the supplementary figures (Figure EV3), but for reasons of clarity is not shown in the main figures. We have included analytical SEC data confirming that there

is no competition for binding and a three-way complex is formed (Appendix Figure S3 C and D, line number 181-182). In the crystal structure we see one potential interaction surface between CEP192 497-504 and a monobody in an adjacent complex (shown in Figure EV3C), but this can be accounted for by crystal contacts and scored an interface score of 0.00 upon PDBePISA analysis.

4. AurkA kinase assays suggest that Cep192 may act as an inhibitor of AurkA substrate phosphorylation. Given the overall activating function of Cep192 on AurkA in cells, the relevance of this inhibitory complex should be better investigated/discussed.

What we see in biochemical assays is incomplete inhibition, suggesting that CEP192 is involved in fine-tuning Aurora-A activity, not blocking it. In cells it appears that oligomerisation and concentration of Aurora-A at the centrosome by CEP192 is vital for the stimulation of Aurora-A activity. The minimal binding region may be able to hold Aurora-A in an inactive conformation until some threshold of oligomerisation/dimerization is obtained that we cannot recapitulate in an *in vitro* assay. This is discussed in the manuscript and we included a reference to the Joukov paper (Joukov et al 2010) to clarify how our observations build on this previous work (lines 611-618).

5. I find data with U251 Δ 11 cell lines confusing. WB blot and PLA characterization are missing. Based on FACS analysis results they are not analyzed further. On these bases I find U251 Δ 11 data are not informative in the actual form and should not be included.

We believe the U251 data is valuable as it confirms that CEP192(exon11) is required for Aurora-A auto-phosphorylation in a second cell line. In response to these comments, we have included further characterisation of Δ 11 U251 cells (P-T288 levels on western blots in Fig EV5A-B, CEP192:Aurora-A PLA in Fig 5H and I, Aurora-A:TPX2 PLA in Fig 6I and EV5D and NuMA quantification in Fig. EV6H-I) but did not study spindle morphology and mitotic progression in these cells due to their polyploidy, which can impede mitotic spindle assembly independent of CEP192(exon11) loss. To better illustrate increased ploidy of Δ 11 U251 cells, we have included additional data of nuclear size quantification (Figure EV4I).

6. Related to the previous point, the manuscript does not include analysis of chromosome segregation defects or genome instability in the Δ 11 cell lines. Based on the available data no conclusions should be drawn on genome stability (abstract, legends titles, discussion).

This is a valid point. In our previous submission we used the term genome instability based on our results from Δ 11 U251 cells; all three clones show an increase in 4N and >4N populations, and tetraploidy and polyploidy are known to generate genomic instability. While Δ 11 RPE1 did not show a marked change in ploidy, prompted by the Reviewer's comments, we tested if Δ 11 RPE1 cells exhibited chromosome missegregation. To this end, we boosted the number of mitotic cells by enriching cells in G2 with a CDK1 inhibitor. Cells were fixed 1 hour after the washout of the inhibitor, a timepoint when frequency of anaphase cells appeared the highest. This protocol revealed a significant 2-fold increase in lagging chromatids in Δ 11 cells relative to controls (Fig. 7C and D), thus providing conclusive evidence for elevated chromosomal instability in the mutants. Note that controls and Δ 11 cells are all functional null for p53, and therefore they are expected to tolerate mitotic delays and aneuploidy much better than parental RPE1 cells.

7. I have a concern about the choice to co-target p53 in the RPE cell lines. It has been shown that p53 has mitotic functions, in particular at centrosomes (Contadini et al., 2019). Did authors check the status of mitotic centrosomes in the WT-p53^{-/-} cell line vs controls? And, based on this and on p53/AurkA co-regulation, how can authors exclude a contribution of lack of p53 to the observed phenotypes?

We have not seen any evidence of the centrosome fragmentation phenotype described by Contadini et al. (their study concerned acute depletion of p53), indicating that constitutive loss of p53 may not have the same negative impact on centrosome integrity.

We have included new data (Figures 5D, EV7G) to show that gamma tubulin and CEP192 levels are comparable between multiple RPE-1 p53^{-/-} and parental RPE1 p53^{+/+} cell lines (except for one outlier control cell line, where levels are somewhat elevated but centrosomes still appear intact, Figure EV4D and EV7H). In our study, the number, morphology and maturation of centrosomes all seem normal not only in control RPE-1 p53^{-/-} but also in the Δ 11 p53^{-/-} clones (Figures 5D, EV6D and EV7F-I).

Given the obvious difference in p-Aurora-A levels and Aurora-A localisation between control and CEP192(exon11) knockouts both in U251 cells 'naturally' lacking functional p53 and RPE-1 p53^{-/-} cells, the phenotype must be predominantly due to CEP192(exon11) loss. Our attempts to isolate CEP192(exon11) knockouts in p53^{+/+} RPE1 cells have failed. This may be due to the mitotic delay observed in Δ 11 clones (Fig 7A,B). Such a delay could trigger the mitotic surveillance checkpoint in p53^{+/+} cells, resulting in irreversible cell cycle arrest and loss of p53^{+/+} Δ 11 clones.

8. Authors describe two RPE- Δ 11 lines but most shown results are in one of them (Δ 11-1) only. Main results should be shown in both lines.

As requested, we have included a more extensive analysis of the Δ 11-2 clone, such as quantitative immunofluorescence, proximity ligation assay, live cell experiments (Figures 5-7 and EV4-8). In fact, every measurement reported in Δ 11-1 has now been repeated in Δ 11-2. For all new experiments, the two control and two Δ 11 clones were processed and quantified together. For experiments where complete sets of data were already available for con1 and Δ 11-1 in our first submission, a separate analysis was performed in the (independently derived) con-2: Δ 11-2 clone pairs. Whenever data is obtained from experiments performed separately in con-1: Δ 11-1 and con-2: Δ 11-2 pairs, this is depicted by blank vs grey backgrounds on graphs (Figs. 5-7).

9. Is mitotic entry and progression normal in the Δ 11 cell lines? Are phenotypes measured in the same mitotic stages? In particular, are the pole-to-pole distance and spindle orientation measured in aligned metaphases only?

The reviewer makes an excellent point and we have addressed this in the following ways.

In our first submission, while we tried to bias our analysis toward metaphase cells, this was not straightforward due to mitotic spindles being off angle. In our revision, live cell imaging revealed a delay in anaphase onset (Fig. 7A, B), suggesting that the increased prometaphase-to-metaphase ratio could indeed skew spindle length measurements. While the same set of movies also confirmed increase in the incidence of anaphases occurring at an angle to the substratum in the mutants (Fig 7A), we decided to repeat spindle length and angle measurement using the proteasome inhibitor MG132 to arrest cells in metaphase, and replaced our previous analyses with this new data (Fig. 7E-G). Importantly, these results recapitulate our previous findings and indicate that even a prolonged metaphase arrest cannot revert the phenotypes caused by the loss of CEP192:Aurora-A binding.

10. Mitotic phenotypes. It is not obvious why the spindle diameter is measured on DAPI and not tubulin staining. Parallel to tubulin area, tubulin intensity would also be informative. More accurate measurements would be required to draw conclusions on MT length. Spindle orientation in RPE metaphase cells should not be much higher than 10 degrees. Having set a first class of 0-30 degrees is therefore misleading. Smaller intervals should be shown. Additionally, given that the pole-to-pole distance varies and this can influence the calculation of the spindle angle, raw data of z-distances should be provided to unambiguously show that orientation is affected.

Given that chromosomes in these rosettes are attached to microtubules, the diameter of the circle encompassing the chromosomes is considered a good approximation for microtubule length, and this measurement has been used in previous published studies (e.g., Fig 1C in Sobajima et al, 2023,

JCB). The spindle length and angle measurements are now shown in a box plot without binning (Fig. 7E-G). The raw values will be provided with the manuscript as source data.

11. Why is PLA measured by area? Intensity (or number of spots) should also be shown, together with a negative control for each interaction (particularly for the "diffused" signals).

We showed PLA area because number of foci was difficult to determine, especially when these focused around spindle poles. Prompted by the reviewer, we have decided to present intensities in our resubmission. Importantly, this has not affected the statistical significance of the data or our conclusions.

12. For a more robust demonstration of the conclusions drawn on the complexes with Cep192, TPX2 and TACC3 in the Δ11 cell lines it is important to perform AurkA immunoprecipitation experiments.

We made several attempts to perform these co-immunoprecipitations but failed to get clear results. We must add that such experiments are not trivial when using endogenous protein pools. In particular, as our PLA data highlights, Aurora-A and CEP192 interact predominantly in the centrosome. This pool however cannot be isolated with standard IP experiments because solubilisation of centrosomal CEP192 requires lysis conditions (high salt and high detergent) that are not amenable to preservation of protein-protein interaction. An alternative approach to improve chances of successful IPs is to overexpress the proteins of interest (frequently done in the literature for Aurora-A and its binding partners). We however wanted to avoid overexpression of a single component of the system because doing so can alter the balance, distribution and regulation of endogenous complexes, the subject of our study.

Given that Aurora-A:co-factor complexes are well-characterised including in-depth structural insights into binding interfaces, we believe that our well-controlled PLA experiments (Figs 5-7, for controls see Fig EV4E or Fig 7L)) are more informative than the IPs could ever be. PLAs show molecular proximity simultaneously with protein localisation, something IPs are not capable of. In addition, we can be certain that all our measurements are taken from prometaphase/ metaphase cells, which would not be the case for IPs. Lastly, similar results were obtained with PLAs from multiple Δ11 clones derived from two different cell lines.

Importantly, we did not base our conclusions on PLA alone; we also show subcellular localisation of the individual proteins, including Aurora-A and phospho-Aurora-A in cells, and also demonstrate loss of phospho-Aurora-A signal in mitotic Δ11 cell lysates (Figs. 5 and 6).

13. The conclusion that "the residual spindle localisation of Aurora-A in RΔ11-1 cells is still dependent on TPX2 because TPX2 depletion in these knockouts further diminishes the Aurora-A signal" should be supported by quantitative analysis.

We removed this data because we realised these results were obvious from PLA experiments where we showed that small amounts of TPX2 and Aurora-A still interacted in RΔ11 cells, in contrast to TPX2 depletion, which caused complete loss of Aurora-A from mitotic spindles (Fig. 1). Instead, we performed additional experiments in support of the newly discovered requirement for TPX2 in localising TACC3 (Fig. 7K-M).

14. Authors propose that the lack of effects on centrosome maturation are explained by the fact that "Deletion of CEP192(exon 11) should not impact the interaction between CEP192 and PLK1, and...centrosomal PLK1, previously activated by BORA:Aurora-A, facilitates PCM expansion". This needs support by experimental data by quantifying Plk1 localization and phosphorylation, and possibly by assessing AurkA levels and activation in G2 cells.

We have performed these analyses and found no obvious change in PLK1 localisation, T-loop phosphorylation and PLK1 substrate phosphorylation in mitotic $\Delta 11$ cells (Figs 6E and EV8)

15. pTACC3 should be tested to support the TACC3 proposed mechanism and an explanation for lack of p-Lats2 decrease should be provided. Eg5 is shown but not quantified. Since existing data indicate a decrease of centrosomal Eg5 in Cep192i, it would be interesting to know how it is in the $\Delta 11$ cells.

We quantified Eg5 intensity on spindles and found not obvious change in $\Delta 11$ cells (Fig. 7H-I). We also established that TACC3 localisation to the spindle and its interaction with Aurora-A are both TPX2-dependent (Fig. 7K-M). We did not study TACC3 phosphorylation, because currently we do not have a working antibody against phospho-TACC3. Furthermore, cellular and structural roles of Aurora-A-dependent phosphorylation of TACC3-S558 are well-established, in particular how this phosphorylation impact assembly of the TACC3-CHC-ChTog complex on its spindle recruitment (Barros *et al*, 2005; Booth *et al*, 2011; Burgess *et al*, 2018; Burgess *et al*, 2015; Cheeseman *et al*, 2011; Gergely *et al*, 2003; Giet *et al*, 2002; Kinoshita *et al*, 2005). Loss of TACC3 from spindles in $\Delta 11$ and Aurora-A inhibitor treated control cells can be seen/compared in Fig EV7A and B.

Regarding the p-Lats2 signal, in RPE1 cells we find that Aurora-A inhibition reduces its intensity by about 50%, whereas this figure is >80% in U251. This may be due to differences in the timing of Lats2 phosphorylation between U251 and RPE1 cells, or in p-Lats2 turnover in the centrosome. Nonetheless, our finding that p-Lats2 levels do not change in RPE1 $\Delta 11$ centrosomes suggests that i) Lats2 may be phosphorylated by TPX2:Aurora-A rather than centrosomal Aurora-A kinase, and ii) in $\Delta 11$ cells TPX2:Aurora-A (with no p-T288) is sufficient to phosphorylate Lats2. The latter is now mentioned in the text (lines 502-504).

16. Results in $\Delta 11$ cells are very similar to those previously obtained by RNAi/reconstitution or by transfection of dominant negative forms of Cep192. Interestingly, authors show that TPX2 interaction is also decreased, and propose that Cep192-induced AurkA autophosphorylation is required for TPX2 efficient binding. On the other hand, since Cep192 is required for localizing AurkA to the mitotic centrosome, it could also be that the decrease that they observe at spindle poles reflect this effect and that this then results in less AurkA/TPX2 interaction. More experimental data should be provided to support authors' model. For example, how is total AurkA and AurkA/TPX2 PLA in WT + Aurki condition? How does a kinase dead mutant behave?

With the ability to constitutively delete the Aurora-A binding domain of CEP192, we believe our gene edited cell-based experimental system is superior to previous work that combined transient silencing with transient overexpression of CEP192 and specific CEP192 domains. Nonetheless, it is reassuring that several phenotypes are shared between these studies and ours. However, the robustness of our engineered cells has allowed us to draw conclusions about the critical nature of CEP192 for accumulation of T-loop phosphorylated Aurora-A on spindles and in cells. Furthermore, our study is the first to show that TPX2 cannot facilitate/maintain T-loop phosphorylation of Aurora-A on its own without the CEP192-Aurora-A interaction. We were also surprised to see that the interaction is not essential for cell survival (at least in absence of p53), allowing for the first time the dissection of Aurora-A functions relying (or not) on a T-loop phosphorylated pool of the kinase.

It is a valid point that lack of centrosomal Aurora-A in $\Delta 11$ cells could be the cause of reduced Aurora-A-TPX2 interaction at spindle poles, and we have now mention this in lines 632-636. However, this does not contradict our model, because it still suggests that the interaction between CEP192 and the kinase is required for the effective spindle recruitment of Aurora-A onto the spindle by TPX2. Moreover, $\Delta 11$ cells have no detectable auto-phosphorylated kinase (at spindle poles or in cell lysates). Thus, despite TPX2 localising normally to the spindle and recruiting ~50% of Aurora-A in $\Delta 11$ cells, it cannot facilitate/maintain T loop-phosphorylation. Most likely, loss of auto-phosphorylation in $\Delta 11$ cells is a combined consequence of i) poor auto-phosphorylation outside of

centrosomes (lack of concentrating platform), ii) poor binding to TPX2 that would provide protection against PP1 and iii) lack of protection against phosphatases in the cytoplasm.

We decided against performing PLA between TPX2 and Aurora-A in the presence of Aurora-A inhibitors because the inhibitor completely removes Aurora-A from the spindle (Figure 1), whilst TPX2 recruitment to the spindle is unaffected by Aurora-A depletion (Kufer et al JCB 2002), showing the two proteins are largely in different pools under these conditions. Aurora-A inhibitors are also known to interfere with TPX2-Aurora-A interaction. Thus, results would be difficult to interpret and would add little additional insight if any.

We are also doubtful that overexpression of a kinase-dead Aurora-A would be informative. First, the main strength of our study is that it focuses on the behaviour of endogenous pools of Aurora-A kinase in cells. Overexpressing the kinase will inevitably skew the balance of these pools, thus hindering interpretation. Second, in WT cells, the kinase dead mutant is expected to get recruited to centrosomes where it may interfere with Plk1 activation/ CEP192 accumulation and centrosome maturation. This would contrast with $\Delta 11$ cells where the mutant kinase would be absent from centrosomes, which makes it difficult to control for the specificity of a phenotype.

Minor points:

- Page 5: the lack of a clear and concise explanation of the rationale behind the transition from CEP192-405-649 (Appendix Figure S1B) to CEP192-442-533 in the experimental design and analysis makes it challenging for readers to grasp the logical flow of the text. Clarification on this transition would significantly enhance the overall clarity and coherence of the text.

We have made it clearer in the manuscript that we selected the region aa468-533 as the minimal binding region in CEP192 (reduced from 406-649 and 442-533) because it encompasses the critical residues involved in Aurora-A binding (lines 162-163). This also happens almost entirely encoded by a single exon (exon 11, aa464-511) that was in-frame with the coding sequence, thus enabling removal of the complete exon without any further change to the coding sequence. The revised manuscript has a more concise description of this data binding data to improve clarity as well. Nanomolar binding was seen when either the 468-533 or 442-533 constructs of CEP192 were tested in ITC with the Aurora-A kinase domain (Now Figure 3C and EV2B, 80.6 nM and 72 nM).

- Fig. 1A. Histone H3-pS10, PPP1CA-pT320, PRC1-pT481 are included and not mentioned, as AurBi and Plki controls. Cep192 signal is not clear. Bora is missing and total AurkA is not quantified. Both 1A and EV1B indicate fluctuations of analyzed proteins in the RNAi samples that may be relevant but are not commented (e.g, EV1B, Cep192 decrease in TPX2i and Bora fluctuations, or in 1A TACC3 levels).

We now describe these markers in the text and/or figure legend. We do not show BORA in the blot in Fig 1A as BORA levels are almost undetectable in STLC-arrested mitotic cells (due to its PLK1-dependent degradation). Depletion efficiency is confirmed in asynchronous cells shown in Fig EV1B. We agree that there are fluctuations seen on the blots, some of which can be meaningful. We tried to comment on known changes in the manuscript but due to differences in cell viability, depletion efficiency, impacts on cell cycle progression and protein degradation between these samples, without further work we cannot draw firm conclusions. Describing these effects across the blots from Figs 1 and EV1 would distract from the points relevant to our study. We left in the blots, so readers can inform themselves and potentially pursue some of the interesting changes visible but are willing to remove these if asked. Importantly, these blots are not meant to provide conclusive evidence, they serve to highlight that in mitotic CEP192-depleted cells Aurora-A p-T288 levels are comparably low to those in TPX2-depleted cells (despite incomplete depletion of CEP192, which is now mentioned in the text). These results are in line with the massive drop in p-T288 signal relative to total Aurora-A at spindle poles in CEP192-depleted cells, and of course this is just the starting point for the rest of the paper.

- panels in Figures should be ordered in the order they appear in the text;
We have changed these accordingly.

- sometimes $\Delta 11$ -R1 is used instead of RD11-1

We have changed these accordingly.

- *cep192 $\Delta 11$ migration is not completely consistent in the WBs shown in Expanded View figures.*

Due to the large size of CEP192 (>250kDa), bands for CEP192($\Delta 11$) and wild-type CEP192, which differ only by 50 amino acids, can only be distinguished with increased duration of electrophoresis. However, the need to retain proteins of smaller sizes (ie P-histone H3) on the same gel sometimes prevented obvious separation of the mutant and wild-type forms of CEP192 proteins. We included asterisks to help visualise the shift between wild-type CEP192 and CEP192($\Delta 11$) in Fig 6E (left of blot).

Referee #3:

This manuscript by Holder, Miles, and colleagues reports that the centrosomal protein CEP192 plays a crucial and unexpected role in the accumulation of the activation loop-phosphorylated (and therefore active) form of the Aurora A kinase. Aurora A localises predominantly to the spindle poles and to the spindle during mitosis. It is known to achieve these different localisations by interacting with specific binding partners. TPX2 and TACC3 are spindle-associated partners of Aurora A, while CEP192 is a PCM component surrounding the centrosome. Yet another partner, Bora, is predominantly cytosolic. The authors performed a very thorough biochemical and biophysical characterisation of the CEP192-Aurora A complex. The main conclusions from this part of the work, which extends through Figures 2-4, are that 1) CEP192 binds competitively with the other above-mentioned factors (TPX2 and TACC3); 2) It may act as an activator under specific conditions, but in its most fundamental, high-affinity binding state, it seems to act as an inhibitor of Aurora A kinase activity. In the rest of the manuscript, described in Figures 1 and 5-6, the authors present data that CEP192 appears to act at the apex of an activation pathway for Aurora A and is essential for the accumulation of the active form of Aurora A, identified by phosphorylation on T288 (P-T288). The authors reach this conclusion not only through more classical RNAi-based depletion experiments, but also by generating cell lines expressing a mutant CEP192 deleted of the Aurora A binding region. Collectively, these experiments revealed that CEP192 is required for cells to achieve a normal spindle lengths and spindle orientation relative to a substrate, and more generally to maintain the correct ploidy.

Overall, the results presented in this manuscript support the concept that CEP192 is a master regulator of Aurora A activation. This conclusion emerges with clarity from the presented experiments and is surprising and exciting. The mechanistic part of the work and the cell biological analyses do not bring about a fully coherent picture yet. The reason for this is that there is no clear explanation for why a potential biochemical inhibitor of Aurora A, like CEP192, is also absolutely necessary for Aurora A activation. This should not be seen as a major limitation in our opinion, as the main conclusions of this very data-rich study are sufficiently novel to guarantee the general interest of the paper. So we are supportive of the work and would like to see it published. However, we would also like the authors to consider the following points (in no specific order):

We are pleased that the Reviewer finds aspects of our study surprising and exciting.

-The presentation of the manuscript could be improved significantly. We encourage the authors to consider the following five main points:

a) The Abstract is not sufficiently informative and should be rewritten to communicate the papers's main points. E.g. which "spindle-associated partners"? The following sentence repeats the concept that the binders are competitors, but forgets to highlight the interesting apparent contradiction that CEP192 is biochemically an inhibitor, which is not to be read anywhere in the abstract. The apparent paradox with the biological observations is not highlighted. We would recommend avoiding using "license" in the absence of clear evidence of a licensing step. CEP192 is necessary for the accumulation of phosphorylated Aurora A. How it does it, remains unclear. The final sentence seems a cut-and-paste from troves of other papers where authors hope to awake the editor's interest by invoking genome stability. We would recommend giving a more specific and informative conclusion.

We have taken these points into account in our revision and removed 'licensing' from the abstract. We have decided to leave in our reference to genome stability because in addition to the polyploidisation of U-Δ11 cells we now also show chromosome segregation errors in RΔ11 cells.

b) The description of the structural work feels endless. Sorry, we don't want to sound insulting and we are aware of the massive amount of work that went into Figure 2-4, but that remains a poor reason to launch a lot of apparently redundant data onto readers. In essence, we think that the crystal structure (obtained through elegant thinking on how to eliminate unwanted crystal contacts) is just enough for the main paper, and the rest (HDX, NMR, AF2 predictions), being confirmatory, should be presented in the extended or supplementary data. I.e. we would only present Figures 3 and 4 and would GREATLY streamline the description of the other approaches, which should be presented rapidly and as confirmatory of the structure.

We have taken these suggestions into account when preparing our revision. We understand that Figure 2 is more confirmatory given that we were able to solve the crystal structure. We have now moved this to be Figure EV2. The description of the HDX and NMR in the main text is concise (lines 141-148).

c) A major point of the paper is that CEP192 operates upstream of TPX2, but the evidence supporting this notion is limited if at all existing. Depletion of TPX2 leads to an equally dramatic disappearance of Aurora A activation loop phosphorylation. Without clearer mechanistic evidence, this conclusion seems unsupported.

We do not fully understand why the Reviewer questions whether CEP192 operates upstream of TPX2 at least in terms of T loop phosphorylation. TPX2 is clearly essential to achieve normal pT288-Aurora-A levels in cells (see below), but our results argue that instead of CEP192 aiding T-loop phosphorylation of Aurora-A at centrosomes, and TPX2 simultaneously doing the same on the spindle, it is only CEP192 that can generate pT288-Aurora-A in cells. The near complete loss of pT288-Aurora-A signal from mitotic $\Delta 11$ cell lysates is particularly striking as these contain normal TPX2 levels.

If CEP192 and TPX2 acted independently (i.e., in parallel), Aurora-A and pT288-Aurora-A levels should plummet in centrosomes of $\Delta 11$ cells but their (TPX2-dependent) spindle pool should remain unaffected. Instead, despite TPX2 being expressed and localising normally in mitotic $\Delta 11$ cells, p-Aurora-A signal disappears not only from centrosomes but also from spindle poles (and from cell lysates) accompanied by a ~50% drop in Aurora-A levels on spindles. Therefore, without the CEP192-Aurora-A interaction, TPX2 can neither recruit/maintain normal levels of Aurora-A nor stimulate/maintain Aurora-A T loop-phosphorylation. By contrast, in the absence of TPX2, CEP192 still recruits Aurora-A to the centrosome and generates T-loop phosphorylated kinase (Fig. 1).

Nevertheless, cellular p-Aurora levels are indeed very low in TPX2-depleted cells but that is because pT288-Aurora-A generated at centrosomes will neither get recruited to the spindle nor will it enjoy TPX2-dependent protection from PP1. Thus, even though TPX2 cannot seem to generate pT288-Aurora-A (without CEP192), TPX2-dependent removal of p-Aurora-A from the centrosome and the protection it provides from PP1 seem absolutely crucial for amplifying p-Aurora-A levels. We have made these points clearer in our model and emphasised that the upstream/downstream terms refer purely to p-Aurora-A generation (**line 677**).

d) We feel that the paper would make for an easier read if it began with the structure and biochemistry, and moved on to describe the surprising observation that CEP192 seems to be an activator, not an inhibitor. Figure 5 seems like a logical follow up to Figure 1, so we would suggest that the authors use Figure 1 as the first figure of the biological characterisation after the description of the structure. A most exciting result presented in Figure EV6A should be brought to the main figures unless the authors had a concern with it justifying its relation in the Extended result.

Having taken into account comments from all three reviewers we have decided to keep the original order as the profound effect of CEP192 depletion on p-T288 levels was the trigger for our study. We have added data as requested by Reviewer 1 to Figure EV1 and have now included Fig EV6A in the main figure while providing extra quantification and repeats with all clones.

e) A comparative discussion of the results from the Park et al. Science Advances 2023 paper would add clarity. There are similarities and differences between the papers, and the authors have to try to rationalize them for their audience.

We have included a more in-depth description and discussion of the Park et al study in light of our findings (lines 581-591). Just to note, we initiated this study nearly 3 years ago, so the Park et al was published after we obtained our structure and generated some of the knockout cell lines, and so our experiments were designed without prior knowledge of their data.

-Figure 1A, the depletion of CEP192 in this particular blot does not seem striking. It is much clearer in EV1.

We agree. It is difficult to achieve strong depletion of CEP192 within 48 hours, because the protein is highly stable and is also essential for cell survival. Indeed, siRNA-treated mitotic cells almost invariably have a small pool of CEP192 in their centrosome, indicating partial depletion. Although increased duration of siRNA treatment could certainly improve depletion efficiency, this will yield acentriolar cells, a scenario we would like to avoid due to its complex effects on spindle assembly. We have highlighted in the text that we achieve only partial depletion of CEP192.

-Panel 2E is described earlier than most other panels in the figure

We have reordered the panels so that they are referred to in the correct order.

-Hard to understand the logic of colour assignments in panels 2C-E. Colours for high and low exchange are too similar and hard to distinguish. Second, we assume there is a third category of rates of exchange that remain unaltered in the bound and unbound state. Shouldn't those be indicated as well with an appropriate colour?

We have ensured that the regions where there is no difference are clearly assigned a colour and that all of the categories of rates of exchange are correctly mentioned in the updated figure legend. This is now Figure EV2.

-Figure 3A: Please indicate N and C-termini at least for CEP192 and name the main secondary structure elements (e.g. alphaS, alpha L)

We have added these labels to Figure 2 in the final version of the manuscript.

-Figure 4G is too small and packed to be effective at making the important point it is trying to make. It is too out of context (which part of the structure is being shown?) for most readers to interpret. One of the structures of Aurora A-CEP could be removed, just TPX2 vs one of the CEP structures could be retained. An associated problem is that this figure only makes sense after discussion of panel H, as it provides an explanation for the paradoxical effects of the two parts of CEP192 that interact with Aurora A. This is an exciting point, but it needs to be presented in a way that readers can appreciate it. In 4H, why not show all Aurora A with the same color, rather than changing color for every panel? This would apply also to the CEP192 species. Overall, this is an important figure. Maybe consider splitting figure 4 after panel F and create a new figure, which will be possible if Figure 2 is moved into the Supplementary.

Thank you for your suggestion, we have split Figure 4 to produce a new figure (now Figures 3 and 4) and to allow us to increase the size of some panels. We have also recoloured Figure 4H (now Figure 4B) to make it easier to comprehend, keeping Aurora-A the same shade of green throughout.

-Figure 4C-E and every other sensor-gram, when multiple mutations are present, please separate them with a hyphen or slash

Thank you for your suggestion, this edit has been made to the manuscript.

-Page 3, first line "...found at the centromeres and at the kinetochores"; In the following line replace "the spindle assembly checkpoint" with "a pathway" (reason being that Aurora B is credited to work in error correction, whereas its role in the SAC is still considered by many less well established)

We have rectified this.

-Page 4, the paragraph starting "Consistent with previous studies..." is somewhat confusing. First, it is unclear to us that the authors really quantified spindle localization. "Despite more Aurora-A..." We assume the authors mean "relative to TPX2" but it is not immediately clear and we would suggest being explicit.

Good point. We have re-phrased this sentence and made it clearer that we quantified spindle pole localisation and that we were indeed comparing CEP192 depletion to that of TPX2.

-Page 6. Please rewrite: "The longer (CEP192-442-533) and minimal (CEP192-468-533) CEP192 constructs....Aurora-AM3KD: 72 nM and 74 nM, respectively (Figure 2B and AF S2A)

We have done so.

-Consider adding "Thus, in summary..." to the sentence starting "Using three tag-free..."
Thank you for your suggestion, this edit has been made to the manuscript.

-Page 7: "Hot-spot" of what? Mutation?

We have clarified this part of the manuscript to add in a reference to further details about the F-pocket in Aurora-A kinase.

-Page 7: Is the predicted "solvation energy" dimensionless? What is the source of the reported number, 2.37?

The predicted solvation energy effect of residues in the interface between CEP192 and Aurora-A was calculated using PDBE PISA (10.1016/j.jmb.2007.05.022). Thank you for spotting that the units have been omitted from the manuscript, this has now been amended in the manuscript (kcal/mol).

-Page 7, last paragraph, second line "(Figure 3C)" We think it should read Figure 3D

Thank you for spotting this, the text has been edited to rectify this.

-Page 8, first paragraph: "Kd values > 5 μ M" should come together with a value for the control.

Thank you for spotting this, the text has been edited to rectify this.

-Page 8, first paragraph: in "and also showed", "and" should be removed.

Thank you for spotting this, the text has been edited to rectify this.

-Page 10, first paragraph: "had no significant effect on the kinase activity"; but the activity appears to increase.

We agree there is a slight increase in activity seen in the presence of the CEP192 501-533 sequence in the ADP-Glo assay. We have edited the text to indicate that this activation is seen (lines 302-303 and 615-616).

-Figure 2 legend "over by over"

We have changed this.

-Figure 5I-J, maybe easier to use grayscale? The two blue tones are not easily recognised in printouts

We have obtained new data related to Figure 5J and these are now shown in Figure 7 with a different colour scheme.

-Figure 7A: green for INCENP and green for Aurora A makes color resolution really poor. Add tags to C- and N-termini and consider replacing the arrows.

We have edited 7A (now Figure 8A) to make the colour resolution clearer and added the suggested tags.

-Figure EV1B: does TPX2 control the stability of CEP192?

We think this may be an anomaly of this western blot as in other experiments or in cells we found no evidence of TPX2 levels influencing CEP192, or the centrosomal pool of CEP192 in particular.

-"Deletion of CEP192(exon11) should not impact the interaction between CEP192 and PLK1." This is based on the idea that PLK1 binds to a different region, but Joukov et al. 2014 suggested co-dependence of PLK1 and Aurora A on CEP192. This point is somewhat tangential to the paper, yet quite relevant and the authors could

have elected to test PLK1 localization. In the absence of a direct test, they could nevertheless refer to the observations of Joukov et al.

We find that PLK1 localisation is normal in $\Delta 11$ cells and have included this new data (Fig. EV8)

-Figure 5A-B: The scheme could be made clearer. Why is CEP192 depicted in blue and grey in panel A, green and grey in panel B (grey in scheme of excision sites). Why not use color consistently? "PLK1 and 4 binding" implies there is a single bipartite binding site for both, or individual binding sites currently displayed in red and dark red?

We have changed the scheme to improve clarity.

-Figure 5C-E is a control and could be moved to EV or SF figures.

We added new data to these figures and decided to leave these in the main figure as they validate the cellular model used for the rest of the paper.

-Figure 4F reports effects of CEP192 on Aurora A activity. CEP192 (468-533) is shown to be inhibitory (IC50 18 nM compared to alisertib 3 nM), while the region 500-533 is slightly activatory (claimed to have no significant effect). This analysis would benefit from a comparison of the authors' results with those of Park et al. 2023, who did not identify inhibitory activity with a longer fragment of CEP193. The authors claim that their results are consistent with those of Joukov et al. 2010. If we correctly understood Joukov et al. 2010, they obtained results with a dominant-negative approach where overexpression of the Aurora A binding region of CENP193 caused Aurora A displacement from the poles, and failed activation due to decreased local concentration. It is difficult to compare these results with those from a biochemical assay in vitro.

Joukov et al. (2010) Figure 3G shows an *in vitro* assay where recombinant GST-tagged CEP192 521-757 (encompassing the binding site in *Xenopus* CEP192) was added to complexes isolated from *Xenopus* egg extract using anti-Aurora-A coated beads. Western blot then indicated that the presence of CEP192 521-757 prevents autophosphorylation of Aurora-A at Thr288 and phosphorylation of the Aurora-A substrate Histone H3. In another section of the publication *Xenopus* egg extract was used to look at the effect of CEP192 521-757 on Maskin/TACC3 phosphorylation by Aurora-A (Figure 4C). CEP192 521-757 is able to inhibit the activity of Aurora-A again, indicated by lower levels of Maskin/TACC3 phosphorylation. Although these techniques that were used are not entirely the same as the *in vitro* ADP-Glo based approach, we feel that in a combination of different experiments, Joukov et al. see this inhibition as well with a fragment of CEP192.

In the publication by Park et al. (2023), there is no effect seen when either a longer fragment of CEP192 (1-995) or CEP192 506-536 were included in an ADP-Glo based assay (Park et al. 2023 Figure 3 F-H). However, the previous assay conditions were not optimal for assessing inhibition as the Aurora-A activity was low. Furthermore, the CEP192 1-995 used in their assay is not characterised: nothing is shown in the paper about the oligomerisation state, purity of this material or the strength of the binding between CEP192 1-995 and Aurora-A, which would help to make a comparison easier. They also used a different short fragment of CEP192 (506-536) that lacks the short helix present in our fragment. We have added a brief comparison of these data into the revised results section (lines 304-308), and re-emphasised that the activation we see is much less than that mediated by TPX2.

-Figure 4F: Another concern regarding this figure is that the authors use Aurora A at 10 nM concentration. Hard to see how TPX2 could cause 50 % max activation at 1 nM concentration. At this concentration, even assuming the highest possible binding affinity, only 10 % of the 10 nM kinase can be saturated (1 nM of 10 nM; half-maximal is expected at 5 with infinitely high affinity for a 1:1 complex). Furthermore, half maximal activation at 1 nM implies TPX2 binds Aurora A at this concentration, which would contradict the authors' claim that CEP192 is the tightest binding partner of Aurora A (70 nM).

We agree that you would expect the EC₅₀ to be not be as low as we have stated. This may be because the fit is poor and so we have repeated this assay and update the figure in the revised manuscript. The calculated EC₅₀ is now 6.8nM, which is compatible with 50% activation of the kinase at a concentration of 10 nM (Appendix Figure S4D).

We have also revised the manuscript to remove the idea that CEP192 has a higher affinity than TPX2. In the literature there seems to be a wide range of affinities measured between Aurora-A and TPX2. The highest published affinity is 9 nM (Tavernier *et al*, 2021), (FAM-TPX2 1-43 in a direct binding assay with Aurora-A (Figure 6B)). Whereas in other publications the K_d has been measured in the low micromolar range (Hadzipasic *et al*, 2020; McIntyre *et al*, 2017; Rennie *et al*, 2016; Zorba *et al*, 2014) We hypothesise this is due to different conditions for these experiments.

-Figure 3: Aurora A M7KD interacts with K_d 1.4 μM to FAM-CEP192 peptide, while CAKD construct K_d was 420 nM. The author state it is similar, but it is a 3-fold difference. Furthermore, another binding domain, Mb2, was added to improve diffraction quality. To understand this system quantitative, it would be necessary to test if there is any difference in affinity with and without Mb2. The authors should not worry about this, but should probably be more cautious stating that the approach is entirely neutral towards the binding interaction.

A three-way complex between the Aurora-A kinase domain, CEP192 468-533 and the monobody survived size exclusion chromatography to provide the material for use in the crystallisation trials. We have clarified this further through the use of analytical SEC to confirm that the presence of the monobody does not affect binding of Aurora-A to CEP192 468-533 (Appendix Figure S3C and D). It is clear that the presence of the monobody has no significant effect on the binding between Aurora-A and CEP192.

We have repeated the fluorescence anisotropy to measure the interaction of FAM-CEP192 501-533 with the different versions of Aurora-A (CAKD and M7KD) to confirm whether any difference is seen (Appendix Figure S3B). This repeated assay gave us a K_d of FITC-CEP192₅₀₁₋₅₃₃ to Aurora-A^{M7KD} (K_d 1.25 μM ± 0.15 μM) and Aurora-A^{CAKD} (K_d 1.17 μM ± 0.07 μM).

References

- Barros TP, Kinoshita K, Hyman AA, Raff JW (2005) Aurora A activates D-TACC-Msps complexes exclusively at centrosomes to stabilize centrosomal microtubules. *J Cell Biol* 170: 1039-1046
- Booth DG, Hood FE, Prior IA, Royle SJ (2011) A TACC3/ch-TOG/clathrin complex stabilises kinetochore fibres by inter-microtubule bridging. *EMBO J* 30: 906-919
- Burgess SG, Mukherjee M, Sabir S, Joseph N, Gutierrez-Caballero C, Richards MW, Huguenin-Dezot N, Chin JW, Kennedy EJ, Pfuhl M *et al* (2018) Mitotic spindle association of TACC3 requires Aurora-A-dependent stabilization of a cryptic alpha-helix. *EMBO J* 37: e97902
- Burgess SG, Peset I, Joseph N, Cavazza T, Vernos I, Pfuhl M, Gergely F, Bayliss R (2015) Aurora-A-Dependent Control of TACC3 Influences the Rate of Mitotic Spindle Assembly. *PLoS Genet* 11: e1005345
- Cheeseman LP, Booth DG, Hood FE, Prior IA, Royle SJ (2011) Aurora A kinase activity is required for localization of TACC3/ch-TOG/clathrin inter-microtubule bridges. *Commun Integr Biol* 4: 409-412
- Gergely F, Draviam VM, Raff JW (2003) The ch-TOG/XMAP215 protein is essential for spindle pole organization in human somatic cells. *Genes Dev* 17: 336-341
- Giet R, McLean D, Descamps S, Lee MJ, Raff JW, Prigent C, Glover DM (2002) Drosophila Aurora A kinase is required to localize D-TACC to centrosomes and to regulate astral microtubules. *J Cell Biol* 156: 437-451
- Hadzipasic A, Wilson C, Nguyen V, Kern N, Kim C, Pitsawong W, Villali J, Zheng Y, Kern D (2020) Ancient origins of allosteric activation in a Ser-Thr kinase. *Science* 367: 912-917

- Kinoshita K, Noetzel TL, Pelletier L, Mechtler K, Drechsel DN, Schwager A, Lee M, Raff JW, Hyman AA (2005) Aurora A phosphorylation of TACC3/maskin is required for centrosome-dependent microtubule assembly in mitosis. *J Cell Biol* 170: 1047-1055
- McIntyre PJ, Collins PM, Vrzal L, Birchall K, Arnold LH, Mpamhanga C, Coombs PJ, Burgess SG, Richards MW, Winter A *et al* (2017) Characterization of Three Druggable Hot-Spots in the Aurora-A/TPX2 Interaction Using Biochemical, Biophysical, and Fragment-Based Approaches. *ACS Chem Biol* 12: 2906-2914
- Rennie YK, McIntyre PJ, Akindele T, Bayliss R, Jamieson AG (2016) A TPX2 Proteomimetic Has Enhanced Affinity for Aurora-A Due to Hydrocarbon Stapling of a Helix. *ACS Chem Biol* 11: 3383-3390
- Tavernier N, Thomas Y, Vigneron S, Maisonneuve P, Orlicky S, Mader P, Regmi SG, Van Hove L, Levinson NM, Gasmi-Seabrook G *et al* (2021) Bora phosphorylation substitutes in trans for T-loop phosphorylation in Aurora A to promote mitotic entry. *Nat Commun* 12: 1899
- Zorba A, Buosi V, Kutter S, Kern N, Pontiggia F, Cho YJ, Kern D (2014) Molecular mechanism of Aurora A kinase autophosphorylation and its allosteric activation by TPX2. *Elife* 3: e02667

Dr. Fanni Gergely
University of Oxford
Department of Biochemistry
South Parks Road
Oxford OX1 3QU
United Kingdom

6th Aug 2024

Re: EMBOJ-2024-116662R
CEP192 drives spatio-temporal control of mitotic Aurora-A

Dear Fanni and Richard,

Thank you for submitting your revised manuscript for our consideration. It has now been re-reviewed by two of the original referees, who -as can be seen from the reports below- were fully satisfied with the revisions. Following addressing of the following editorial issues, we shall therefore be happy to accept the study for The EMBO Journal:

- Please upload all main Figures and all Expanded View (EV) figures as individual files, with sufficient resolution/quality for production. Both main and EV figure legends should be collated at the end of the manuscript text. Importantly, since our Expanded View figure content is limited to 5, please only retain the most important ones of the current 8 EV Figures here, and move the others into the Appendix (making sure to update all Figure names and in-text call-outs accordingly).
- Please use the attached "Reagents and Tools Table" template for entering the information currently provided as "Table 1", and upload it separately. Please consider moving the other tables 2-3, currently included in the method section, into the Appendix, given that there is already an Appendix Table S4 containing oligonucleotide sequence information. (For detail, see <https://www.embopress.org/page/journal/14693178/authorguide#structuredmethods>)
- In the Appendix, please make sure to adhere to the nomenclature "Appendix Figure S1/2..." and "Appendix Table S1/2/..." throughout (also in the respective legends), and make sure to reference each Appendix Table at least once from the main text or from the Reagents & Tools Table. Please rename the included reference section into "Appendix References" and include it as an item in the Appendix table of contents. Finally, the "Appendix Source Data" on pp. 19-21 should either be uploaded as separate Source Data files (if they are raw data directly connected to any of the main/EV/Appendix figures), or renamed as additional Appendix Figures and referenced from the text in appropriate places (e.g., from figure legends or methods sections).
- Please reorganize the uploaded Source Data so that individual ZIP archives per figure are only provided for the main figures. Source Data for EV Figures and for Appendix Figures should be combined in one ZIP for EV and a separate ZIP for Appendix Figures.
- On the abstract page of the manuscript, please include 4-5 general keyword terms to enhance searchability.
- As we are switching from a free-text author contribution statement towards a more formal statement based on Contributor Role Taxonomy (CRediT) terms, please remove the present Author Contribution section and instead specify each author's contribution(s) directly in the Author Information page of our submission system during upload of the final manuscript. See <https://casrai.org/credit/> for more information.
- In the Data Availability section, please include direct URLs pointing to the datasets deposited in PDB, BMRB, and BiImage Archive, respectively; and make sure to initiate prompt public release upon formal acceptance.
- During routine pre-acceptance checks, our data editors have raised the following queries regarding figures, data, and legends, which I would ask you to address (ideally using the Track Changes option):
 1. Please provide the EXACT p-values in the legends of Figures 1b, d-e, g; 5d, g, i; 6c-d, f, h-i; 7a, d, f, i-j, m; EV 1d; EV 4d, i; EV 6a-b, i; EV 7b, d, f, h; EV 8d-e.
 2. Please note that in Figures 6c-d, f, h-i; there is a mismatch between the annotated p-values in the figure legend and the annotated p-values in the figure file - this should be corrected.
 3. Please note that the box plots need to be defined in terms of minima, maxima, centre, bounds of box and whiskers, in the legends of Figures 1d-e, g; 5d-e, g; 6c-d, h-i; 7a, f-g, i-j, m; EV 4d, i; EV 6b, i; EV 7b, d, f-i; EV 8b.
 4. Although 'n' is provided, please describe the nature/entity for 'n' in the legends of Figures 1b; 6f; 7d; EV 1d; EV 5a, c; EV 6a; EV 8d-g.
- Please provide suggestions for a short 'blurb' text prefacing and summing up the study in two sentences (max. 250 characters),

followed by 3-5 one-sentence 'bullet points' with brief factual statements of key results of the paper; they will form the basis of an editor-written 'Synopsis' accompanying the online version of the article, together with the already provided synopsis image.

- Finally, please consider a somewhat more explicit title, instead of the somewhat colloquial and ambiguous phrasing "drives spatio-temporal control"

I would appreciate if you could make these requested changes and upload all finalized files within the next two days, in order to expedite our final checking and prevent unnecessary delays with acceptance and publication of the study.

With kind regards,

Hartmut

9) To facilitate reproducibility and cross-laboratory adoption of methodologies, please structure the Materials & Methods section as outlined in our guide to authors, including a completed Reagents and Tools Table that can be downloaded from our author guidelines as well (<https://www.embopress.org/page/journal/14602075/authorguide#structuredmethods>).

10) Digital image enhancement is acceptable practice, as long as it accurately represents the original data and conforms to community standards. If a figure has been subjected to significant electronic manipulation, this must be clearly noted in the figure

legend and/or the 'Materials and Methods' section. The editors reserve the right to request original versions of figures and the original images that were used to assemble the figure. Finally, we generally encourage uploading of numerical as well as gel/blot image source data; for details see: embopress.org/page/journal/14602075/authorguide#sourcedata

At EMBO Press, we ask authors to provide source data for the main manuscript figures. Our source data coordinator will contact you to discuss which figure panels we would need source data for and will also provide you with helpful tips on how to upload and organize the files.

In the interest of ensuring the conceptual advance provided by the work, we recommend submitting a revision within 3 months (4th Nov 2024). Please discuss the revision progress ahead of this time with the editor if you require more time to complete the revisions. Use the link below to submit your revision:

Link Not Available

Referee #1:

The authors have addressed my suggestions for minor changes. As I outlined in my review of the first version, I am very positive to this manuscript.

Referee #3:

I am grateful to the authors for submitting a greatly improved revision of their manuscript. I strongly support publication of this manuscript in The EMBO Journal.

Dr. Fanni Gergely
University of Oxford
Department of Biochemistry
South Parks Road
Oxford OX1 3QU
United Kingdom

3rd Sep 2024

Re: EMBOJ-2024-116662R1
CEP192 localises mitotic Aurora-A activity by priming its interaction with TPX2

Dear Fanni,

Thank you for submitting your final revised manuscript for our consideration. I am pleased to inform you that we have now accepted it for publication in The EMBO Journal.

With kind regards,

Hartmut
